# The Stochastic Toolbox User's Guide - xSPDE3: extensible software for stochastic ordinary and partial differential equations

Simon Kiesewetter[1], Ria R. Joseph[1,2] and Peter D. Drummond[1]⋆

**1** Centre for Quantum Science and Technology Theory,  Swinburne University of Technology, Melbourne, Victoria, Australia.
**2** School of Information Technology,  Deakin University, Melbourne, Victoria, Australia.
⋆ peterddrummond@protonmail.com

March 8, 2023

## Abstract

**The xSPDE toolbox treats stochastic partial and ordinary differential equations, with applications in biology, chemistry, engineering, medicine, physics and quantum technologies. It computes statistical averages, including time-step and/or sampling error estimation. xSPDE can provide higher order convergence, Fourier spectra and probability densities. The toolbox has graphical output and $\chi^2$ statistics, as well as weighted, projected, or forward-backward equations. It can generate input-output quantum spectra. All equations may have independent periodic, Dirichlet, and Neumann or Robin boundary conditions in any dimension, for any vector field component, and at either end of any interval.**

# 1   Introduction

## 1.1   The xSPDE distribution

***xSPDE is an eXtensible Stochastic Partial Differential Equation solver*** [1].

xSPDE has functions that can numerically solve both ordinary and partial differential stochastic equations of any type, obtaining correlations, probabilities and averages. There are many equations of this type [2–6] in physics, chemistry, engineering, biology, medicine, and finance.

Previous applications are in physics and quantum technology [7–26], but the code has general applicability. The emphasis in xSPDE is on combining a simple user interface with a wide range of useful functions, including the essential features of averaging and global error estimates. The code enables an efficient use memory and parallelism, which is vital for large stochastic models, and it is able to be further extended if needed.

The extensible structure of the code-base permits drop-in replacements of the algorithms. Different simulations can be carried out sequentially. This models different stages in an experiment or simulated environment. It can be used with or without noise terms, and can use a range of either built-in or user defined integration algorithms. This user guide describes xSPDE3, which is an improved version of an earlier toolbox [1].

xSPDE calculates and plots averages and probabilities of arbitrary functions of any number of complex or real fields, as well as Fourier transforms in time or space with any given dimensionality. Importantly, it gives error estimates for both the discretization and sampling error, but the algorithm, the step-size and the number of samples used is up to the user to control to obtain the required error levels.

Ordinary and stochastic differential equations of many types can be treated numerically [27, 28], including stochastic partial differential equations with space dependence [29]. Comparative $\chi^2$ statistical tests are available. Additional libraries exist for projected, forward-backward, and weighted equations.

The algorithms included are designed to be useful and fast in many practical applications. Higher order convergence is obtained through order extrapolation. This allows higher-order convergence to be realized in a uniform way. More complex higher-order algorithms are known [28, 30], which can be included if preferred, as the code is extensible.

The code can be used interactively or in batch mode. All graphs, data, and input parameters, including default values, can be stored permanently using standard file-types. It has a fully integrated graphics program, xGRAPH. This is able to handle data of any dimensions, with multiple types of graphical output, error-bars and comparisons.

xSPDE supports parallelism at both vector instruction and multiple core level using array and parallel loop syntax. This version is Octave/Matlab based. Matlab is a commercial product, GNU Octave [31] is free and open-source. They each have excellent user interfaces and reliable implementations. Full parallel operation currently requires the Matlab parallel toolbox.

Using the toolbox can be compared to dining in a very friendly restaurant. It allows you to choose recipes from a large menu. It does the work of solving equations and doing averages. Estimating errors and graphing results is carried out as well. But it has a communal kitchen too. If you want more choice, you can add your own dishes.

See: **www.github.com/peterddrummond/xspde_matlab.** For those familiar with earlier versions, a list of the main xSPDE changes since the documentation of the previously published

version (v1.04) [1] is as follows:

1. More graphics options, color-mapped contours, batch graph storage

2. Computational arrays with the ensemble index last for more speed.

3. Scatter plots with multiple lines for individual trajectories

4. Expansion of packed internal arrays combined with index broadcasting

5. Extended documentation with new examples, key-words and error-messages

6. Auxiliary fields for storing random noises and quantum input-output spectra

7. Methods for weighted and projected stochastic differential equations

8. Computational methods for forward-backward stochastic equations

9. Finite-difference solutions to SPDEs with nonlinear boundaries

10. Fast DST and DCT spectral methods for SPDEs with non-periodic boundaries

11. Multidimensional comparison functions and experimental data input

12. Chi-squared tests and probability densities of arbitrary functions

There are some internal keyword changes. Older keywords are deprecated, although they are still compatible in this release. The removal of packed arrays means that some high-dimensional equations in earlier inputs will require changes to unpacked arrays. For example, *a(i,:)* may have to be replaced by correctly dimensioned, unpacked arrays like *a(i,:,:,..),* although the internal xSPDE code unpacks these if possible.

*xSPDE is distributed with no guarantee, under an open-source license. Contributions and bug reports are welcome.* An alternative approach to SPDEs [32,33] is available in C++ at http://www.xmds.org/.

## 1.2 Structure of the user's guide

Sections 2 and 3 give background information. Readers who are simply interested in how to use the code can go directly to section 4.

Section 2 has definitions and notations for stochastic differential equations (SDEs). This is useful for understanding later sections. The section includes Ito and Stratonovich calculus, probability distributions and Fokker-Planck equations. It also explains and defines the Fourier input-output spectra used in quantum technology.

Section 3 gives the concepts of stochastic partial differential equations (SPDEs). It includes details of spectral methods and the interaction picture approach. It has an explanation of how Fourier transforms and discrete sine or cosine transforms are implemented. It also explains how boundary conditions can be implemented using finite differences.

Section 4 describes the numerical solution of SDEs with xSPDE. This includes an explanation of the user interface, how to input parameters and equations, how to define the output in terms of functional averages or probabilities, and how to define and access auxiliary fields and noises. This section uses the default algorithms, because a more detailed explanation is given in section 7.

In section 5, the practical approach to solving stochastic partial differential equations with xSPDE is explained. The techniques used in section 5 are an extension of the previous one, so a thorough understanding of section 4 is strongly recommended. Both sections contain examples. Section 6 explains how to create projects with separated computation and graphics, as well as workflow and data storage.

Section 7 outlines the integration algorithms used in the manual. It includes a number of extended integration libraries, applicable to more specialized problems. Section 8 outlines how integration errors, including time-step and stochastic errors, can be estimated and displayed.

Section 9 provides a reference for the details of the internals as well as a comprehensive explanation of the input parameters useful in xSPDE simulations. Section 10 provides an extensive description of the visualization aspects of xSPDE, using the integrated xGRAPH function. This includes an automatic 'cascade' of graphic output, where high dimensional data is successively reduced to lower dimensional, visualizable data through projections.

Input parameters related to this are described as well. Data can also be graphed externally or stored for later analysis if preferred. Both average and raw trajectory data can be stored. However, the storage of raw data is generally not recommended, due to the large storage requirements. Additional examples in section 11 demonstrate how to obtain parametric plots against input parameters. Plots of one component value against another can be graphed. A function that analyses convergence rates is also available.

## 2 SDE theory

*This section describes the basics of stochastic differential equation (SDE) theory, in order to explain the background to the numerical methods.*

### 2.1 General form

A stochastic differential equation (SDE) is an equation with random noise terms. These were introduced by Langevin to treat small particles in fluids [4], and extended by Wiener, Ito and Stratonovich [34–36]. The theory and its applications to biology, chemistry, engineering, economics, physics, meteorology and other disciplines are treated in many texts [2, 3, 5, 37–39].

An ordinary stochastic differential equation in one time dimension is,

$$\frac{\partial \mathbf{a}}{\partial t} = \mathbf{A}(\mathbf{a}, t) + \underline{\mathbf{B}}(\mathbf{a}, t) \cdot \mathbf{w}(t). \tag{1}$$

Here $\mathbf{a}$ is a real or complex vector, $\mathbf{A}$ is a vector function, $\underline{\mathbf{B}}$ a matrix function and $\mathbf{w}$ is usually a delta-correlated real Gaussian noise vector such that:

$$\langle w_i(t) w_j(t') \rangle = \delta(t - t') \delta_{ij}. \tag{2}$$

One can also have non-Gaussian noise or noise that is not delta-correlated. Although these are somewhat less commonly treated, these alternatives are often found in real applications.

#### 2.1.1 Observables

In all cases, there are multiple independent trajectories, and one is interested in probabilistic averages, where the unweighted average of an observable $\mathbf{O}(\mathbf{a})$, for $N_s$ trajectories $\mathbf{a}^{(n)}$ is:

$$\langle \mathbf{O} \rangle_{N_s} = \frac{1}{N_s} \sum_n \mathbf{O}(\mathbf{a}^{(n)}). \tag{3}$$

In other types of stochastic equation [40, 41], there is a weight $\Omega(t)$ for each trajectory. This has an additional equation of motion, where:

$$\frac{\partial \Omega}{\partial t} = A_\Omega(\mathbf{a}, \Omega, t) + \underline{B}_\Omega(\mathbf{a}, \Omega, t) \cdot \mathbf{w}(t). \tag{4}$$

The results for all mean values are then weighted by the term $\exp(\Omega(t))$, so that:

$$\langle \mathbf{O} \rangle_\Omega = \frac{\sum_n \mathbf{O}(\mathbf{a}^{(n)}) \exp(\Omega^{(n)}(t))}{\sum_n \exp(\Omega^{(n)}(t))}. \tag{5}$$

This expression reduces to the usual average if the weights are zero, i.e, $\Omega = 0$. Apart from the way that averages are treated, the weight can simply be regarded as an additional term in the stochastic differential equations. This simply means that one now has an equation with an extra random field, so that $\mathbf{a} \rightarrow [\mathbf{a}, \Omega]$, together with a modified expression for the averages. This, in fact, is how these equations are solved.

For reasons of efficiency, it is best to use "breeding" algorithms to treat these numerically. This replicates highly weighted trajectories with $\Omega^{(n)}(t) \gg 0$ and removes trajectories with $\Omega^{(n)} \ll 0$, that have negligible weight. The numerical method is described in section 7. The remainder of this section will focus on the most commonly treated case of unweighted, Gaussian, delta-correlated noise.

## 2.2 Stochastic calculus

In the case of delta-correlated noise, the trajectories are not differentiable. As a result, there are two main variants of stochastic calculus used to define the derivatives, called Ito or Stratonovich [2, 39], and xSPDE can be used for either type. The default algorithms are designed for Stratonovich cases, since this is just ordinary calculus. Ito calculus can be treated also, either using the directly applicable Euler method, or else by appropriate transformations to a Stratonovich form. One can also have a time-reversed or implicit Ito calculus [27], which is directly solved using an implicit Ito-Euler method.

A single step in time of duration $\Delta t$ uses finite noises $\mathbf{w}$ which are defined to be delta-correlated in the small time-step limit, so that $\langle w_i w_j \rangle = \delta_{ij}/\Delta t$.

### 2.2.1 Types of stochastic calculus

The limits as $\Delta t \to 0$ are taken differently for the different types of stochastic calculus. Let $\mathbf{a}_0 = \mathbf{a}(t_0)$, $t_1 = t_0 + \Delta t$, $\mathbf{a}_1 = \mathbf{a}(t_1)$, $\bar{\mathbf{a}} = (\mathbf{a}_1 + \mathbf{a}_0)/2$, and $\bar{t} = t + \Delta t/2$, then the next step in time is:

- Ito calculus - uses **initial-time** derivative evaluations

$$\mathbf{a}_1 = \mathbf{a}_0 + \left[ \mathbf{A}^{(I)}(\mathbf{a}_0, t_0) + \underline{\mathbf{B}}(\mathbf{a}_0, t_0) \cdot \mathbf{w} \right] \Delta t . \tag{6}$$

- Stratonovich calculus - uses **midpoint** derivative evaluations

$$\mathbf{a}_1 = \mathbf{a}_0 + \left[ \mathbf{A}(\bar{\mathbf{a}}, \bar{t}) + \underline{\mathbf{B}}(\bar{\mathbf{a}}, \bar{t}) \cdot \mathbf{w} \right] \Delta t . \tag{7}$$

- Implicit Ito calculus - uses **final-time** derivative evaluations

$$\mathbf{a}_1 = \mathbf{a}_0 + \left[ \mathbf{A}^{(I+)}(\mathbf{a}_1, t_1) + \underline{\mathbf{B}}(\mathbf{a}_1, t_1) \cdot \mathbf{w} \right] \Delta t . \tag{8}$$

The drift term $\mathbf{A}$ is changed in Ito or implicit Ito calculus, if the noise coefficient $B$ depends on the stochastic variable:

$$A_i^{(I)} = A_i + \frac{1}{2} B_{jk} \partial_j B_{ik}, \tag{9}$$

$$A_i^{(I+)} = A_i - \frac{1}{2} B_{jk} \partial_j B_{ik}. \tag{10}$$

Here we define $\partial_n \equiv \partial/\partial a_n$ and we use an Einstein convention of summing over all repeated indices.

## 2.3 Interaction picture

The interaction picture allows one to eliminate linear terms in the time derivatives. It is especially useful for stochastic partial differential equations, but it is applicable to stochastic equations as well. Suppose there are linear terms $\underline{\mathbf{L}}$, so that $\mathbf{A}(\mathbf{a}, t) = \mathbf{A}_1(\mathbf{a}, t) + \underline{\mathbf{L}} \cdot \mathbf{a}$, where $\underline{\mathbf{L}}$ is a constant matrix. The interaction picture defines local variables $\tilde{\mathbf{a}}$ for the fields $\mathbf{a}$.

It is convenient to introduce an abbreviated notation as:

$$D(\mathbf{a}) = \mathbf{A}_1(\mathbf{a}, t) + \underline{\mathbf{B}}(\mathbf{a}, t) \cdot \mathbf{w}(t), \tag{11}$$

so that one can write the differential equation as:

$$\frac{\partial \mathbf{a}}{\partial t} = D(\mathbf{a}) + \underline{\mathbf{L}} \cdot \mathbf{a}. \tag{12}$$

### 2.3.1 Linear propagator

Next, we define a linear propagator. This is given formally by:

$$\underline{\mathbf{P}}(\Delta t) = \exp\left(\Delta t \underline{\mathbf{L}}\right). \tag{13}$$

where $\Delta t = t - \bar{t}$, and $\bar{t}$ is the interaction picture origin. Transforming the field $\mathbf{a}$ to an interaction picture is achieved on defining:

$$\tilde{\mathbf{a}} = \underline{\mathbf{P}}^{-1}(\Delta t)\,\mathbf{a}. \tag{14}$$

As a result, the equation of motion is:

$$\frac{\partial \tilde{\mathbf{a}}}{\partial t} = D\left(\underline{\mathbf{P}}(\Delta t)\tilde{\mathbf{a}}\right). \tag{15}$$

This removes linear terms, which can cause stiffness in the equations, increasing the discretization error. Given the case of a completely linear ODE or SDE, the trajectory solutions will be exact up to round-off errors.

## 2.4 Probability distributions

Stochastic equations generate trajectories distributed with a probability density $P(\mathbf{a})$. These can be defined as an average and hence can be evaluated stochastically, since:

$$P\left(\mathbf{a}'\right) = \left\langle \delta\left(\mathbf{a}' - \mathbf{a}\right)\right\rangle. \tag{16}$$

Here $\langle .. \rangle \equiv \langle .. \rangle_\infty$ is the infinite ensemble limit of the average over many trajectories. The probability can be shown to follow a Fokker-Planck equation (FPE) with positive semi-definite diffusion matrix, [2, 42]:

$$\frac{\partial P}{\partial t} = \mathcal{L}P = \left[-\partial_n A_n^{(I)} + \frac{1}{2}\partial_n \partial_m B_{nk}B_{mk}\right]P, \tag{17}$$

where the differential operators act on all terms to their right.

### 2.4.1 Distribution averages

The average of any observable $\mathbf{O}(\mathbf{a})$ is obtained either by averaging over the stochastic trajectories numerically, or by analytic calculations, using:

$$\langle \mathbf{O} \rangle \;\; = \;\; \int \mathbf{O}(\mathbf{a})P(\mathbf{a})\,d\mathbf{a}. \tag{18}$$

The dynamics of an observable or moment follows an adjoint equation, where $\tilde{\mathcal{L}}$ is the adjoint of $\mathcal{L}$:

$$\left\langle \frac{\partial \mathbf{O}}{\partial t}\right\rangle = \left\langle \tilde{\mathcal{L}}\mathbf{O}\right\rangle, \tag{19}$$

where:

$$\left\langle \tilde{\mathcal{L}}\mathbf{O}\right\rangle = \left\langle \left[A_n^{(I)}\partial_n + \frac{1}{2}B_{nk}B_{mk}\partial_n \partial_m\right]\mathbf{O}\right\rangle. \tag{20}$$

This equation allows the time-evolution of averages to be calculated analytically in simple cases, given an initial distribution. However, in more complex cases, a numerical simulation of the stochastic equations is more practical, and this can be carried out with xSPDE or other software.

## 2.5 Example: random walk

The first example of an SDE is the simplest possible stochastic equation or Wiener process:

$$\dot{a} = w(t). \tag{21}$$

This has the solution that

$$a(t) = a(0) + \int_0^t w(\tau) d\tau, \tag{22}$$

which means that the initial mean value does not change in time:

$$\langle a(t) \rangle = \langle a(0) \rangle. \tag{23}$$

### 2.5.1 Variance solution

The noise correlation is non-vanishing from Eq (2), so the variance must increase with time:

$$\begin{aligned}
\langle a^2(t) \rangle &= \langle a^2(0) \rangle + \int_0^t \int_0^t \langle w(\tau) w(\tau') \rangle d\tau d\tau' \\
&= \langle a^2(0) \rangle + \int_0^t \int_0^t \delta(\tau - \tau') d\tau d\tau'.
\end{aligned} \tag{24}$$

Integrating the delta function gives unity, which means that the second moment and the variance both increase linearly with time:

$$\begin{aligned}
\langle a^2(t) \rangle &= \langle a^2(0) \rangle + \int_0^t d\tau \\
&= \langle a^2(0) \rangle + t.
\end{aligned} \tag{25}$$

The probability follows an elementary diffusion equation:

$$\frac{\partial P}{\partial t} = \frac{1}{2} \frac{\partial^2 P}{\partial a^2}, \tag{26}$$

which is an example of Eq (17). From this equation and using Eq (19), the first two corresponding moment equations in this case are

$$\begin{aligned}
\frac{\partial}{\partial t} \langle a \rangle &= \left\langle \frac{1}{2} \frac{\partial^2}{\partial a^2} a \right\rangle = 0 \\
\frac{\partial}{\partial t} \langle a^2 \rangle &= \left\langle \frac{1}{2} \frac{\partial^2}{\partial a^2} a^2 \right\rangle = 1.
\end{aligned} \tag{27}$$

These differential equations are satisfied by the solutions obtained directly from the stochastic equations, namely Eq (23) and Eq (25).

## 2.6 Probability densities

The Wiener process with an arbitrary noise strength has the stochastic equation:

$$\dot{a} = bw(t). \tag{28}$$

The probability density satisfies the Fokker-Planck equation for diffusion,

$$\frac{\partial P}{\partial t} = \frac{b^2}{2} \frac{\partial^2}{\partial a^2} P. \tag{29}$$

Then, if $x$ initially is Gaussian distributed, this has a Gaussian distribution at time $t$ with:

$$P(a) = \frac{1}{\sqrt{2\pi\sigma^2(t)}} \exp\left[-\frac{(a - \bar{a}(t))^2}{2\sigma^2(t)}\right]. \tag{30}$$

Here:

$$\bar{a}(t) = \bar{a}(0) \tag{31}$$
$$\sigma^2(t) = \sigma^2(0) + b^2 t.$$

### 2.6.1 Distributions of functions

Any function of the stochastic variables has a corresponding probability density. For example, the distribution of $a^2$ has a $\chi^2$ distribution with a single degree of freedom, such that if $y = (a - \bar{a}(t))^2/\sigma^2(t)$, then:

$$P(y) = \frac{1}{\sqrt{2\pi y}} \exp\left[-\frac{y}{2}\right]. \tag{32}$$

Hence:

$$P\left(a^2\right) = \frac{1}{|a - \bar{a}(t)| \sqrt{2\pi\sigma^2(t)}} \exp\left[-\frac{(a - \bar{a}(t))^2}{2\sigma^2(t)}\right]. \tag{33}$$

More generally, it is often not known what the exact analytic solutions are, and a numerical solution is employed. This can either use the stochastic equation directly, or the Fokker-Planck equation, although it is generally difficult to scale this to many variables or to partial differential equations,

That is why we focus on the stochastic equation approach here, which can be used to numerically calculate either the mean values or the probability distributions in general cases.

## 2.7 Fourier transforms

Frequency spectra have many uses, especially for understanding the steady-state fluctuations of any physical system in the presence of noise, typically either thermal or quantum-mechanical, although the noise could have other sources.

The time-domain spectral definition used here is:

$$\tilde{a}(\omega) = \frac{1}{\sqrt{2\pi}} \int e^{i\omega t} a(t) dt$$

$$a(t) = \frac{1}{\sqrt{2\pi}} \int e^{-i\omega t} \tilde{a}(\omega) d\omega. \tag{34}$$

As a simple example, a sinusoidal oscillation in the form

$$a(t) = \cos(\omega_0 t). \tag{35}$$

between $t = -T/2$ and $t = T/2$ has a Fourier transform given by:

$$\tilde{a}(\omega) = \frac{1}{2\sqrt{2\pi}} \int_{-T/2}^{T/2} \left[ e^{i(\omega - \omega_0)t} + e^{i(\omega + \omega_0)t} \right] dt \tag{36}$$

$$= \frac{T}{2\sqrt{2\pi}} \left[ sinc\left( (\omega - \omega_0)\frac{T}{2} \right) + sinc\left( (\omega + \omega_0)\frac{T}{2} \right) \right].$$

### 2.7.1 Discrete Fourier transforms

While exact in this analytic case, the definition above is impractical for numerical calculations. In taking measurements and doing simulations, one has a discrete set of data-points. Assuming the samples are at fixed intervals, the best one can do in practical cases is a discrete Fourier transform, with samples $\bar{a}(t_j)$ that are defined as integrals over each small interval $dt$:

Let $\bar{a}(t_j)$ be the average over a small time interval:

$$\bar{a}(t_j) = \int_{t_j - dt/2}^{t_j + dt/2} e^{i\omega t} a(t) dt, \tag{37}$$

then to a good approximation as $dt \to 0$, provided $\omega_n$ is not too large,

$$\tilde{a}(\omega_n) = \frac{\Delta t}{\sqrt{2\pi}} \sum_{j=1}^{N} e^{i\omega_n t_j} \bar{a}(t_j)$$

$$\bar{a}(t_j) = \frac{\Delta \omega}{\sqrt{2\pi}} \sum_{n=1}^{N} e^{-i\omega_n t_j} \tilde{a}(\omega_n). \tag{38}$$

These also form an invertible pair provided that $\Delta t \Delta \omega = 2\pi/N$. As well as being more practical, this is very efficient due to the fast Cooley-Tukey (FFT) algorithm [43], allowing computation on time-scales of $O(N \ln N)$ rather than $O(N^2)$ as one might expect.

When taking Fourier transforms in the time-domain, xSPDE does a time-averaging of all fields over the current time-step, using the available coarse and fine time-samples. This is done by averaging the field before and after the stochastic time-step. The methods used for this are described in greater detail in Subsection (4.7).

## 2.8 Quantum phase-space

One useful application of stochastic equations is in quantum technologies, where stochastic methods are generally much more scalable than other methods [44, 45]. This approach started when Schrodinger [46] pointed out that quantum oscillators can have classical equations. Wigner, Moyal and Glauber extended this to other systems [47–49]. In lasers and quantum optics [3, 50–52], it is used to obtain SDEs for quantum systems coupled to reservoirs.

For the case of bosons, any $M$-mode quantum density matrix $\hat{\rho}$ may be written in a unified quantum phase-space form as:

$$\hat{\rho} = \int d^{2M}\alpha \, d^{2M}\beta \, P_\sigma(\boldsymbol{\alpha}, \boldsymbol{\beta}) \hat{\Lambda}_\sigma(\boldsymbol{\alpha}, \boldsymbol{\beta}), \tag{39}$$

where $P_\sigma$ is the $\sigma$-ordered phase-space distribution function, and $\boldsymbol{\alpha} = [\alpha_1, \ldots \alpha_M]$. The basis, $\hat{\Lambda}_\sigma$, is a Gaussian function of annihilation and creation operators ([53]), whose variance depends on $\sigma$. This is defined as the variance of $\alpha\beta$ due to vacuum fluctuations, in the operator ordering of the representation.

For clarity, we hats like $\hat{a}$ are used here to indicate operators that do not commute with each other, as opposed to stochastic variables like $\alpha$ that do commute. For any given operator ordering, it is always possible to find a probability distribution such that the expectation of an operator product equals the stochastic variable correlations [54].

### 2.8.1 Positive-P representation

The above expansion leads to different statistics and noise terms depending on the operator ordering. For example, in the normally ordered positive P-representation where $\sigma = 0$, the operator basis is

$$\hat{\Lambda}(\boldsymbol{\alpha}, \boldsymbol{\beta}) \equiv \frac{|\boldsymbol{\alpha}\rangle \langle \boldsymbol{\beta}^*|}{\langle \boldsymbol{\beta}^*| \boldsymbol{\alpha}\rangle}. \tag{40}$$

Here $\boldsymbol{\beta} = \boldsymbol{\alpha}^\dagger \sim \hat{a}^\dagger$ is a stochastic variable conjugate in the mean to $\boldsymbol{\alpha}^*$. Any quantum state has a positive representation of this type [55], and normally ordered coherence functions are moments of the distribution with:

$$\left\langle \hat{a}^\dagger_{m_1} \ldots \hat{a}_{m_n} \right\rangle = \int d^{2M}\boldsymbol{\alpha} d^{2M}\boldsymbol{\beta} \left[ \beta_{m_1} \ldots \alpha_{m_n} \right] P(\boldsymbol{\alpha}, \boldsymbol{\beta}). \tag{41}$$

The Glauber-Sudarshan representation used in laser physics is obtained for $\boldsymbol{\beta} = \boldsymbol{\alpha}^*$, so that the two variables are exactly conjugate:

$$\hat{\rho} = \int d^{2M}\boldsymbol{\alpha} P(\boldsymbol{\alpha}) |\boldsymbol{\alpha}\rangle \langle \boldsymbol{\alpha}|. \tag{42}$$

In this case, the total number of stochastic variables is halved, but nonclassical squeezed or entangled states cannot be represented as a positive distribution. Two additional frequently utilized representations are the Wigner ($\sigma = 1/2$) and Husimi ($\sigma = 1$) representations, characterized by symmetric ordering and anti-normal ordering, respectively. These also have a classical phase-space with $\boldsymbol{\beta} = \boldsymbol{\alpha}^*$.

### 2.8.2 Master equations

The dynamics of a quantum system coupled to a reservoir is described by a master equation. In the Markovian (high-frequency) limit, the general quantum master equation for a dissipative quantum system with damping rates $\Gamma_j$ is

$$\frac{\partial \hat{\rho}}{\partial t} = \frac{1}{i\hbar} \left[ \hat{H}_{sys}, \hat{\rho} \right] + \sum_j \Gamma_j \left( \bar{n}_j + 1 \right) (2\hat{A}_j \hat{\rho} \hat{A}_j^\dagger - \hat{A}_j^\dagger \hat{A}_j \hat{\rho} - \hat{\rho} \hat{A}_j^\dagger \hat{A}_j)$$
$$+ \sum_j \Gamma_j \bar{n}_j (2\hat{A}_j^\dagger \hat{\rho} \hat{A}_j - \hat{A}_j \hat{A}_j^\dagger \hat{\rho} - \hat{\rho} \hat{A}_j \hat{A}_j^\dagger), \tag{43}$$

where $\Gamma_j$ is a damping rate for reservoir couplings to the operator $\hat{A}_j$, $\bar{n}_j$ is the finite temperature reservoir occupation, and the typical damping operators are:

The operator equations are mapped to differential equations with the equivalences:

| Damping operator $(\hat{A}_j)$ | $\Gamma_j$ | Physical interpretation |
|:---:|:---:|:---:|
| $\hat{a}_j$ | $\gamma_j$ | Linear amplitude loss (units $s^{-1}$) |
| $\hat{a}_j^\dagger$ | $g_j$ | Linear amplitude gain (units $s^{-1}$) |
| $\hat{a}_j^\dagger \hat{a}_j$ | $\gamma_j^p$ | Phase decay rate gain (units $s^{-1}$) |
| $\hat{a}_j^2$ | $\kappa_j/2$ | nonlinear amplitude loss (units $s^{-1}$). |

Table 1: Typical types of quantum damping term

$$
\begin{aligned}
\hat{a}_n^\dagger \hat{\rho} &\rightarrow \left[\beta_n + (\sigma - 1)\frac{\partial}{\partial \alpha_n}\right] P_\sigma \\
\hat{a}_n \hat{\rho} &\rightarrow \left[\alpha_n + \sigma \frac{\partial}{\partial \beta_n}\right] P_\sigma \\
\hat{\rho} \hat{a}_n &\rightarrow \left[\alpha_n + (\sigma - 1)\frac{\partial}{\partial \beta_n}\right] P_\sigma \\
\hat{\rho} \hat{a}_n^\dagger &\rightarrow \left[\beta_n + \sigma \frac{\partial}{\partial \alpha_n}\right] P_\sigma .
\end{aligned}
\tag{44}
$$

The operator mappings give a differential equation. If it has a second-order positive-definite form it is a Fokker-Planck equation equivalent to an SDE, or an SPDE for quantum fields [56]. The noise can be additive or multiplicative, depending on the problem. Not all cases give stable FPE equations [57], and truncation is required for the Wigner representation if the Hamiltonian is nonlinear [58].

The total noise includes internal quantum noise generated from the Hamiltonian term $\hat{H}_{sys}$, as well as reservoir noise terms generated from the coupling to the reservoir operators, which is proportional to $\Gamma_j$. There is a similar behavior in classical systems, except that these correspond to a high-temperature limit, and in most cases only have external reservoir noise from thermal fluctuations.

## 2.9 Damped harmonic oscillator

As an example, take the damped quantum harmonic oscillator. This has the Hamiltonian $H = \omega_0 \hat{a}^\dagger \hat{a}$. If damping is added, it obeys the master equation

$$
\begin{aligned}
\frac{d\hat{\rho}}{dt} = \frac{-i}{\hbar}[\omega_0 \hat{a}^\dagger \hat{a}, \rho] &+ \gamma(1 + \bar{n})(2\hat{a}\rho\hat{a}^\dagger - \hat{a}^\dagger \hat{a}\rho - \rho\hat{a}^\dagger \hat{a}) \\
&+ \gamma\bar{n}(2\hat{a}^\dagger \rho \hat{a} - \hat{a}\hat{a}^\dagger \rho - \rho\hat{a}\hat{a}^\dagger).
\end{aligned}
\tag{45}
$$

This leads to a random walk in a complex space [2, 3],

$$
\begin{aligned}
\frac{d\alpha}{dt} &= -(\gamma + i\omega_0)\alpha + \sqrt{2\gamma(\sigma + \bar{n})}\zeta(t) \\
\frac{d\beta}{dt} &= -(\gamma - i\omega_0)\beta + \sqrt{2\gamma(\sigma + \bar{n})}\zeta^*(t),
\end{aligned}
\tag{46}
$$

where the noise is complex and $\zeta(t) = (w_1(t) + iw_2(t))/\sqrt{2}$. The correlations are

$$
\begin{aligned}
\langle \zeta(t)(\zeta(t'))^* \rangle &= \delta(t - t') \\
\langle \zeta(\omega)(\zeta(\omega'))^* \rangle &= \delta(\omega - \omega').
\end{aligned}
\tag{47}
$$

### 2.9.1 Wigner representation

In the zero temperature Wigner case with $\gamma = 1$, $\sigma = 1/2$, and in a rotating frame so that $\omega_0 = 0$, the probability follows the Fokker-Planck equation:

$$\frac{\partial P}{\partial t} = \left[ \frac{\partial}{\partial \alpha_x} \alpha_x + \frac{\partial}{\partial \alpha_y} \alpha_y + \frac{1}{4} \left( \frac{\partial^2}{\partial \alpha_x^2} + \frac{\partial^2}{\partial \alpha_y^2} \right) \right] P, \tag{48}$$

which is an example of Eq (17). Ignoring terms that vanish or can be obtained from symmetry, the first corresponding moment equations in each of the real and imaginary directions are

$$\frac{\partial}{\partial t} \langle \alpha_x \rangle = \left\langle -\alpha_x \frac{\partial}{\partial \alpha_x} \alpha_x \right\rangle = -\langle \alpha_x \rangle$$

$$\frac{\partial}{\partial t} \langle \alpha_x \alpha_y \rangle = \left\langle -\left( \alpha_x \frac{\partial}{\partial \alpha_x} + \alpha_y \frac{\partial}{\partial \alpha_y} \right) \alpha_x \alpha_y \right\rangle = -\langle \alpha_x \alpha_y \rangle$$

$$\frac{\partial}{\partial t} \langle \alpha_x^2 \rangle = \left\langle \left( -\alpha_x \frac{\partial}{\partial \alpha_x} + \frac{1}{4} \frac{\partial^2}{\partial \alpha_x^2} \right) \alpha_x^2 \right\rangle = \frac{1}{2} - 2 \langle \alpha_x^2 \rangle. \tag{49}$$

The steady-state is therefore a Gaussian distribution with $\langle \alpha_{x,y} \rangle = 0$, $\langle \alpha_x \alpha_y \rangle = 0$ and $\langle \alpha_{x,y}^2 \rangle = 1/4$. One can use an initial condition of $\alpha = (\nu_1 + i\nu_2)/2$, with $\langle \nu_i^2 \rangle = 1/2$, in order to replicate the steady state, which is a Gaussian with $\langle \alpha_x \rangle = \langle \alpha_y \rangle = 0$ and $\langle \alpha_x^2 \rangle = \langle \alpha_y^2 \rangle = 1/4$.

### 2.9.2 Internal spectrum

Neglecting any boundary terms, the equation in frequency space is:

$$-i\omega \tilde{\alpha}(\omega) = -\tilde{\alpha}(\omega) + \tilde{\zeta}(\omega). \tag{50}$$

For sufficiently long times, the solution in frequency space - where $\omega = 2\pi f$ is the angular frequency - is therefore given by:

$$\tilde{\alpha}(\omega) = \frac{\tilde{\zeta}(\omega)}{1 - i\omega}. \tag{51}$$

The expectation value of the noise spectrum, $\langle |\tilde{\alpha}(\omega)|^2 \rangle$ in the long time limit, is:

$$\begin{aligned} \langle |\tilde{\alpha}(\omega)|^2 \rangle &= \frac{1}{2\pi(1 + \omega^2)} \int \int e^{-i\omega(t - t')} \langle \zeta(t) \zeta^*(t') \rangle \, dt \, dt'. \\ &= \frac{T}{2\pi(1 + \omega^2)}. \end{aligned} \tag{52}$$

This equation can also be used for some classical problems, which correspond to the high-temperature limit of $\bar{n} \gg 1$.

## 2.10 Input-output spectra

The spectrum of an internal field variable is not the one that is usually measured. An important application of stochastic equations is therefore in calculating output, measured spectra of lasers, quantum optics, opto-mechanics and quantum circuits [51, 59]. These have the feature that the measured output spectrum may also include noise from reflected fields at the input/output

ports. If the quantum noise term in the Heisenberg equations for a cavity operator $\hat{a}_c$ is given by:
$\dot{\hat{a}}_c \sim .. + \sqrt{2\gamma}\hat{a}_{in}(t)$, then the corresponding operator input-output relations are $\hat{a}_{out}(t) + \hat{a}_{in}(t) = \sqrt{2\gamma}\hat{a}_c$.

In quantum phase-space for the case of the harmonic oscillator or similar systems, $\alpha_{in} = \sqrt{\sigma + \bar{n}}\zeta$ is the noise term in the Langevin equation. The output fields $\alpha_{out}$ that are measured are given by:

$$\alpha_{out} = \sqrt{2\gamma}\alpha - \alpha_{in}. \tag{53}$$

Hence one must include in the spectrum both the internal mode variables and the noise terms themselves. Solving for the spectra, one obtains auxiliary fields with

$$\tilde{\alpha}_{in}(\omega) = \sqrt{\sigma + \bar{n}}\tilde{\zeta}(\omega) \tag{54}$$
$$\tilde{\alpha}_{out}(\omega) = \sqrt{2\gamma}\tilde{a}(\omega) - \sqrt{\sigma + \bar{n}}\tilde{\zeta}(\omega).$$

In summary, it is the output fields that are amplified and measured. Hence one must be able to compute the spectra of the output fields for experimental comparisons. These have the additional feature that they include the reservoir noise $\tilde{\zeta}(\omega)$, evaluated at the same time as the field is evaluated, since the reservoir noise is the input here. In xSPDE these are called *auxfields*.

### 2.10.1 Steady-state result

Consider the example of Subsection (2.9), in the Wigner representation case with $\gamma = 1$, $\sigma = 1/2$ and $\bar{n} = 0$. Over long time-scales, so that one is in the steady state, the solution for $\tilde{a}_{out}$ is that:

$$\tilde{\alpha}_{out}(\omega) = \sqrt{2}\left[\frac{1}{1 - i\omega} - \frac{1}{2}\right]\tilde{\zeta}(\omega)$$
$$= \frac{1}{\sqrt{2}}\left[\frac{1 + i\omega}{1 - i\omega}\right]\tilde{\zeta}(\omega). \tag{55}$$

This gives the following expectation values:

$$\left\langle\tilde{\alpha}_{out}(\omega)\left(\tilde{\alpha}_{out}(\omega)(\omega')\right)^*\right\rangle = \frac{1}{2}\delta\left(\omega - \omega'\right)$$
$$\left\langle\tilde{\alpha}_{in}(\omega)\left(\tilde{\alpha}_{in}(\omega)(\omega')\right)^*\right\rangle = \frac{1}{2}\delta\left(\omega - \omega'\right). \tag{56}$$

These are the expectation values of the zero temperature quantum fluctuations in the input and output channels. This means that the harmonic oscillator in its ground state is in equilibrium with an external vacuum field reservoir, also in its ground state. However, from Eq (52), the internal spectral correlations of the harmonic oscillator are modified by the coupling.

While this is a simple result, exactly the same general type of behavior occurs in more sophisticated cases. These may include many coupled modes with nonlinearities. Additional or auxiliary fields that depend both on noise terms and internal stochastic variables are required. The soluble case given above is a useful test case, and it is treated numerically later in the manual.

# 3 SPDE theory

*This section describes the basics of stochastic partial differential equation (SPDE) theory, in order to explain the background to the numerical methods.*

## 3.1 SPDE definitions

A stochastic partial differential equation or SPDE is defined in both time $t$ and one or more space dimensions $\mathbf{x}$. We suppose there are $d$ total space-time dimensions. The space-time coordinate is denoted as $\mathbf{r} = \left(r^1, \ldots r^d\right) = (t, \mathbf{x}) = (t, x, y, z, \ldots)$.

The stochastic partial differential equation solved is written in differential form as

$$\frac{\partial \mathbf{a}}{\partial t} = \mathbf{A}[\nabla, \mathbf{a}, \mathbf{r}] + \underline{\mathbf{B}}[\nabla, \mathbf{a}, \mathbf{r}] \cdot \mathbf{w}(\mathbf{r}) + \mathbf{L}[\nabla, \mathbf{a}, \mathbf{r}].\mathbf{a}. \tag{57}$$

Here, $\mathbf{a} = \left[a_1, \ldots a_f\right]$ is a real or complex vector field, $\mathbf{A}$ is a vector function of fields and space and $\underline{\mathbf{B}}$ a matrix function. The new feature is that terms can now include the operator $\nabla$, which is a differential term in a real space $\mathbf{x}$. The exact structure of these terms is important, and not all such equations have well-behaved solutions [60, 61].

In many common cases, the noise term $\mathbf{w}$ is delta-correlated in time and space:

$$\left\langle w_i(\mathbf{r}) w_j\left(\mathbf{r}'\right)\right\rangle = \delta\left(t - t'\right)\delta\left(\mathbf{x} - \mathbf{x}'\right)\delta_{ij}. \tag{58}$$

One can also have noise with a finite correlation length defined by a noise correlation function $N_{ij}\left(\mathbf{x} - \mathbf{x}'\right)$ in space so that:

$$\left\langle w_i(\mathbf{r}) w_j\left(\mathbf{r}'\right)\right\rangle = \delta\left(t - t'\right)N_{ij}\left(\mathbf{x} - \mathbf{x}'\right). \tag{59}$$

It is even possible to have noise with a finite correlation time. Currently, these are not directly treated in xSPDE, although user definitions of this are possible by adding a customized noise function.

Additionally, the initial field has a probability distribution. In most examples, we suppose that this initial random field distribution can be generated as a function of Gaussian distributed initial random fields $\mathbf{v}(\mathbf{x})$, where:

$$\left\langle v_i(\mathbf{x}) v_j\left(\mathbf{x}'\right)\right\rangle = \delta\left(\mathbf{x} - \mathbf{x}'\right)\delta_{ij}. \tag{60}$$

However, it is also possible that the initial random fields are also not delta-correlated, so that

$$\left\langle v_i(\mathbf{x}) v_j\left(\mathbf{x}'\right)\right\rangle = R_{ij}\left(\mathbf{x} - \mathbf{x}'\right). \tag{61}$$

Both finite correlation length and delta-correlated noise and random terms can be used in xSPDE simulations, with finite correlation lengths defined through a Fourier transform method.

## 3.2 Boundary conditions

There are three types of boundaries that are available in xSPDE. They are specified independently for each space dimension $j = 2, \ldots d$, field component $i = 1, \ldots f$, and lower or upper location $\ell = 1, 2$. Each has an xSPDE boundary type. These are specified with a numerical code $bt$, as:

**Dirichlet** (specified value, $bt = 1$): $a_i\left(r^1, r^2, \ldots \hat{r}_\ell^j, \ldots\right) = f_{ij\ell}(\mathbf{r}, \mathbf{a})$ .

**Periodic** $(bt = 0)$: $a_i\left(r^1, r^2, \dots \hat{r}^j_\ell, \dots\right) = a_i\left(r^1, r^2, \dots \hat{r}^j_{3-\ell}, \dots\right)$.

**Robin/Neumann** (specified derivative, $bt = -1$): $\frac{\partial}{\partial r^j} a_i\left(r^1, r^2, \dots \hat{r}^j_\ell, \dots\right) = g_{ij\ell}(\mathbf{r}, \mathbf{a})$.

The coordinates $\hat{r}^j_\ell = \left(r^j_1, r^j_2\right)$ are locations where boundary conditions are enforced. There are five types of boundary *combinations* of these for each dimension and field variable. Note that the boundary type can change the error stability properties of an equation. The most general boundaries can only be specified using finite differences currently, as the spectral method boundary types are more limited.

Periodic boundaries can't be combined with other types, as this defines both boundaries:

**a)** periodic-periodic- P-P: "0,0"

**b)** Dirichlet-Dirichlet- D-D: "1,1"

**c)** Robin-Robin- R-R: "-1,-1"

**d)** Robin-Dirichlet- R-D: "-1,1"

**e)** Dirichlet-Robin- D-R: "1,-1"

Just as with the derivative term, each of these types can change with dimension and field component. Specified field or derivative values can be any user-defined functions of space, time, and field amplitude or simply have fixed values. Currently, all combinations of boundaries can be treated in xSPDE using finite difference derivatives. Spectral methods are restricted to periodic or zero Dirichlet/Neumann boundary conditions.

### 3.3 Spatial grid and boundaries

The precise location of the boundary at $\hat{r}^j_\ell$ is important in solving (S)PDEs, especially if high accuracy is required, or if field values at the boundary are needed.

Suppose the spatial grid spacing is $\Delta x$ and the number of grid points in a particular dimension $d$ is $points(d) = N$, then the maximum range from the first to last computed point is **always**:

$$R = (N - 1)\Delta x = ranges(d). \tag{62}$$

Noting that $\mathbf{r} = (t, \mathbf{x})$, and $\Delta \mathbf{r} = (\Delta t, \Delta \mathbf{x})$, this means that the space-time points are at:

$$r_i = O_i + (i - 1)\Delta r_i. \tag{63}$$

There are two distinct spatial boundary locations used in xSPDE, depending on the type of boundary conditions specified, as follows:

#### 3.3.1 Periodic boundary

Due to periodicity, the logical boundary location is arbitrary. For the default case of a periodic boundary, the indices are arranged as though on a circle from $1 : N$. It is useful to suppose the boundary as at both $\hat{r}^j_1 = r^j_1 - \Delta r^j/2$ and at $\hat{r}^j_2 = r^j_{N_j} + \Delta r^j/2$. Neither upper or lower logical 'boundary' is at a grid point. The effective range of the domain is $R^j + \Delta r^j$. Only the values at $N$ points are computed, and one must regard the point where the periodicity is enforced as interpolating between the last and first point.

### 3.3.2 Dirichlet/Robin boundary

For the case of a non-periodic boundary, including Dirichlet, Robin and Neumann boundary conditions, the indices are simply in a line from $1 : N$. The lower and upper lower boundaries are at $\hat{r}_1^j = r_1^j$ and at $\hat{r}_2^j = r_{N_j}^j$. In some PDE methods the logical boundaries are outside the grid boundaries, but that is not the case here. Unlike the periodic case, boundaries are enforced at the first and last point. This is different to what is found in standard trigonometric transform software, but this approach allows for a unified and simpler treatment of multiple types of algorithm.

## 3.4 Multidimensional walk

The simplest example of an SPDE is the multidimensional Wiener process:

$$\dot{a} = w(t, \mathbf{x}). \tag{64}$$

This has a solution that is identical in appearance to an SDE:

$$a(t, \mathbf{x}) = a(0, \mathbf{x}) + \int_0^t w(\tau, \mathbf{x}) \, d\tau. \tag{65}$$

Just as for an SDE, this means that the initial mean value does not change in time:

$$\langle a(t, \mathbf{x}) \rangle = \langle a(0, \mathbf{x}) \rangle. \tag{66}$$

Since there are no spatial derivatives here, boundary values are not important. One can regard this as having periodic boundaries, which by the xSPDE conventions means that no boundary conditions are enforced - since periodic boundaries do not alter computed values when there are no derivatives.

### 3.4.1 Variance solution

The noise correlation is non-vanishing from Eq (2), so the variance must increase with time:

$$
\begin{aligned}
\langle a^2(t, \mathbf{x}) \rangle &= \langle a^2(0, \mathbf{x}) \rangle + \int_0^t \int_0^t \langle w(\tau, \mathbf{x}) w(\tau', \mathbf{x}) \rangle \, d\tau \, d\tau' \\
&= \langle a^2(0, \mathbf{x}) \rangle + \delta^{d-1}(0) \int_0^t \int_0^t \delta(\tau - \tau') \, d\tau \, d\tau'.
\end{aligned} \tag{67}
$$

Integrating the temporal delta function gives unity. The spatial delta-function is replaced by $1/\Delta V$ in a discretized lattice calculation at points $\mathbf{x}_j$ with cell volume $\Delta V = \prod \Delta x_j$, which means that the second moment and the variance both increase linearly with time:

$$\langle a^2(t, \mathbf{x}_j) \rangle = \langle a^2(0, \mathbf{x}_j) \rangle + t/\Delta V. \tag{68}$$

The probability on the lattice for observing lattice field values $a_j$ follows an elementary diffusion equation:

$$\frac{\partial P}{\partial t} = \frac{1}{2\Delta V} \sum_j \frac{\partial^2 P}{\partial a_j^2}, \tag{69}$$

which is an example of Eq (17). From this equation and using Eq (19), the first two corresponding moment equations in this case are

$$
\frac{\partial}{\partial t}\left\langle a_j \right\rangle = \left\langle \frac{1}{2}\frac{\partial^2}{\partial a_j^2}a_j \right\rangle = 0
$$

$$
\frac{\partial}{\partial t}\left\langle a_j^2 \right\rangle = \left\langle \frac{1}{2\Delta V}\frac{\partial^2}{\partial a_j^2}a_j^2 \right\rangle = \frac{1}{\Delta V}. \tag{70}
$$

These differential equations are satisfied by the solutions obtained directly from the stochastic equations, but as one can see, the coupling between the lattice points provides more interesting behavior. This requires derivative terms such as Laplacians.

## 3.5 Interaction picture

To treat Laplacians, spectral or interaction-picture methods can be very efficient, and in certain cases give both much lower errors and much faster run-times. They do not have the large errors and stability problems of finite difference methods, which allows much larger time-steps to be used.

To explain the algorithm, (S)PDEs often contain terms which are linear in the field variables **a**, including derivative operators acting on **a**. This can be treated using an *interaction picture*, which leads to dramatically reduced time-step errors and higher stability [29, 62]. In the literature on partial differential equations, this is called a spectral method.

The interaction picture provides a means to solve for linear terms in the time derivatives in a very efficient way. This is based on introducing local variables **ã** for the field variables **a**. It is convenient for the purposes of describing such interaction picture methods to introduce an abbreviated notation as:

$$
\mathcal{D}[\mathbf{a},\mathbf{r}] = \mathbf{A}[\nabla,\mathbf{a},\mathbf{r}] + \underline{\mathbf{B}}[\nabla,\mathbf{a},\mathbf{r}]\cdot\mathbf{w}(\mathbf{r}) \tag{71}
$$

Hence, provided that $\underline{\mathbf{L}}[\nabla]$ has no explicit space-dependence, we can write the differential equation as:

$$
\frac{\partial \mathbf{a}}{\partial t} = \mathcal{D}[\mathbf{a},\mathbf{r}] + \underline{\mathbf{L}}[\nabla]\cdot\mathbf{a}. \tag{72}
$$

### 3.5.1 Linear propagator

Next, we define a linear propagator. This is given formally by:

$$
\mathcal{P}(\Delta t) = \exp\left(\Delta t\underline{\mathbf{L}}[\nabla]\right). \tag{73}
$$

where $\Delta t = t - \bar{t}$, and $\bar{t}$ is the interaction picture origin. Transforming the field **a** to an interaction picture is achieved on defining:

$$
\tilde{\mathbf{a}} = \mathcal{P}^{-1}(\Delta t)\mathbf{a}. \tag{74}
$$

As a result, the equation of motion is:

$$
\frac{\partial \tilde{\mathbf{a}}}{\partial t} = \mathcal{D}[\mathcal{P}(\Delta t)\tilde{\mathbf{a}},t]. \tag{75}
$$

This allows an SPDE to be treated like an SDE, if transformations are used. These can be efficiently implemented using Fourier or discrete sine or cosine transforms. The xSPDE implementation of this currently requires either periodic or zero boundary conditions and a diagonal linear operator $L$ without space-dependence. The linear operator can have any derivative in the periodic case, but only even order derivatives in the Dirichlet and Neumann case.

As well as the linear term, derivatives and nonlinear functions that are not tractable with spectral methods can appear in the residual term $\mathcal{D}[\mathbf{a}, \mathbf{r}]$, where they are treated using finite difference techniques. As a result, while the interaction picture does not handle all derivative terms, it also does not restrict them from being used elsewhere in the equations.

Other methods exist in the literature. Improved convergence properties are obtained for some problems in a spectral picture using an exact solution of a linear part of the drift term [63, 64], or stochastic noise terms [65], as well as the Laplacian terms. The xSPDE code has user-definable functions that can be adapted to include these.

## 3.6 Fourier transforms

It is often useful to transform a field to implement the interaction picture, or to extract nonlocal correlation properties in space. The Fourier transforms or spectrum definitions used in xSPDE are given by the symmetric Fourier transform definition:

$$
\tilde{a}(\mathbf{k}) = \mathcal{F}(a(\mathbf{x}))
$$
$$
= \frac{1}{[2\pi]^{(d-1)/2}} \int e^{-i\mathbf{k}\cdot\mathbf{x}} a(\mathbf{x}) d\mathbf{x}. \tag{76}
$$

The inverse Fourier transform is the function:

$$
a(\mathbf{x}) = \mathcal{F}^{-1}(\tilde{a})
$$
$$
= \frac{1}{[2\pi]^{(D-1)/2}} \int e^{i\mathbf{k}\cdot\mathbf{x}} \tilde{a}(\mathbf{k}) d\mathbf{k}. \tag{77}
$$

In simulations, this is not combined with any time (or space) averaging as in the temporal Fourier transforms. The reason for this is that the interaction picture transformations must be invertible, which is the case for a point-based discrete Fourier transform.

### 3.6.1 Normalization

During propagation, we define temporary internal fields $A(\mathbf{k_n})$, that are normalized using FFT conventions:

$$
A(\mathbf{k_n}) = \sum_{j_2=1}^{N_2} \cdots \sum_{j_d=1}^{N_d} e^{-i\mathbf{k}_n\cdot\mathbf{x_j}} a(\mathbf{x_j})
$$
$$
a(\mathbf{x_j}) = \frac{1}{\prod_{k=2}^{D} N_k} \sum_{n_2=1}^{N_2} \cdots \sum_{n_D=1}^{N_D} e^{i\mathbf{k_n}\cdot\mathbf{x_j}} A(\mathbf{k_n}). \tag{78}
$$

Otherwise, for graphical and output averages, we define Fourier transforms using physics and mathematics conventions:

$$\tilde{a}(\mathbf{k_n}) = \prod_{d=2}^{D} \left[ \frac{\Delta x_d}{\sqrt{2\pi}} \right] \sum_{j_2=1}^{N_2} \cdots \sum_{j_d=1}^{N_D} e^{-i\mathbf{k}_n \cdot \mathbf{x_j}} a(\mathbf{x_j})$$

$$a(\mathbf{x_j}) = \prod_{d=2}^{D} \left[ \frac{\Delta k_d}{\sqrt{2\pi}} \right] \sum_{n_2=1}^{N_2} \cdots \sum_{n_d=1}^{N_D} e^{i\mathbf{k_n} \cdot \mathbf{x_j}} \tilde{a}(\mathbf{k_n}) . \tag{79}$$

Note that this rescaling is consistent, because

$$\Delta x_d \Delta k_d = \frac{2\pi}{N_d}. \tag{80}$$

## 3.7 Trigonometric transforms

Taking the interaction picture approach, we now consider other types of boundary conditions, which we assume here are either a zero field (Dirichlet) or a zero derivative (Neumann). We will only treat cases of even order derivatives, which do not change the trigonometric function. Any odd order derivatives are taken to be included in the finite difference ($\mathcal{D}$) term.

In the spectral transform method in one space dimension, one uses a trigonometric function, $T(kx) = T_1 \sin(kx) + T_2 \cos(kx)$ to expand as:

$$a_i(t, x) = \sum_n a_{i,n}(t) T(k_{i,n} x), \tag{81}$$

The discrete inverse transform allows evaluation at sample points $x_j$, in order to satisfy the boundary conditions:

$$a_{i,n}(t) = \sum_j a_i(t, x_j) \tilde{T}(k_n x_j), \tag{82}$$

The trigonometrical function is defined such that:

$$\partial_x^{2p} T(kx) = \left( -k^2 \right)^p T(kx). \tag{83}$$

### 3.7.1 Propagator solution

This means that the propagator equation is now soluble for the sampled points, since for each component

$$\mathcal{L} \cdot a(t, x_j) = \sum_{ijn} \mathcal{L} a_n(t) T(k_n x_j),$$

$$= -\sum_{ijnp} L_p \left( -k_n^2 \right)^p a_n(t) T(k_n x_j)). \tag{84}$$

Hence,

$$a_n(t) = \exp\left( \sum L_p \left( -k_n^2 t \right)^p t \right) a_n(0). \tag{85}$$

This is an exact solution, provided the initial condition has the given expansion. There are no approximations made on the transverse derivative. Provided the $k$ values are the same, this propagator is identical for all types of trigonometric and Fourier transforms.

As explained above in (3.2), there are five types of boundary combinations that are possible in each dimension and field component. Each has a corresponding xSPDE boundary type and spectral integrator. Just as with the derivative term, each of these types can change with dimension and field component.

Currently, all can be treated in xSPDE using finite differences, and each type of boundary has a particular spectral method that preserves the boundary requirement. In principle, more than one transform can be used. It is possible to define the relevant trigonometric transforms to correspond to whole symmetries whose boundary is either at a grid point, or half symmetries which are half-way between two grid points.

Differential equations can also have first order terms, which currently require using finite differences or periodic boundaries. These make use of boundaries at a grid point, in order to compute the relevant terms, which means that there is greater compatibility with the finite difference methods when the boundaries are at the grid points. With this restriction, the available transforms are reduced.

It is possible to compute first-order derivatives with spectral methods, but these turn sine transforms into cosine transforms. This is not compatible with simple interaction picture transformations, except for the periodic case.

In summary, spectral transforms can all be implemented using fast FTT, discrete sine (DST) or cosine (DCT) transforms, but the trigonometric method to be used is specific to the boundary type. The definitions used here mostly correspond to the definitions used in the FFTW [66, 67] software.

## 3.8 Transforms and boundaries

Suppose that there are Dirichlet or Neumann boundaries, then the following expansion can be employed in each dimension. We only describe one space dimension for simplicity with:

$$a = \sum_{n=1}^{\infty} \left[ S_n \sin(k_n x) + C_n \cos(k_n x) \right] e^{\sum L_p (-k_n^2)^p t}, \tag{86}$$

where $k_n, C_n, S_n$ are chosen to satisfy the initial and boundary conditions. Boundaries are taken, for the purposes of explanation, as being from $x = 0$ to $x = R$. This is not quite the case in the actual code, which can treat arbitrary boundary locations due to the use of the optional *origins* input to change the origin.

Suppose there are $N$ computational grid-points. For the spatial grid (1-based), this corresponds to $x_n = (n-1) \Delta x$, $n = 1, ..., N$ with $\Delta x = \frac{R}{N-1}$ , so we have $x_1 = 0$ and $x_N = R$.

In carrying out a discrete transform on $N_T$ points, using standard trigonometric transform definitions, there may be less grid points required, so we may have $N_T \leq N$ if some of the boundary values are defined due to Dirichlet boundaries. Internal xSPDE definitions always use the full computational grid range, $N$, which *includes* boundary values.

An unnormalized inverse results in the original array multiplied by $N_{FT}/2 = (N-1)/2$, where $N_{FT} = N - 1$ is the FFTW 'logical' size, so our definitions include a factor of $\sqrt{2/(N-1)}$. Here $N_T$, the number of points in the standard DST/DCT definitions, can differ from **both** the xSPDE computation grid size $N$ that includes both boundaries, and also from the FFTW 'logical' size, which always includes one (periodic) boundary.

The notation in this section is based on the usual discrete sine and cosine transform (FFTW) definitions. However, we use 1−based indices, often found in mathematics and in Octave/Matlab/Julia/Fortran.

For all coordinates, including these examples of discrete Fourier transforms, we remind the reader that:

$$r_n^d = R^d (n-1)/(N^d - 1).$$ (87)

If we regard the transforms as having arguments of form $k_n (x_0 + x_n)$, the momentum spacings given below are such that:

$$\Delta k = \frac{\pi}{R}$$
$$\Delta x \Delta k = \frac{\pi}{N-1}.$$ (88)

These internal definitions used in the propagator calculations are therefore different to those used in external graphs and in periodic boundary cases.

**The following lists the inverse trigonometric transforms required to obtain $a(x)$ from $a_k$, for the four different non-periodic boundary types in each dimension and field index.**

### 3.8.1  D-D case: Discrete map (DST-I)

Take $a(0) = a(R) = 0$, with a sine transform. The representation of $a$ is

$$a(x_n, t) = \sqrt{\frac{2}{N-1}} \sum_{j=2}^{N-1} a_j(t) \sin\left(\pi \frac{(j-1)(n-1)}{N-1}\right).$$ (89)

**Forward transform: this is also DST-I**

### 3.8.2  R-R case: Discrete map (DCT-I)

Take $a'(0) = a'(R) = 0$. The representation of $a$ is

$$a(x_n, t) = \sqrt{\frac{2}{N-1}} \left(\frac{1}{2}\left(a_1 + (-1)^{n-1} a_N\right) + \sum_{j=2}^{N-1} a_j(t) \cos\left(\pi \frac{(j-1)(n-1)}{N-1}\right)\right).$$ (90)

**Forward transform: this is also DCT-I**

### 3.8.3  D-R case: Discrete map (DST-II)

Take $a(0) = a'(R) = 0$. The representation of $a$ is:

$$a(t, x_n) = \sqrt{\frac{2}{N-1}} \left(\sum_{j=1}^{N-1} a_j(t) \sin\left[\frac{\pi}{N-1}\left(j - \frac{1}{2}\right)n\right]\right).$$ (91)

**Forward transform: this is DST-III.**

### 3.8.4   R-D case Discrete map (DCT-II)

Take $a'(0) = a(R) = 0$. The representation of $a$ is:

$$a(t, x_n) = \sqrt{\frac{2}{N-1}} \sum_{j=1}^{N-1} a_j(t) \cos\left[\frac{\pi}{N-1}\left(j - \frac{1}{2}\right)(n-1)\right]. \tag{92}$$

**Forward transform: this is DCT-III.**

## 3.9   Frequency or momentum grid

The frequency or momentum grid spacing is defined for all *output* graphs and periodic Fourier transforms as

$$\Delta k = \frac{2\pi}{N\Delta x}. \tag{93}$$

However, the internal momentum grid spacing used can differ from this, depending on the transforms used in the interaction picture. As explained above in Subsection (3.8), the internal momenta for trigonometric transforms are:

$$\Delta k = \frac{\pi}{(N-1)\Delta x}. \tag{94}$$

This is because the xSPDE algorithms allow the use of a sequence of interaction pictures. Each successive interaction picture is referenced to $t = t_n$, for the n-th step starting at $t = t_n$, so $a_I(t_n) = a(t_n) \equiv a_n$. It is also possible to solve stochastic partial differential equations in xSPDE using explicit derivatives, but this is less efficient.

A discrete Fourier transform (DFT) using a fast Fourier transform method is employed for the interaction picture (IP) transforms used with periodic boundaries. This is normalized differently to the graphed Fourier transforms, but the difference is not computationally significant. However, the $\Delta k$ used internally changes with the precise type of trigonometric transform used in other cases.

In one dimension, the DFT is usually defined by a sum over indices starting with zero, rather than the Matlab convention of one. Hence, if $\tilde{m} = m - 1$:

$$A_{\tilde{n}} = \mathcal{F}(a) = \sum_{\tilde{m}=0}^{N-1} a_{\tilde{m}} \exp\left[-2\pi i \tilde{m}\tilde{n}/N\right]. \tag{95}$$

For periodic boundaries, the IP Fourier transform can be written in terms of an FFT as

$$A(k_n) = \prod_j \left[\sum_{\tilde{m}_j} \exp\left[-i\left(dk_j dx_j\right)\tilde{m}_j\tilde{n}_j\right]\right]. \tag{96}$$

The inverse FFT Fourier transforms divide by the correct factors of $\prod_j N_j$ to ensure invertibility. Due to the periodicity of the exponential function, negative momenta are obtained if we consider an ordered lattice such that:

$$\begin{aligned} k_j &= (j-1)\Delta k \ (j \leq N/2) \\ k_j &= (j-1-N)\Delta k \ (j > N/2) \end{aligned}. \tag{97}$$

This Fourier transform is then multiplied by the appropriate factor to propagate in the interaction picture, then an inverse Fourier transform is applied. While it is not scaled for interaction picture transforms, an additional scaling factor is applied to obtain transformed fields in any averages for output plots.

In other words, in the averages

$$\tilde{a}_n = \frac{\Delta x}{\sqrt{2\pi}} A_{\tilde{n}'}. \tag{98}$$

where the indexing change indicates that graphed momenta are stored from negative to positive values. For plotted frequency spectra a **positive** sign is used in the frequency exponent of the transform to frequency space, to agree with common physics conventions.

### 3.10 Derivatives

#### 3.10.1 Spectral derivatives

For spectral derivatives in the interaction picture, we define $D_x(k)$ to obtain a derivative. To explain, one integrates by parts:

$$D_x^p \tilde{a}(k) = [ik_x]^p \tilde{a}(k) = \frac{1}{(2\pi)^{d/2}} \int dx e^{-ik\cdot x} \left[\frac{\partial}{\partial x}\right]^p a(x). \tag{99}$$

This means, for example, that to calculate a one dimensional space derivative in a Fourier interaction picture routine, one uses:

$$\nabla_x \rightarrow D_x. \tag{100}$$

Here $Dx$ is an array of momenta in cyclic order in dimension $d$ as defined above, suitable for an FFT calculation. The imaginary $i$ is not needed to give the correct sign, as it is included in the derivative array. In two dimensions, a full two-dimensional Laplacian is:

$$\nabla^2 = \nabla_x^2 + \nabla_y^2 \rightarrow D_x^2 + D_y^2. \tag{101}$$

Then, on inverting the transform

$$\left[\frac{\partial}{\partial x}\right]^p a(x) = \frac{1}{(2\pi)^{d/2}} \int dx e^{ik\cdot x} [D_x(k)]^p \tilde{a}(k). \tag{102}$$

#### 3.10.2 Finite difference derivatives

For calculating derivatives using finite differences, the following central differencing method is used, away from the boundaries:

$$\nabla_x a(x_i) \rightarrow \frac{1}{2\Delta x} [a(x_{i+1}) - a(x_{i-1})]$$

$$\nabla_x^2 a(x_i) \rightarrow \frac{1}{\Delta x^2} [a(x_{i+1}) - 2a(x_i) + a(x_{i-1})]. \tag{103}$$

This raises the question of how to calculate derivatives at the boundary, for example at the lower boundary $x_1$, where $a(x_0)$ is not known, and similarly at the upper boundary. The answer depends on the boundary type [68], and is obtained by extending the boundary to additional points $a(x_0)$ and $a(x_{N+1})$ that are assumed to extend the boundary condition:

**Periodic:** $a(x_0) = a(x_N)$

$$\nabla_x a(x_1) \rightarrow \frac{1}{2\Delta x} [a(x_2) - a(x_N)]$$

$$\nabla_x^2 a(x_1) \rightarrow \frac{1}{\Delta x^2} [a(x_2) - 2a(x_2) + a(x_N)]. \tag{104}$$

**Dirichlet:** $\tilde{a}(x_1)$ **specified:** $a(x_0) = \tilde{a}(x_1)$

$$\nabla_x a(x_1) \rightarrow \frac{1}{2\Delta x} [a(x_2) - \tilde{a}(x_1)]$$

$$\nabla_x^2 a(x_1) \rightarrow \frac{1}{\Delta x^2} [a(x_2) - \tilde{a}(x_1)]. \tag{105}$$

**Robin/Neumann:** $\tilde{a}'(x_1)$ **specified:** $a(x_0) = a(x_2) - 2\tilde{a}'(x_1)\Delta x$

$$\nabla_x a(x_1) \rightarrow \tilde{a}'(x_1)$$

$$\nabla_x^2 a(x_1) \rightarrow \frac{2}{\Delta x^2} [a(x_2) - a(x_1) - \tilde{a}'(x_1)\Delta x]. \tag{106}$$

In all cases the boundary value is evaluated as part of the derivative evaluation, so it can be a nonlinear function of **a**.

# 4 Solving an SDE

*This section describes how to use the xSPDE numerical toolbox to solve an SDE to obtain and graph averages, spectra or probability distributions.*

## 4.1 Using xSPDE

Stochastic equations have very few analytic solutions, except in unusually simple cases. They generally require numerical solutions. A stochastic toolbox helps to streamline the job of writing and executing code. The xSPDE simulation program is straightforward to use, and provides many options. To run it, an Octave or Matlab environment is needed.

The current xSPDE distribution includes:

- The toolbox: $xspde.mltbx$, or a folder: xSPDE.

- simulation (xSIM) and graphics (xGRAPH) functions.

- xAMPLES: examples that can also be used as templates

- xDOC: should contain the current user guide

xSPDE can be run interactively as a script, or as a function in batch mode, either at a local workstation or on a remote cluster. Data can be either plotted immediately, or saved then plotted later. To simulate a stochastic equation interactively, first check that the xSPDE toolbox is installed.

**If you have the toolbox file,** $xspde.mltbx$, **just open it and click on** $install$**.** Otherwise the Octave/Matlab path must point to the xSPDE folder and subfolders. If you have the folders, but not the toolbox, proceed as follows:

- Click on the Octave/Matlab HOME tab (top left), then Set Path

- Click on Add with Subfolders

- Find the xSPDE folder in the drop-down menu, and select it

- Click on close to save the path.

Type $clear$ to clear old data, and enter the xSPDE inputs and functions into the command window. For the simplest cases, one can do this by cutting and pasting from an electronic file of this manual.

For more advanced cases, it is best to create a function that calls xSPDE. The Octave/Matlab built-in editor is also useful. There are many examples listed in this manual, and there are more in the xAMPLES folder. Any of these can be used as templates for building your own simulation.

### 4.1.1 Wiener process

To solve for a single trajectory of Eq (21) with xSPDE, just type in:

```
p.deriv = @(a,w,p) w;
xspde(p);
```

Here $p.deriv$ defines the time derivative $\dot{a}$ in the input parameter structure $p$, while $w$ is a delta-correlated Gaussian noise generated internally. There are no other parameters, so default values are used. This produces the graph shown in Fig (1), which gives a single trajectory.

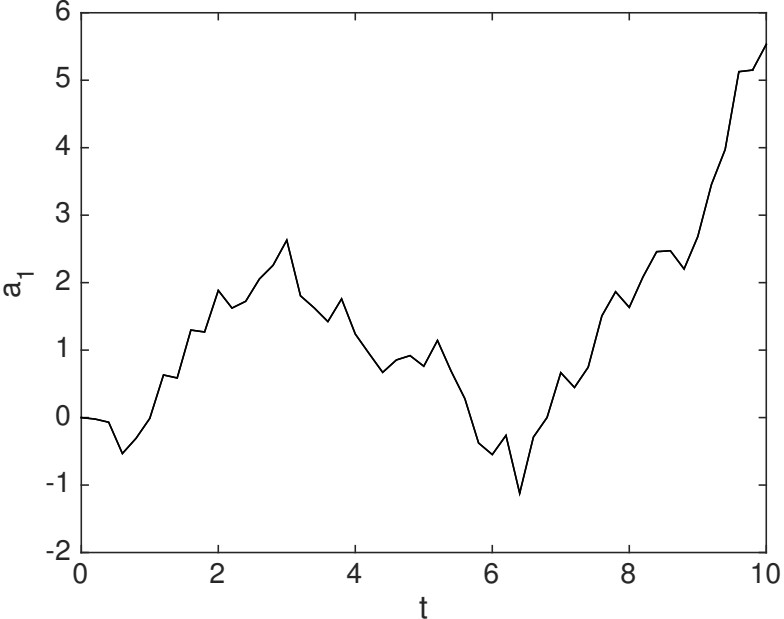

Figure 1: The simplest example: a random walk.

At the end of the run, xSPDE reports the RMS errors. There are discretization, sampling and comparison errors, all normalized by the maximum observable value, unless compared to a result of zero. In the present simulation, the discretization or step error is about $10^{-16}$, due to round-off. This is just a single trajectory, but more can be added.

### 4.1.2 General derivatives

All important xSPDE procedures use functions. Functions can be specified inline, which is the simplest, or externally. The last argument of any xSPDE function is the parameter structure. An example already introduced is the derivative function, labeled $p.deriv$.

For example, consider the stochastic differential equation,

$$\frac{da}{dt} = -ga + w. \tag{107}$$

The corresponding derivative code definition is:

```
p.deriv = @(a,w,p) - p.G*a + w;
```

This code defines the function handle $p.deriv$, which gives the derivative function, $da/dt$. In this example, it simply returns the derivative, in terms of the variable $a$, loss parameter $p.G$, and stochastic noise term $w$. This user specified inline function is known internally by the function handle $p.deriv$.

Inside a complete xSPDE simulation input with a parameter values, it would look like:

```
p.G = 0.25;
p.deriv = @(a,w,p) - p.G*a + w;
xspde(p);
```

External function handles can also be used. They are useful for complex functions with more internal logic.

A typical script first defines parameters and function specifications, in a structure, then runs the simulation code with the parameter structure as an input, as follows:

```
p.[label1] = [parameter1];
...
p.[label2] = [parameter2];
p.deriv = @(a,w,p) [derivative];
xspde(p);
```

Note the following points to remember:

- p.[$label1$] = [$parameter1$] defines a parameter in the structure $p$.

- There are many possible inputs, which all have default values.

- You don't have to save the data if you want an immediate plot.

- The notation p.deriv = @(a, w, p) [$derivative$] defines a function, $da/dt$.

- In this example, a is the stochastic variable, w the random noise, $p$ a structure.

- Other labels can be used instead of *(a,w,p)* if preferred.

## 4.2 Input parameters

All xSPDE simulations use a structure for input data. Most functions also require a parameter structure, combining the data input with additional internal parameters. Any naming convention will do for either structure, as long as you are consistent.

User-defined parameters can be added freely. To ensure that there is no clash with internal variables, it is best if user defined parameters start with a capital letter.

The xSPDE inputs have default values, which are used if the input values are omitted. If you only need the first element of a vector or array, just input the value required. Parameters can be output with the verbose switch, p.verbose. This has four levels of output: $-1, 0, 1$ or $2$, with p.verbose=0 as default, giving final error reports. To get more progress details and individual errors, use p.verbose=1. To eliminate almost everything, use p.verbose=-1. For maximum information, including all the internal parameter values, use;

```
p.verbose = 2;
```

While this level of detail is not usually needed, it can be useful to print out all the internal parameters and default values to understand how the program operates.

### 4.2.1  Simulation parameters table

The most common xSPDE input parameters used to define the equations in a simulation, together with their default values are:

| Label | Type | Default value | Description |
|:---:|:---:|:---:|:---:|
| *fields* | integer vector | 1 | Number of stochastic *fields* |
| *noises* | integer vector | 1 | Number of *noises* |
| *name* | string | ' ' | Simulation *name* |
| *deriv* | function | 0 | The stochastic *deriv*ative |
| *initial* | function | 0 | Function to *initial*ize variables |
| *method* | function | [see 7] | Integration *method* |
| *ensembles* | integer vector | [1,1,1] | Stochastic *ensemble* sizes |
| *ranges* | real vector | [10] | Time and space *ranges* |
| *points* | integer vector | [51] | Output lattice *points* in [t,x,y,z,..] |
| *steps* | integer | [1] | Intermediate *steps* per time point |
| *observe{n}* | function | *a* | *Observ*able function for averages |
| *compare{n}* | function | *0* | *Compari*son function for averages |
| *binranges{n}{m}* | vector | [0] | *Bin*ning *ranges* for probabilities |

A more detailed explanation of these parameters is found below, and a complete table is given in section 9.2.

### 4.2.2  Graphics parameters

The generated average data can be graphed using any graphics editors, or else using the internal xGRAPH function defined for this purpose. An xSPDE simulation can return many different averages. These are defined in a cell array with indices in braces. The index is used to address the output data produced.

For each index, one can define parameters that define the quantity stored, together with corresponding graphics outputs. Some commonly used options are:

| Label | Type | Default value | Description |
|:---:|:---:|:---:|:---:|
| *olabels{n}* | string | 'a' | Observable label |
| *transverse{n}* | integer | 0 | Transverse slices in time |
| *transforms{n}* | vector | 0 | Set to 1 for Fourier transforms in time |
| *scatters{n}* | integer | 0 | Set to s for s scatter plots in the observable |

The full definition of the options is given in the user guide in sections 9.4 and 10, although most will be clear from examples.

### 4.3  Fields and noises

Stochastic variables in an SDE are *fields*, stored in a real or complex matrix, $a(i, j)$. Here, $i$ is an internal field index, while $e$ is the ensemble index.

**fields** gives the range of the first internal index. This is the total number of SDE variables or fields. It has a default value of $fields = 1$.

**ensembles** allows multiple trajectories to be integrated. This has up to three components. The first component, *ensembles(1),* gives a vector of local trajectories, so $e = 1, \ldots ensembles(1)$. The two other ensemble values specify serial or parallel processing, as explained below.

**noises** are noise dimensions, similar to *fields*, and used as $w(i, j)$, where the first noise index has *noises* components. The default value is $noises = fields$.

In the example above, we could add the fields, dimensions, ensembles and noises:

```
p.fields = 1;
p.dimensions = 1;
p.noises = 1;
p.ensembles = 1;
```

As these are all default values, this is superfluous in a simple case. The full definition of ensembles as a vector is given below, and in some cases uses the parallel toolbox in Matlab.

### 4.3.1 Initial values, points and ranges

Initial values are required to define any differential equation, and in a numerical calculation one must also have a defined lattice.

**initial** The initial value is defined by a function $p.initial$. This must return either an initial vector of size *fields*, or else a random array of size $fields \times ensembles(1)$. The default function simply returns zero.

**inrandoms** are initial random number dimensions, similar to *fields*, and used as $v(i, j)$, where the first random dimension has $randoms$ components. The default value is $randoms = noises$. Specifies the first argument of the function $p.initial(v, p)$ as a real Gaussian noise vector $v$ with unit variance. The same noise is used when error-checking, so that changes are from the step-size, not from random fluctuations.

**points** The number of integration points. The default setting is currently 51.

**steps** The number of integration steps used for each output time-step. The default is 1.

**ranges** The total integration range in each dimension, the first element being the maximum integration time $T$. The default setting is currently 10.

### 4.3.2 Observables

**observe** is a cell array of functions of stochastic fields, each defining an average. xSPDE expects a (named or anonymous) function that takes two parameters, namely the field matrix $a$ and the input structure $p$. The function must return a real or complex matrix of dimension $(\ell, ensembles(1))$, where $\ell$ indexes over a vector observable. xSPDE then averages over the second index, to calculate the observable.

To plot the variance, for example:

```
p.observe{1} = @(a,p) (a(1,:)-mean(a,2)).^2;
```

**rawdata** By setting *p.rawdata=1* (see section 9.4), one can also store every trajectory including both fine and coarse time-step values, and but this is very memory-intensive for large simulations.

**olabels** is cell array of graph labels associated with each average, although one can also define a function of the averages to be graphed with this label.

Observables are computed as a two-dimensional packed array, then unpacked for storage, giving an array of dimension $(d1, dspacetime, ensemble(1))$. Here $d1$ is the local observable dimension, so $d1 = 1$ for a scalar observable. The space-time dimension is $dspacetime = 1$ for an SDE, otherwise a vector for a SPDE, and $ensemble(1)$ is the size of the ensemble of trajectories computed in each processor. Once data is averaged internally over $ensemble(1)$, further transforms of the resulting averages are available.

### 4.3.3 Using the dot

All equations entered in xSPDE utilize the Matlab syntax. This is designed to handle scientific or mathematical matrix and array-based formulae. It has features to simplify matrix or array equations which often require a 'dot' or a 'colon'.

- Stochastic variables in xSPDE are matrices or arrays, where the last index is used to treat parallel stochastic trajectories, for greater efficiency. This requires use of the 'dot' notation to perform multiplication inside equations.

- To multiply vectors, matrices or arrays element-wise, like $a_{ij} = b_{ij}c_{ij}$, the notation $a = b. * c$ indicates that all the elements are multiplied. This is used to speed up calculations in parallel.

- An equation in xSPDE can apply to many stochastic trajectories in parallel. Using the dot shortens the equation, and it also means that a fast parallel arithmetic will be used. The same principle holds for larger arrays with spatial lattices, treated in in section 5.

- Broadcasting occurs if one or more dimensions has a unit size. For example, arrays of size (1,100) and (6,1) can be added or multiplied to give a (6,100) matrix.

- A formula for a stochastic field may require you to address the first index - which is the field component - and treat all the other elements in parallel. To do this in a compact way, one may use the notation $a(1, :)$, which indicates that all the subsequent index elements are being addressed as well.

- For an an SPDE this can "flatten" a larger array into a matrix. This requires care to make sure all the terms have the same dimensionality as described in section 5. xSPDE includes routines to help this issue.

In summary, whenever a formula combines multiplication operations over spatial lattices or ensembles, **USE THE DOT**.

## 4.4 Advanced random walk

We now return to the random walk, but with some more advanced features:

$$\dot{a} = w(t), \tag{108}$$

This is integrated numerically and graphed with $N = points(1)$ points. The first point stored is the initial value, so there are $N-1$ integration steps, of length $dt = ranges(1)/(N-1)$. Numerical graphs have discrete steps, and more detail is obtained if more time steps are used. The default value is $N = 51$, which is predefined in the $xpreferences$ file. This is adjustable by the user. It can also be changed for a simulation, by inputting a new value of $points$.

### 4.4.1 Simple xSPDE example

Unless you type *clear* first, any changes to the input structure are additive; so in the exercises you should get the combination of all the previous structure inputs as well as your new input.

- **Run the complete xSPDE script of Example 1 in Matlab.**

It is simple to cut and paste from an electronic file to the command window. Be careful; pasting can cause subtle changes that may require correction. Some generated characters may be invalid input characters, and these will need retyping if this occurs.

You should get the output in Fig (1).

- **What do you see if you average over** 10000 **trajectories ?**

```
p.ensembles = 10000;
xspde(p);
```

- **What do you see if you plot the mean square distance? Note that variances should increase linearly with** $t$**.**

```
p.observe = @(a,p) a.^2;
p.olabels = '<a^2>';
xspde(p);
```

- **What if you add a force that takes the particle back to the origin?**

$$\dot{a} = -a + w(t), \tag{109}$$

```
p.deriv = @(a,w,p) -a+w;
xspde(p);
```

The corresponding Fokker-Planck equation from Eq (17) is:

$$\frac{\partial P(a)}{\partial t} = \left[ \frac{\partial}{\partial a} + \frac{1}{2}\frac{\partial^2}{\partial a^2} \right] P(a). \tag{110}$$

It is easy to verify that inserting this dynamical equation into Eq (19) gives the result:

$$\frac{\partial}{\partial t}\left\langle a^2 \right\rangle = 1 - 2\left\langle a^2 \right\rangle \tag{111}$$

- Solve for $\left\langle a^2(t) \right\rangle$ and use xSPDE to compare the numerical and analytic solutions. The current time is accessible as the parameter $p.t$. Can you explain the graph differences?

## 4.5 Probability binning

It is possible to graph probability densities of real observables instead of averages, if $p.ensembles$ is large. This is achieved by inputting the observable number and binning range:

$$p.binranges\{n\} = \{oa : ostep : ob\}; \tag{112}$$

If present, this returns probability density of the $n$-th observable $o\{n\}$, through binning into ranges of width $ostep$ around the centers of each bin, starting at $oa$, and ending at $ob$. The simulation returns a result of $1/ostep$ in the $j - th$ bin if the trajectory is inside the bin, so that $o(j) - ostep/2 < o < o(j) + ostep/2$, and zero otherwise. This gives a probability density on output, plotted against time. Note that on graphing, an extra dimension is added for the variable $o$. The probability density at $ntimes$ equally spaced simulation times can be plotted with $p.transverse\{n\}=ntimes$.

The probability can be plotted for any $observe$ function of the stochastic variable.

### 4.5.1 Multivariate probabilities

The probability density is multivariate for vector observables. This is possible because the binning ranges are stored in a cell array, which may contain several bin vectors. If the observable $o\{n\}$ is two-dimensional, then one can input:

$$p.binranges\{n\} = \{oa(1) : ostep(1) : ob(1), oa(2) : ostep(2) : ob(2)\}; \tag{113}$$

On graphing, $two$ extra dimensions are added for the variable $o$ in this case. The graphics program $xGRAPH$ will attempt to graph them, but it is limited by graphical visualization constraints. In general, an arbitrary observable dimension is possible, but this is also limited by the sampling and memory, since the number of samples per bin will decrease rapidly with dimensionality.

The graphics program extracts slices and windows of probabilities if required. To plot the probabilities of two observables, one for a range of $-5 : 5$ and the other for 0:25 for a range of 0:1, add the following inputs before the $xsim$ or $xspde$ command:

```
p.binranges{1} = {-5:0.25:5};
p.binranges{2} = {0:0.5:25};
```

In the case of a two-dimensional probability density, plotted against time, there are a total of four graphics dimensions. That is, one dimension for time, two for the observable dimensions, and one for the probability itself. One can also plot how the probability density changes in space for the case of a stochastic partial differential equation, as described in section 5.

## 4.6 Auxiliary fields and noises

In some problems, it is useful to access the noise terms, or functions of the noises and their correlations with the fields at the same time. This is handled in xSPDE with auxiliary fields or $auxfields$. These are fields that are functions of noise terms and the integrated fields. The number of these is defined in the input structures using the parameter $p.auxfields$, which is arbitrary.

Auxiliary fields are calculated using a function $p.define$, which is similar to $p.deriv$, except that it returns the current value of the auxiliary field, not the derivative. These fields are defined as

the *average* over the previous step in time of the auxiliary function. This is essential in calculating spectra, in order to eliminate systematic errors in Fourier transforms.

More details on this are given in Section (4.7). To access the auxiliary fields, one can compute any observable average using a *p.observe* function as usual, or else store the *raw* trajectories including auxiliary fields by setting *p.rawdata=1*. In either case, the auxiliary fields are appended to the integrated fields by adding extra rows to the field matrix. The number of variables or matrix rows for calculating averages in observe or rawdata is *fields+auxfields*.

### 4.6.1 Outputting the noise

As a simple example, suppose one wishes to calculate the noise terms and compare them with the field trajectories in a simple Wiener process. The following code can be used:

```
clear
p.auxfields = 1;
p.deriv = @(a,w,p) w;
p.define = @(a,w,p) w;
p.olabels = {'a, w'};
xspde(p);
```

The default observe function is used. This plots both rows of the field array, including the auxiliary field which is defined as the noise term and plotted as a dashed line. There is no ensemble averaging, and hence no ensemble error-bars. This is simply because because no ensembles were specified in the input parameters. Similarly, there are no time-step error-bars for this observable, because the fine and coarse noises are equal to each other after time averaging.

The result that is plotted is therefore the coarse noise, whose correlation time equals the time step. This is plotted below in Fig ( 2), which plots the same Wiener process as before, except adding the driving noise term as well. The standard deviation of the noise in a single step here is $\sqrt{1/dt}$, where $1/dt = 50/10 = 5$ for the default range of 10 and default time points of 51. Note that noise terms do not converge at small time-steps for delta-correlated noise, even when the integrated stochastic process does converge. This is why it is necessary to choose to plot one or the other, or else to time-average to obtain a converged result.

If multiple steps are used, the noise during the last step prior to the time-point is plotted.

## 4.7 Time-domain spectra

To get an output from a temporally Fourier transformed field, set $transforms\{n\} = 1$ for the observable ($n$) you need to calculate in transform space. This parameter is a cell array. It can have a different value for every observable and for every dimension in space-time, if you have space dimensions as well.

To obtain spectra from Eq (37) with greater accuracy, all fields are must be averaged internally. The code will use trapezoidal integration in time over the integration interval, to give the average midpoint value. This employs the same interval for fine and coarse integration, to allow comparisons for error-checking. After this, the resulting step-averaged fields are then Fourier transformed.

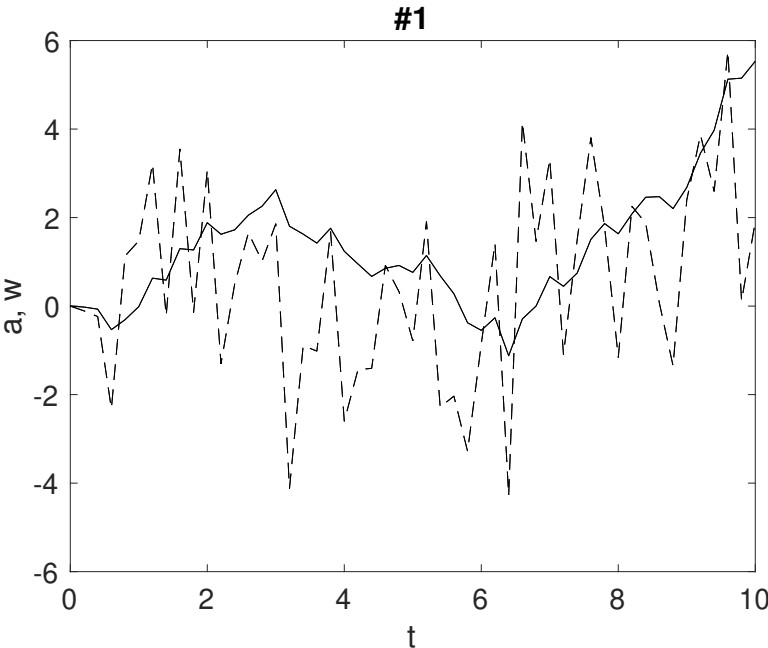

Figure 2: A single trajectory of a random walk, with the noise terms $w$ graphed using dashed lines, and the integrated variable $a$ plotted as the solid line..

In the simplest case of just one internal step, with no error-checking, this means that the field used to calculate a spectrum is:

$$\bar{a}_j = \left(a_{j-1} + a_j\right)/2, \tag{114}$$

which corresponds to the time in the spectral Fourier transform of:

$$\bar{t}_j = \left(t_{j-1} + t_j\right)/2. \tag{115}$$

Note that if any temporal Fourier transform is specified, all the field variables are time-averaged over a step. This is not strictly necessary, but it means that there is a reduced code complexity for cases where there is a Fourier transform for some but not all variables. As described above, the auxiliary variables are always time-averaged to allow error-checking, so there is no change for these.

### 4.7.1   Error-checking

For an error-checking calculation with two internal *steps*, there are three successive valuations: $a_{j-1}$, $a_{j-1/2}$, $a_j$. In this case, for spectral calculations one averages according to:

$$\bar{a}_j = \left(a_{j-1} + 2a_{j-1/2} + a_j\right)/4. \tag{116}$$

In addition, one must define the first field $\bar{a}_1$. Due to the cyclic nature of discrete Fourier transforms, this is also logically the last field value. This is set equal to the corresponding cyclic average

of the first and last field value, in order to reduce potential aliasing errors at high frequencies in the resulting spectrum:

$$\bar{a}_1 = \frac{1}{2}(a_N + a_1), \tag{117}$$

which corresponds to a time in the spectral Fourier transform of:

$$\bar{t}_1 = t_1 - dt/2 \sim t_N + dt/2. \tag{118}$$

The time integral is carried out numerically as a sum which has $N = points(1)$ time points of interval $dt$. In xSPDE, $dt = T/(N-1)$, where $T = ranges(1)$. The 'effective' integration time for the Fourier transform time integrals is $T_{eff} = Ndt = 2\pi/d\omega = T \times N/(N-1) = T + dt$. Aliasing of virtual times higher and lower than the integration time is due to the discrete Fourier transform.

When there are larger numbers of steps, from using the internal *steps* parameter, there are more points to Fourier transform. These additional frequencies are computed while carrying out the Fourier transform, but only the low frequency points near zero are saved. The unused high frequency results are not stored or plotted, to conserve memory.

## 4.8 Examples

### 4.8.1 Complex damped spectrum

Consider the spectrum of Eq (109), with a complex noise,

$$\langle w(t)w^*(t')\rangle = \delta(t-t'), \tag{119}$$

a random initial equation near the equilibrium value, and a range of $t = 100$, with 640 points. Here there are two real noises.

The input parameters are given below. There are parallel operations here, for ensemble averaging, so we **USE THE DOT**.

```
clear
p.points = 640;
p.ranges = 100;
p.noises = 2;
p.ensembles = 10000;
p.initial = @(v,p) (v(1,:)+1i*v(2,:))/sqrt(2);
p.deriv = @(a,w,p) -a + w(1,:)+1i*w(2,:);
p.observe = @(a,p) a.*conj(a);
p.transforms = 1;
p.olabels = '|a(\omega)|^2';
xspde(p);
```

Note that $p.\texttt{transforms} = 1$ tells xSPDE to Fourier transform the field over the time coordinate before averaging, to give a spectrum. Both *observe* and *transforms* could be cell arrays, but the this is not needed with a single observable. The first argument $v$ of the *initial* function is a random field, used to initialize the stochastic variable.

To define as many observables as you like, use a Matlab cell array;

```
p.observe{1} = ..;
p.observe{2} = ..;
```

To learn more, try the following:

- **Simulate over a range of $t = 200$. What changes do you see? Why?**

- **Change the equation to the laser noise equations introduced in the next section (Laser quantum noise). Why is the spectrum much narrower?**

### 4.8.2 Laser amplification noise

Laser quantum noise is commonly modeled [50–52] using SDEs in a normally ordered quantum phase-space representation. Consider a model for the quantum noise of a single mode laser as it turns on, near threshold:

$$\dot{a} = ga + bw(t) \tag{120}$$

where the noise is complex, $w = (w_1 + iw_2)$, so that:

$$\langle w(t)w^*(t')\rangle = 2\delta\left(t - t'\right). \tag{121}$$

Here the coefficient $b$ describes the quantum noise of the laser, and is inversely proportional to the equilibrium photon number.

Try the following, noting that you should type *clear* first when starting new simulations.

- **Solve for the case of $g = 0.1$, $b = 0.01$**

Most lasers have more than 100 photons and hence much less noise than this.

For this exercise, small error-bars will display on the graph. These are calculated from the difference between using steps of size $dt$ and steps of size $dt/2$. They only appear if greater than a minimum relative size, typically 1% of the graph size, which can be set by the user.

```
clear
p.noises = 2;
p.observe = @(a,p) abs(a)^2;
p.olabels = '|a|^2';
p.deriv = @(a,w,p) a + 0.01*(w(1)+1i*w(2));
xspde(p);
```

### 4.8.3 Saturated laser noise

Consider the case where the laser saturates to a steady state:

$$\dot{a} = \left(1 - |a|^2\right)a + bw(t) \tag{122}$$

To learn how to use the function inputs, try the following:

- **Solve for the saturated laser case**

You should get the output graph in Fig (3).

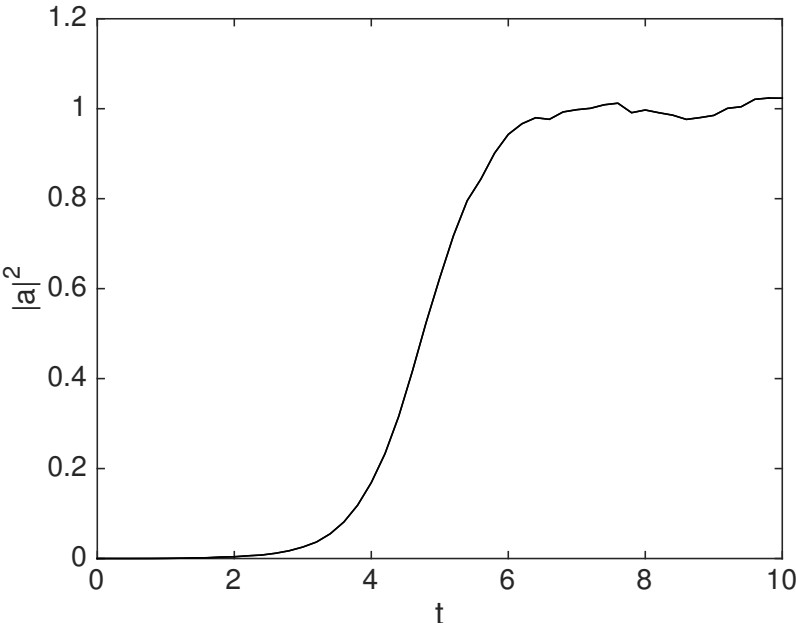

Figure 3: Simulation of the stochastic equation describing a laser turning on.

```
p.deriv = @(a,w,p) (1-abs(a)^2)*a+0.01*(w(1)+1i*w(2));
xspde(p);
```

### 4.8.4 Financial calculus

A well-known Ito-type stochastic equation is called the Black-Scholes equation [69], used to price financial options. It describes the fluctuations in a stock or commodity value:

$$da = \mu a\, dt + a\sigma\, dw, \tag{123}$$

where $\langle dw^2 \rangle = dt$. As the noise is multiplicative, the equation is different in Ito and Stratonovich calculus. The corresponding Stratonovich equation, as used in xSPDE for the standard default integration routine is:

$$\dot{a} = \left(\mu - \sigma^2/2\right)a + a\sigma w(t). \tag{124}$$

An interactive xSPDE script in Matlab is given below with an output graph in Fig (4). This is for a startup with a volatile stock having $\mu = 0.1$, $\sigma = 1$. The spiky behavior is typical of multiplicative noise, and also of the more risky stocks in the small capitalization portions of the stock market.

```
clear
p.initial = @(v,p) 1;
p.deriv = @(a,w,p) -0.4*a+a*w;
xspde(p);
```

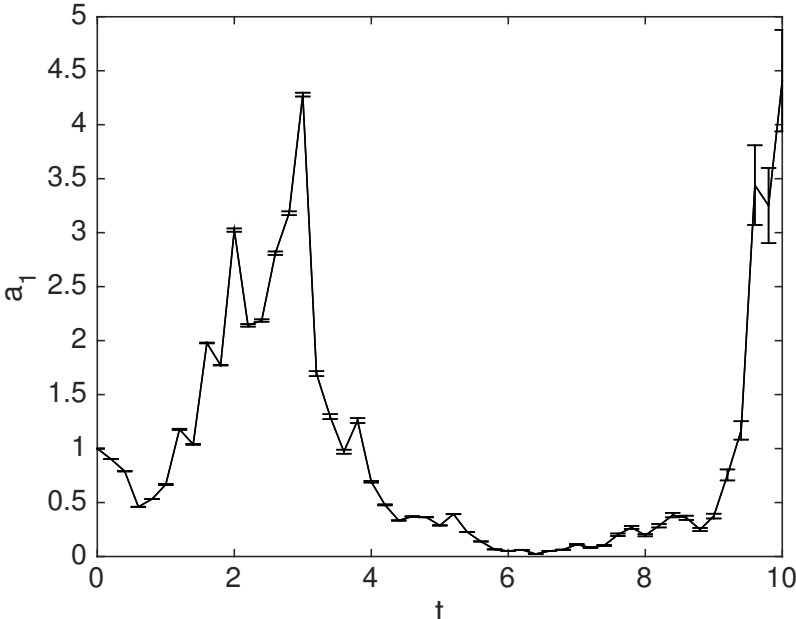

Figure 4: Simulation of the Black-Scholes equation describing stock prices.

Here *p.initial* describes the initialization function. The first argument of $@(v, p)$ is $v$, an initial random variable with unit variance. *The error-bars are estimates of step-size error.* Errors can be reduced by using more time-steps.

To learn more, try the following:

• **Solve for a more mature stock having** $\mu = 0.1$, $\sigma = 0.1$.

### 4.8.5 Nonlinear quantum simulation

This example involves a full nonlinear quantum phase-space simulation using the positive-P representation described in Sec (2.8), in which the two variables are only conjugate in the mean. This allows quantum superpositions of coherent states to be represented, or in fact any state, including squeezed or entangled states in more general cases.

A simple example is the nonlinear driven quantum subharmonic generator - for example, an opto-mechanical, superconducting or nonlinear optical medium in a driven cavity [70–73]. This is derived from the Hamiltonian for a resonant, coupled two-mode nonlinear interferometer, with $\hat{a}_2$ driven externally at twice the frequency of $\hat{a}_1$:

$$\hat{H} = i\hbar \left[ \frac{\kappa}{2} \hat{a}_2 \hat{a}_1^{\dagger 2} + \mathcal{E}_2 \hat{a}_2^{\dagger} - h.c. \right] \tag{125}$$

After including losses in both modes in the positive P-representation, assuming zero temperature reservoirs, and adiabatically eliminating $\alpha_2$ with $\gamma_2 \gg \gamma_1$, one has the following Ito equation:

$$\frac{d\alpha_1}{dt} = -\gamma_1 \alpha_1 + \alpha_1^{\dagger} \frac{\kappa \epsilon_2}{\gamma_2} \left[ 1 - \frac{\kappa}{2\epsilon_2} \alpha_1^2 \right] + \sqrt{\frac{\kappa \epsilon_2}{\gamma_2} - \frac{\kappa^2}{2\gamma_2} \alpha_1^2} w_1(t)$$

$$\frac{d\alpha_1^{\dagger}}{dt} = -\gamma_1 \alpha_1^{\dagger} + \alpha_1 \frac{\kappa \epsilon_2}{\gamma_2} \left[ 1 - \frac{\kappa}{2\epsilon_2} \alpha_1^{\dagger 2} \right] + \sqrt{\frac{\kappa \epsilon_2}{\gamma_2} - \frac{\kappa^2}{2\gamma_2} \alpha_1^{\dagger 2}} w_1(t)$$

Rescaling the fields so that $\alpha_1 = a_1\sqrt{n_c}$, $\alpha_1^\dagger = a_2\sqrt{n_c}$, where $n_c = \frac{2\epsilon_2}{\kappa}$, then rescaling time by letting $\tau = \frac{\kappa\epsilon_2}{\gamma_2}t$, defining $c = \frac{\gamma_1\gamma_2}{\kappa\epsilon_2}$, and using Eq (9) to transform from an Ito to a Stratonovich equation gives:

$$\frac{da_1}{d\tau} = -(c - \frac{1}{2n_c})a_1 + a_2\left[1 - a_1^2\right] + \frac{1}{\sqrt{n_c}}\sqrt{1 - a_1^2}w_1(\tau)$$
$$\frac{da_2}{d\tau} = -(c - \frac{1}{2n_c})a_2 + a_1\left[1 - a_2^2\right] + \frac{1}{\sqrt{n_c}}\sqrt{1 - a_2^2}w_2(\tau),\tag{126}$$

where $w_1, w_2$ are delta-correlated real Gaussian noises.

There is a bistable region, which leads to a discrete time symmetry breaking. The solution in the steady-state is

$$P = \left(1 - a_1^2\right)^{cn_c - 1}\left(1 - a_2^2\right)^{cn_c - 1}e^{2n_c a_1 a_2}\tag{127}$$

The integration manifold is the region of real $a_1$, $a_2$, such that $a_1^2 \leq 1$, $a_2^2 \leq 1$. There are two physically possible metastable values of the amplitudes. The physically observed quantity is the amplitude and number:

$$\langle\hat{a}\rangle = \langle a_1 + a_2\rangle\sqrt{\frac{n_c}{2}}$$
$$\langle\hat{n}\rangle = n_c\langle a_1 a_2\rangle.\tag{128}$$

Parameters that show bistable behavior on reasonable time-scales of $T = 100$ are $c = 0.6$, $n_c = 4$. To learn more, try the following:

- **Simulate the nonlinear oscillator by creating a file, say, $NonlinearQ.m$**

- **Can you observe quantum tunneling in the bistable regime?**

- **Do you see transient Schrodinger 'cat states' with a negative $n = \alpha_1\alpha_2$ value?**

A negative value of $\alpha_1\alpha_2$ is evidence for a quantum superposition! For experimental comparisons, one would measure correlation functions and spectra. These calculations require long time scales, p.ranges, to observe tunneling, and of order 100 time steps per plotted time point, p.steps, to maintain good accuracy in the quantum simulations.

For lower damping and large nonlinearity, other methods should be used, as the stochastic equations can become unstable in this limit.

The model is a simplified version of more recent quantum technologies used to investigate Schrodinger cat formation in superconducting quantum circuits [74], and the CIM machine used to solve NP-hard optimization problems with photonic circuits [75–77], although there are greater complexities in both these cases.

Similar methods can also be used to investigate quantum and chemical non-equilibrium phase transitions [78], tunneling in open systems [79], quantum entanglement [80], Einstein-Podolsky-Rosen paradoxes [81,82], Bell violations [83,84], and many other problems treated in the literature [3,51].

# 5   Solving an SPDE

*This section describes how to simulate a PDE or SPDE, including choosing spectral or finite difference methods and specifying boundary conditions.*

## 5.1   Multidimensional Wiener process

To solve for a single four-dimensional trajectory with three space dimensions, as in Eq (64) , just type in:

```
p.dimensions = 4;
p.deriv = @(a,w,p) w;
xspde(p);
```

Here $p.deriv$ defines the time derivative $\dot{a}$ in the input parameter structure $p$, while $w$ is a delta-correlated Gaussian noise generated internally. Apart from the dimensions, there are no other parameters, so default values are used. This produces the graph shown in Fig (5), which gives a single trajectory.

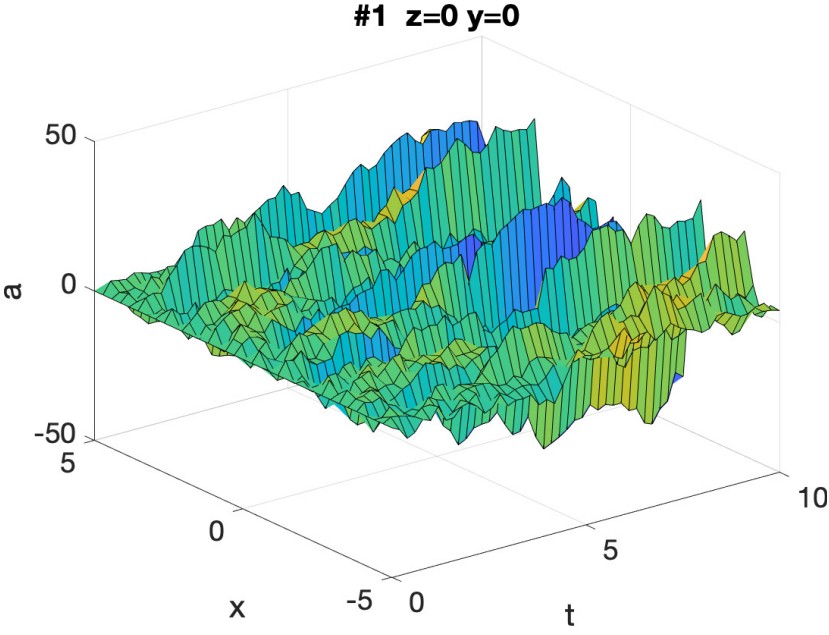

Figure 5: A multidimensional random walk of a three-dimensional field projected onto $y = z = 0$.

For more interesting problems than this, more parameters are needed, as explained next.

## 5.2 SPDE parameters

A stochastic partial differential equation or $SPDE$ for a complex vector field is defined in both time $t$ and space dimension(s) $\mathbf{x}$. The total *dimensions $d$* includes both time and space. To solve a stochastic partial differential equation xSPDE involves a similar procedure to the case of the SDE, covered in section 4.

The numerical solutions require additional parameters to define the spatial grid, and to define the linear transformations in an interaction picture, if spectral methods are used. The SPDE input parameters extend those already introduced in (4.2.1). Some new and extended parameters are listed in the table below:

| Label | Type | Typical value | Description |
|---|---|---|---|
| *dimensions* | integer | 2 | Space-time dimensions |
| *linear* | function | @(p) p.Dx | Linear interaction picture function |
| *ranges* | real vector | $[10,10,...]$ | Ranges in time and space |
| *transforms{n}* | integer vector | $[1,0,1,..]$ | Space-time transform switch |
| *points* | integer vector | $[51,35,..]$ | Output lattice points in [t,x,y,z,..] |
| *origins* | real vector | $[0,-5,..]$ | Space-time integration origin |
| *boundaries{i}* | integer array | $[0,0;0,0]$ | Boundary type per field index |

Setting $dimensions > 1$ defines an (S)PDE as opposed to an ordinary (S)DE. Here the cell index $i$ indicates a field index, and the cell index $n$ gives the observable output or graph index.

In the xSPDE implementation, the total space-time *dimensions* is unlimited, although, large space-time dimensions become memory-intensive and slow. There is a practical limit of about ten space-time dimensions with current digital computers, unless you have a very large, fast computer.

### 5.2.1 SPDE spatial lattice

Stochastic variables in an SPDE are stored in a real or complex array, $a(i,\ell,e)$. Here $i$ is the internal field index, $\ell$ is a $d-1$ dimensional spatial lattice index for $d$ space-time dimensions, and $e$ is the ensemble index. To specify the spatial lattice, one must define:

**dimensions** The dimensionality in time and space. The default is an SDE: $d = 1$.

**points** The number of integration points. The default is $\mathbf{N} = [51, 35, 35..]$.

**ranges** The integration ranges in each dimension. The default is $\mathbf{R} = [10, 10, 10..]$.

**origins** The origins of the space-time integration domains. By default, the origin is $O(1) = 0$ for the time coordinate and $\mathbf{O} = -\mathbf{R}/2$ for the space coordinates ($\mathbf{R}$ is the *ranges* variable) such that the spatial grid is symmetric around $\mathbf{r} = 0$.

### 5.2.2 Initial conditions

Initial conditions are set at the initial time of $t = O_1$ with a user-defined function so that:

$$a(O_1) = initial(v, p) \tag{129}$$

The *initial* function includes initial random fields $v = \left[ v^x, v^k \right]$. Their correlations are either delta correlated or spatially correlated. To allow this, the input parameter $randoms$ is a vector such that:

$randoms(1)$ is the number of delta-correlated random fields, $v^x$, and $randoms(2)$ is the number of correlated random fields, $v^k$. All random fields in the *initial* function, even if correlated using filters in momentum space, are transformed to position space before use. If there is no filtering, $v^x$ and $v^k$ have the same correlations.

## 5.3 Next example

As another very simple example, consider the SPDE

$$\frac{\partial a}{\partial t} = -\frac{1}{4}a + x \cdot w \tag{130}$$

The system has one spatial dimension, or $d = 2$ space-time dimensions, one field and one noise variable. We suppose that the initial noise variance is Gaussian, with:

$$a(0, x) = 10v(x). \tag{131}$$

We want to consider $10,000$ stochastic trajectories per sub-ensemble with $10$ sub-ensembles. We will set the origin for $x$ to 0. The variable $a$ will be initialized as delta-correlated in space with a gaussian standard deviation on the lattice of $\sigma = 10/\sqrt{\Delta V}$. As our observable, we consider the second moment of $a$.

This is simulated through the following xSPDE code:

```
clear;
p.name = 'simple SPDE';
p.dimensions = 2;
p.ensembles = [10000,10];
p.origins = [0,0];
p.noises = 1;
p.initial = @(v,p) 10*v;
p.observe = @(a,~) a.^2;
p.olabels = '<a^2>';
p.deriv = @(a,w,p) -0.25*a + p.x .* w;
xspde(p);
```

With this input, Matlab produces two output graphs:

The second graph shows the time evolution for $x$ at the mid-point, $x = 5$. The variances are larger than they would be in the SDE case, where one might expect an initial variance of $\langle a^2(0) \rangle = 100$. The reason for this is that the initial noise random and noise fields are replaced by a lattice with a variance of $1/\Delta V$. In the default case, this causes an increase in the local noise.

## 5.4 Transverse lattice

In the functions $deriv$, $initial$ and $observe$, the field and noise variables $a$ and $w$ now have extended dimensionality compared to the 1-dimensional case, to index the transverse lattice. The indices are $a(f, \mathbf{i}, e)$, where the:

**field** index $f$ corresponds to the field index for $a$ and the noise index for $w$.

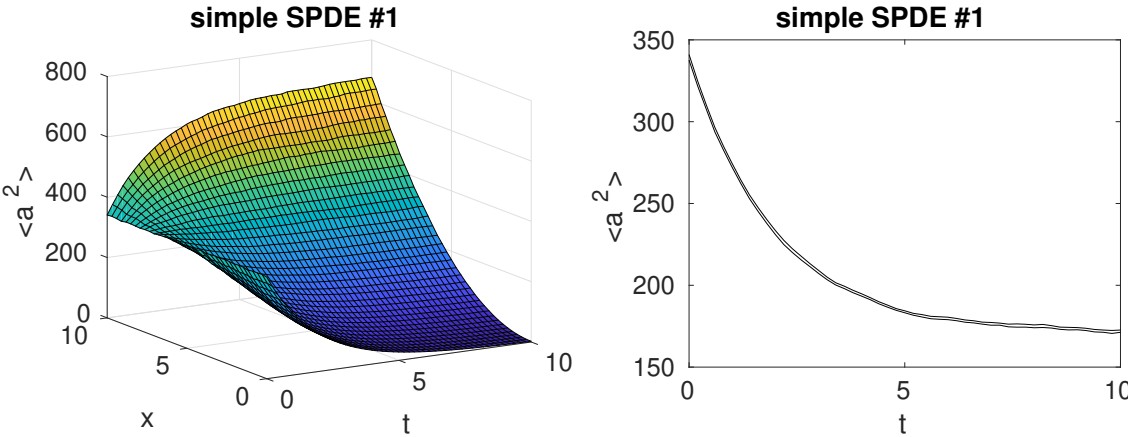

Figure 6: Example: simple SPDE output graphs

**intermediate** indices **i**, which are absent in the 1-dimensional case, correspond to the spatial grid and have the same structure. For example, in the case with *dimensions = 3*, indicating one time index and two spatial dimension, **i** corresponds to the two space indices.

**last** index $e$ corresponds to the stochastic trajectory.

For storing space coordinates like $p.x$, the first and last index are $f = e = 1$. Where Fourier transforms are used internally, the momentum arrays have zero momentum as the first index to follow standard discrete Fourier transform conventions. This is changed to a symmetric convention in all stored graphics data outputs.

As explained in section 3.9, the general equation solved can be written in differential form as

$$\frac{\partial \mathbf{a}}{\partial t} = \mathbf{A}[\mathbf{a}] + \underline{\mathbf{B}}[\mathbf{a}] \cdot \mathbf{w}(t) + \underline{\mathbf{L}}[\nabla, \mathbf{a}] . \tag{132}$$

The linear function $L$ can be input either inside the derivative function using finite difference operators described below, or as a separate *linear* function, to allow for an interaction picture in which case:

$$\underline{\mathbf{L}}[\nabla, \mathbf{a}] = \underline{\mathbf{L}}[\nabla]\mathbf{a} . \tag{133}$$

This depends on momentum space coordinates, which involves Fourier transforms and means that no space dependence is allowed. Spectral methods in xSPDE are currently restricted to cases with linear derivative terms and periodic or zero boundary conditions. It is also possible to use finite differences, in which case the derivative terms are included as part of the derivative function *deriv*.

The usual FFT spectral methods require periodicity. The four other boundary methods can currently only be used with the default boundary values of zero, and with an interaction picture derivative that only has even powers of derivatives. Additional spectral methods will be included in a subsequent release: xSPDE4.

### 5.4.1 Linear operator

The field $x$ is provided by the parameter structure, and corresponds to the variable $x$ in Eq (130). All parameters are preceded by the parameter structure label. Likewise, for higher dimensional

problems, the variables $y$ and $z$ exist. These are placeholders for $r\{1\}, r\{2\}, r\{3\}$, so the spatial variables of even higher dimensional problems can be accessed through $r\{n\}$.

Using a linear operator in an SPDE gives better accuracy, and allows use of the interaction picture. This is included automatically for all built-in xSPDE algorithms, provided the *linear* function is defined in the parameter structure. Variables $p.D\{i\}$ (with placeholders $p.Dx, p.Dy, p.Dz$ for the first 3 spatial dimensions) provide access to the derivative operator. Higher-order derivatives are found through potentiating $p.Dx$ accordingly.

For example, the 2-dimensional Laplacian operator

$$\nabla^2 = \frac{\partial^2}{\partial x^2} + \frac{\partial^2}{\partial y^2} \tag{134}$$

corresponds to a linear differential operator specified as:

$$p.linear = @(p) \ p.Dx.^2 + p.Dy^2; \tag{135}$$

For a comprehensive list of variables accessible through the $p$-structure, refer to sec. 9.8.

### 5.4.2 Integrals and averages

There are functions available in xSPDE for spatial grid averages and integrals, to handle the spatial grid. These are **Ave** and **Int,** which are used to calculate observables for plotting. They operate in parallel over the lattice dimensions, by taking a vector or scalar quantity, for example a single field component, and returning an average or a space integral. In each case the first argument is the field, the second argument is a vector defining the type of operation, and the last argument is the parameter structure. If there are two arguments, the operation vector is replaced by its default value.

Integrals over the spatial grid allow calculation of global quantities. To take an integral over the spatial grid, use the xSPDE function *Int* with arguments *(o, [dx, ] p)*.

This function takes a scalar or vector quantity $o$, and returns a trapezoidal space integral over selected dimensions with vector measure $dx$. If $dx(j) > 0$ an integral is taken over dimension $j$. Dimensions are labelled from $j = 1,2,3$ ... as in all xSPDE standards. Time integrals are ignored at present. Integrals are returned at all lattice locations. To integrate over an entire lattice, set $dx = p.dx$, otherwise set $dx(j) = p.dx(j)$ for selected dimensions $j$.

If momentum-space integrals are needed, first use the *transforms* switch to make sure that the field is Fourier transformed before being averaged, and input $dk$ instead of $dx$.

Spatial grid averages can be used to obtain stochastic results with reduced sampling errors if the overall grid is homogeneous. An average is carried out using the builtin xSPDE function *Ave()* with arguments *(o, [av, ] p)*.

This takes a vector or scalar field or observable, defined on the lattice, and returns an average over the spatial lattice. The input is a field $a$ or observable $o$, and an optional averaging switch $av$. If $av(j) > 0$, an average is taken over dimension $j$. Space dimensions are labelled from $j = 2,3...$ as elsewhere. If the $av$ vector is omitted, the average is taken over all space directions.

### 5.4.3 One space-dimensional example

A famous partial differential equation is an exactly soluble equation for a soliton, the nonlinear Schrödinger equation (NLSE):

$$\frac{da}{dt} = \frac{i}{2}\left[\nabla^2 a - a\right] + ia\,|a|^2\,. \tag{136}$$

Together with the initial condition that $a(0, x) = sech(x)$, this has a soliton, an exact solution that doesn't change in time:

$$a(t, x) = sech(x). \tag{137}$$

The spatial integral is simply:

$$\int sech(x)dx = \pi. \tag{138}$$

An xSPDE code that solves this is given below, together with code that compares the numerical solution with the exact solutions for the soliton and the integral:

```
p.name = 'NLS soliton';
p.dimensions = 2;
p.initial = @(v,p) sech(p.x);
p.deriv = @(a,~,p) 1i*a.*(conj(a).*a);
p.linear = @(p) 0.5*1i*(p.Dx.^2-1.0);
p.olabels = {'a(x)','\int a(x) dx'};
p.observe{2} = @(a,p) Int(a, p);
p.compare{1} = @(p) sech(p.x);
p.compare{2} = @(p) pi;
e = xspde(p);
```

Due to finite boundaries and discrete spatial lattice, the agreement is not perfect. The errors can be reduced by increasing the range of the integration domain and improving the resolution with more points.

### 5.4.4 Two space-dimensional example

As another example, consider the two-dimensional nonlinear stochastic equation, with periodic boundary conditions:

$$\frac{\partial a}{\partial t} = \nabla^2 a(\mathbf{x}, t) + a(\mathbf{x}, t) - a(\mathbf{x}, t)^3 + \eta(\mathbf{x}, t). \tag{139}$$

Using the interaction picture allows for the absorption of both the Laplacian and the first-order term by the *p.linear* parameter, which results in

```
...
p.linear = @(p) (p.Dx.^2+p.Dy.^2) + 1;
p.deriv = @(a,w,~) -a.^3 + w;
xspde(p);
```

With this input, Matlab produces two output graphs:

## 5.5 Finite differences

Instead of using the interaction picture, xSPDE also has finite difference methods for direct differentiation. These derivatives are obtained through function calls $D1$ and $D2$ respectively for first and second derivatives, which use a fixed grid spacing. As elsewhere, they can be replaced by user-written functions if preferred. Generally they require smaller steps in time than spectral methods, when used to define the derivative.

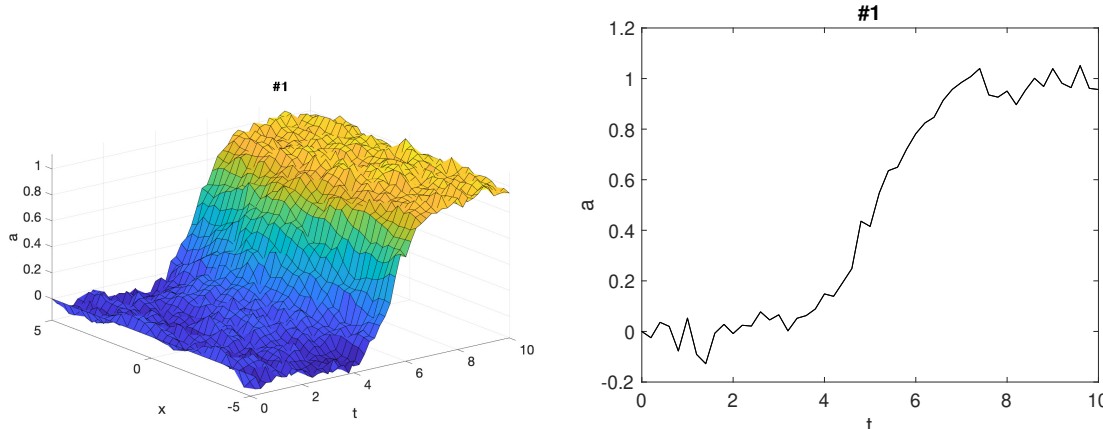

Figure 7: Two space-dimensional example graphs

### 5.5.1 Finite difference first derivatives

The code to take a first order spatial derivative with finite difference methods is carried out using the xSPDE function *D1()* with arguments *(o, [dir, ] p)*.

This takes a scalar or vector *o* and returns a first derivative in an axis direction *dir*. Set *dir = 2* for an x-derivative, *dir = 3* for a y-derivative, and so on. Time derivatives are ignored at present. Derivatives are returned at all lattice locations.

If the direction is omitted, an *x*-derivative is returned. These derivatives can be used both in calculating propagation and in calculating observables. The boundary condition is set by the *boundaries* input. It can be made periodic, which is the default, or Neumann with zero derivative, or Dirichlet with zero field.

### 5.5.2 Finite difference second derivatives

The code to take a second order spatial derivative with finite difference methods is carried out using the xSPDE *D2* function with arguments *(o, [dir, ] p)*.

This takes a scalar or vector *o* and returns the second derivative in axis direction *dir*. Set *dir = 2* for an x-derivative, *dir = 3* for a y-derivative and so on. All other properties are exactly the same as *D1*.

Without using the interaction picture, the stochastic equation of Eq (139) is specified in xSPDE using finite differences as

```
p.dimensions = 3;
p.steps = 50;
p.deriv = @(a,w,p) D2(a,2,p)+D2(a,3,p)+a - a.^3 +
w/10;
xspde(p);
```

This gives the same result as with the linear propagator, although requiring smaller step-sizes for numerical stability, with an output graph shown in Fig (8). Note that the parameters and noises are slightly different!

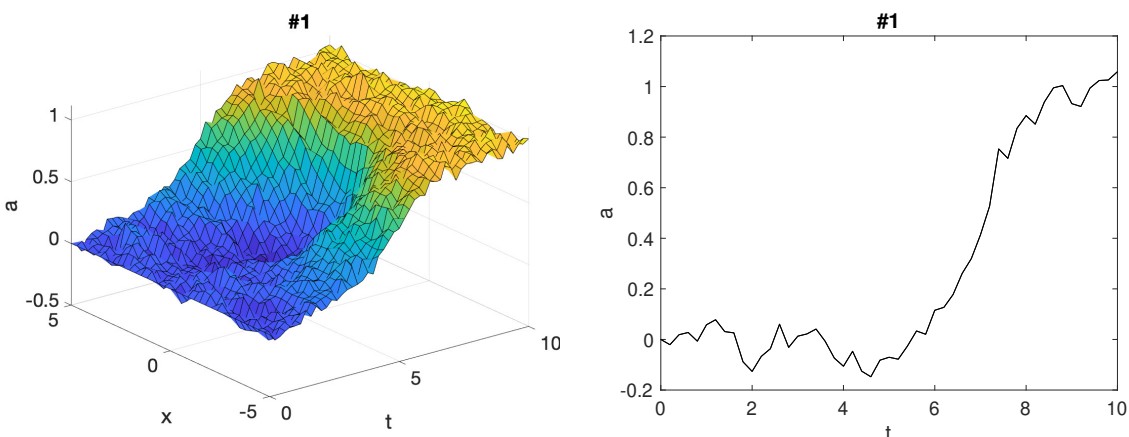

Figure 8: Two space-dimensional example graphs, direct differentiation.

## 5.6 Boundary conditions

### 5.6.1 Transverse boundaries

Transverse boundary conditions must be given for all partial differential equations. Common transverse *boundary types* are of three types: Neumann (specified derivative), periodic, or Dirichlet (specified field). These are obtained using $boundaries\{d\} = -1, 0, 1$, which is specified for each space dimension $d > 1$, field index and boundary.

If *boundaries* are omitted for any dimension the default is 0, which gives periodic boundaries in that dimension for all field indices, and permits the use of Fourier transforms and an interaction picture as described above.

The value of $boundaries\{d\}$ is a matrix whose column index ($i$) is the field index, and whose row index (j) is given by $j = 1, 2$ for the lower and upper boundary type respectively.

Spatial derivatives or other functions linking different spatial points can be specified either in the functionals $A[\mathbf{a}, \mathbf{r}]$, $\underline{B}[\mathbf{a}, \mathbf{r}]$ or else in the *linear* function, provided the derivative terms are linear functions of the fields. Use of the *linear* function allows an interaction picture algorithm, with increased efficiency. The *linear* function is currently only available with periodic boundary conditions.

The default boundary conditions are periodic. The implicit setting of this is that periodicity is enforced such that $a(o_i - dx_i/2) = a(o_i + r_i + dx_i/2)$, which is the usual discrete Fourier transform requirement.

Otherwise, the differential equation boundaries are specified at $a(o_i)$, $a(o_i + r_i)$, using the cell-array input $boundaries\{d\}(i, j)$, which is defined per space dimension ($d = 2, 3..$), field index ($i = 1, 2..$) and boundary $j = (1, 2)$. Here $d > 1$ is the transverse dimension, not including time, which only has an initial condition.

In summary the available boundary types are:

**Neumann:** For specified *derivative* boundaries, $boundaries\{d\}(i, j) = -1$

**Periodic:** For *periodic* boundaries, $boundaries\{d\}(i, j) = 0$

**Dirichlet:** For specified *field* boundaries, $boundaries\{d\}(i, j) = 1$

These are specified in a cell array: $boundaries\{d\}(i, 1)$ sets the lower boundary type in dimension $d$, for the $i$-th field component while $boundaries\{d\}(i, 2)$ gives the upper boundary type. Each space dimension, variable and boundary is set independently. In xSPDE, the equations are always initial value problems in time, so the time dimension boundary specification for $d = 1$ is not included.

**Example: boundary types in a 2-dimensional PDE** Suppose there are two fields, and we wish to set mixed boundaries in space, with Dirichlet in the past and Neumann in the future for the first field $a(1, :)$, with the opposite combination in the second field component, $a(2, :)$:

```
p.boundaries{2} = [1,-1;-1,1];
```

### 5.6.2 Transverse boundary values

For non-vanishing, specified boundary conditions, the boundary function $boundfun(a, d, p)$ is called. This returns the boundary values used for the fields or derivatives in a particular dimension $d > 1$ as an array of dimension $b(\mathbf{j}, e))$, where $\mathbf{j} = i, \mathbf{k}$.

Here $i = j_1$ is the field index, and $\mathbf{k}$ is the space index, where $j_d$ is the index of the dimension whose boundary values are specified. For this dimension, only two values are needed: $j_d = 1, 2$ for the lower and upper boundary values, which could either be field values or their derivatives. An ensemble index $e$ is also needed if the boundary values are stochastic.

Boundary values can be a function of both the fields ($a$) and internal variables like the current time ($t$). These may have stochastic initial values at $t = 0$ which are calculated only once. In such cases the boundary values must first be initialized, so the routine $boundfun(a, d, p)$ is first internally initialized with time $t < origin(1)$, and with random Gaussian values in the input field $a$. These are delta-correlated in space, i.e., with the same definition as "inrandoms". The xSDPE program stores the returned values $b$ for the boundaries in an internal cell array, $boundval\{d\}$, for later use if required.

The default boundary value is zero, set by the default boundary function $xboundfun(a, d, p)$.

**NOTE: Current xSPDE code requires finite-difference methods to be used with $boundfun$. Spectral methods use the default boundary conditions.**

### 5.6.3 Example: boundaries in a 2-dimensional PDE

Suppose there are two fields, and we wish to set boundary values.

We take boundary values as Dirichlet for $x = 0$ and Neumann for $x = 1$ in field variable 1, and Neumann for $x = 0$ and Dirichlet for $x = 1$ in field variable 2, that are different from the default values of $a = 0$, $\partial_x a = 0$, so that:

$$
\begin{aligned}
a_1(x = 0) &= 1, \\
\partial_x a_1(x = 1) &= a_1(x = 1). \\
\partial_x a_2(x = 0) &= -a_2(x = 0) \\
a_2(x = 1) &= -1.
\end{aligned}
\tag{140}
$$

These are set in the following code:

```
p.boundfun = @mybfun
p.boundaries{2} = [1,-1;-1,1];
...
function b = mybfun(a,d,p)
% b = mybfun(a,d,p) calculates boundary values
b(1,2,:)  = a(1,end,:);
b(2,1,:)  = -a(2,end,:);
b(1,1,:)  = 1;
b(2,2,:)  = -1;
end
```

### 5.6.4 Transverse plots

A number of plots at equally spaced points in time can be generated through $p.transverse$ (see section 10). For example, adding the line below creates 3 time-sliced plots at $t = 0, 5, 10$:

```
p.transverse{1} = 3;
```

## 5.7 Output transforms

For graphical output, Fourier transforms involve a sum over the lattice points using a discrete Fourier transform at the lattice points $x_i$, so that:

$$\tilde{a}(\omega_i, \mathbf{k}_i) = \frac{dt d\mathbf{x}}{[2\pi]^{d/2}} \sum_{j_1 \ldots j_d} \exp\left[i\left(\omega_{i_1} t_{j_1} - \mathbf{k_i} \cdot \mathbf{x_j}\right)\right] a(t_{j_1}, \mathbf{x_j}) \tag{141}$$

The momenta $k_i$ have an interval of

$$dk_i = \frac{2\pi}{n_i dx_i} \tag{142}$$

with $k_i$ values given for even n by:

$$k_i = \left(1 - \frac{n_i}{2}\right) dk_i, \ldots \frac{n_i}{2} dk_i \tag{143}$$

and for odd n by:

$$k_i = \frac{1 - n_i}{2} dk_i, \ldots \frac{n_i - 1}{2} dk_i \tag{144}$$

Once Fourier transformed, the $observe$ function can be used to take any further functions or combinations of Fourier transformed fields prior to averaging. Important points to keep in mind are as follows:

- Fourier transforms are specified for the k-th $observe$ function independently of all other functions, by specifying $transforms\{k\} = \left[\ell_{1}, \ldots \ell_{d,}\right]$.

- Here $\ell_j = 0, 1$ is a logical switch, set to to $\ell_j = 1$ if the $j - th$ dimension requires a Fourier transform, and $\ell_j = 0$ if there is no Fourier transform.

- The internal fields $p.k\{1\}, \ldots p.k\{d\}$ are available for use in making functions of momentum for use with observations.

- In propagation calculations, the momentum lattice values start with $k = 0, \ldots$, following standard Matlab and FFT conventions.

- For storing and graphing, momentum lattice values are reordered to start with $k = -k_{max}, \ldots$, following standard graphics and mathematical conventions.

## 5.8  Initial random fields

Fourier transforms are available for use both on initial random values and on noise fields during time-evolution. This is controlled by the second element of *randoms* and *noises*, respectively.

When $randoms(1) > 0$, an initial random field $\mathbf{v}^x$ is generated with delta-correlations in $x$-space. When $randoms(2) > 0$, an initial random field $\tilde{\mathbf{v}}^k$ is generated with delta-correlations in $k$-space. This can be filtered with a user-specified filter function to give $\tilde{\mathbf{v}}^{kf}$, then inverse Fourier transformed to give $v^k$. Both random fields are passed to the *initial* function as an extended vector $\left[ v^x, v^k \right]$, for field *initialization* in space.

There is a user specified filter function available, to modify random fields $\tilde{v}^k$, that are delta-correlated in momentum space using a filter function, '*rfilter*' so that $v_i^{kf}(\mathbf{k}) = f_i^{(r)}\left(\mathbf{v}^k(\mathbf{k})\right)$, before being used. The corresponding correlations are:

$$
\begin{aligned}
\left\langle v_i^x(\mathbf{x}) v_j^x(\mathbf{x}') \right\rangle &= \delta\left(\mathbf{x} - \mathbf{x}'\right) \delta_{ij} \sim \frac{1}{\Delta V} \delta_{\mathbf{x},\mathbf{x}'} \delta_{ij} \\
\left\langle \tilde{v}_i^k(\mathbf{k}) \tilde{v}_j^k(\mathbf{k}') \right\rangle &= \delta\left(\mathbf{k} - \mathbf{k}'\right) \delta_{ij} \sim \frac{1}{\Delta K} \delta_{\mathbf{k},\mathbf{k}'} \delta_{ij} \\
\left\langle \tilde{v}_i^{kf}(\mathbf{k}) \tilde{v}_j^{kf}(\mathbf{k}') \right\rangle &= \left\langle f_i^{(r)}\left(\tilde{\mathbf{v}}^k(\mathbf{k})\right) f_j^{(r)}\left(\tilde{\mathbf{v}}^k(\mathbf{k}')\right) \right\rangle.
\end{aligned}
\tag{145}
$$

Note that on a lattice, we replace the Dirac continuous delta-function by a discrete Kronecker delta function scaled by an inverse volume element either in space ($\Delta V$) or momentum ($\Delta K$). The xSPDE Fourier transforms are given by a symmetric Fourier transform, so that if we inverse Fourier-transform the $k-$space *inrandoms*, without filtering, then:

$$
v^k(\mathbf{x}) = \frac{1}{[2\pi]^{(d-1)/2}} \int e^{i\mathbf{k}\cdot\mathbf{x}} \tilde{v}^k(\mathbf{k}) d\mathbf{k}
\tag{146}
$$

These have random initial values that are real and delta-correlated in space, so that:

$$
\left\langle v^x(\mathbf{x}) v^x(\mathbf{x}') \right\rangle = \delta\left(\mathbf{x} - \mathbf{x}'\right).
\tag{147}
$$

The corresponding noises in position space are correlated according to:

$$
\begin{aligned}
\left\langle v^k(\mathbf{x}) \left(v^k(\mathbf{x}')\right)^* \right\rangle &= \frac{1}{[2\pi]^{(d-1)}} \int e^{i(\mathbf{k}\cdot\mathbf{x} - \mathbf{k}'\cdot\mathbf{x}')} \left\langle \tilde{v}^k(\mathbf{k}) \tilde{v}^k(\mathbf{k}') \right\rangle d\mathbf{k} d\mathbf{k}' \\
&= \frac{1}{[2\pi]^{(d-1)}} \int e^{i(\mathbf{x} - \mathbf{x}')\cdot\mathbf{k}} d\mathbf{k} \\
&= \delta\left(\mathbf{x} - \mathbf{x}'\right).
\end{aligned}
\tag{148}
$$

Similarly, if we don't conjugate the k-noise, then:

$$
\left\langle v^k(\mathbf{x}) v^k(\mathbf{x}') \right\rangle = \delta\left(\mathbf{x} + \mathbf{x}'\right).
\tag{149}
$$

However, if we define $\tilde{v}^c(\mathbf{k}) = \left[\tilde{v}_1^k(\mathbf{k}) + i\tilde{v}_2^k(\mathbf{k})\right]/\sqrt{2}$, then we obtain complex noise that is only delta correlated when conjugated.

$$\left\langle v^c(\mathbf{x})\left(v^c(\mathbf{x}')\right)^*\right\rangle = \delta(\mathbf{x}-\mathbf{x}')$$
$$\left\langle v^c(\mathbf{x})v^c(\mathbf{x}')\right\rangle = 0. \tag{150}$$

This is obtainable with the x-space noise as well, but the utility of the k-space noise is that it can be filtered to have nonlocal correlations in space if required.

During propagation in time, $\mathbf{w} = \left[\mathbf{w}^x, \mathbf{w}^k\right]$ are real noise fields that are delta-correlated in space-time. They are calculated in an analogies way, except with an additional factor of $1/\sqrt{dt}$ because they are delta correlated in time as well. There is a user specified scaling function available, to take random noises $w^k$ in momentum space that are then scaled using a filter function, 'nfilter' so that $w_i^{kf}(\mathbf{k}) = f_i^{(n)}(\mathbf{w}^k(\mathbf{k}))$, before being used:

$$\left\langle w_i^x(t,\mathbf{x})w_j^x(t,\mathbf{x}')\right\rangle = \delta(\mathbf{x}-\mathbf{x}')\delta(t-t')\delta_{ij}$$
$$\left\langle \tilde{w}_i^k(t,\mathbf{k})\tilde{w}_j^k(t,\mathbf{k}')\right\rangle = \delta(\mathbf{k}-\mathbf{k}')\delta(t-t')\delta_{ij}$$
$$\left\langle \tilde{w}_i^{kf}(t,\mathbf{k})\tilde{w}_j^{kf}(t',\mathbf{k}')\right\rangle = \left\langle f_i^{(n)}(\tilde{\mathbf{w}}^k(t,\mathbf{k}))f_j^{(n)}(\tilde{\mathbf{w}}^k(t',\mathbf{k}'))\right\rangle. \tag{151}$$

## 5.9 Examples

### 5.9.1 Stochastic Ginzburg-Landau

Including two space dimensions, or space-time dimensions of $d = 3$, an example of a SPDE is the stochastic Ginzburg-Landau equation. This describes symmetry breaking. The system develops a spontaneous phase which varies spatially as well. The model is used to describe lasers, magnetism, superconductivity, superfluidity and particle physics:

$$\dot{a} = \left(1-|a|^2\right)a + bw(t) + c\nabla^2 a \tag{152}$$

where

$$\left\langle w(x)w^*(x')\right\rangle = 2\delta(t-t')\delta(x-x'). \tag{153}$$

The following new ideas are introduced for this problem:

1. `dimensions` **is the space-time dimension.**

2. **The** 'dot' **notation used for parallel operations over lattices**.

3. `linear` **is the linear operator - a Laplacian in these cases.**

4. `images` **produces movie-style images at discrete time slices.**

5. `Dx` **indicates a derivative operation,** $\partial/\partial x$**.**

6. $-5 < x < 5$ **is the default xSPDE coordinate range in space.**

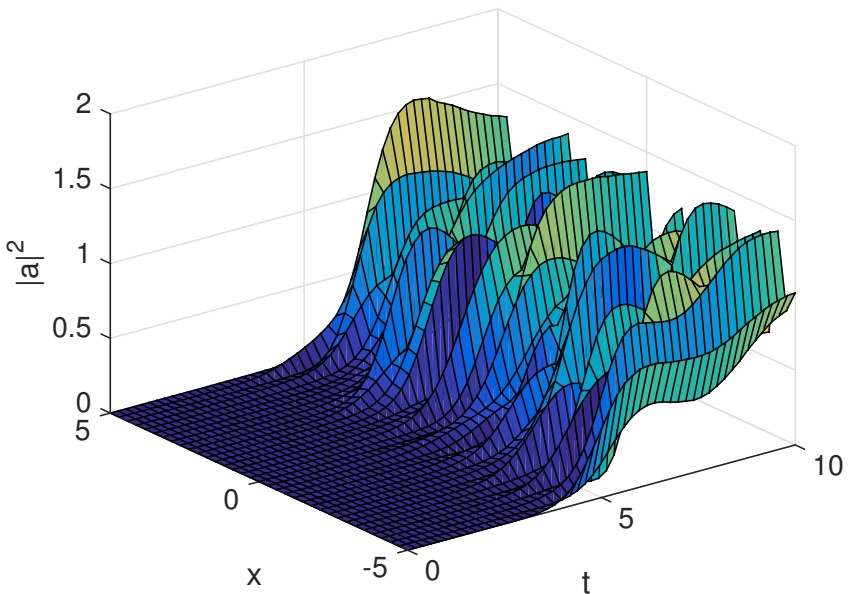

Figure 9: Simulation of the stochastic equation describing symmetry breaking in two dimensions. Spatial fluctuations are caused by the different phase-domains that interfere. The graph obtained here is projected onto the $y = 0$ plane.

**Exercises**

1. **Solve the stochastic G-L equation for $b = 0.001$ and $c = 0.01i$.**

2. **Change to a real diffusion so that $c = 0.1$.**

In the first case, you should get the output graphed in Fig (9) .

```
clear;
p.name = 'Extended laser gain equation';
p.noises = 2;
p.dimensions = 3;
p.steps = 10;
p.linear = @(p) 1i*0.01*(p.Dx.^2+p.Dy.^2);
p.observe = @(a,~) abs(a).^2;
p.images = 6;
p.olabels = '|a|^2';
p.deriv = @(a,w,~) (1-abs(a(1,:).^2)).*a(1,:)+...
                0.001*(w(1,:)+1i*w(2,:));
xspde(p)
```

Here the notation $a(1,:)$ means that the operation is repeated over all values of the subsequent indices, which are the two spatial lattice indices in this case.

### 5.9.2   NLS soliton

The famous nonlinear Schrödinger equation (NLSE) is:

$$\frac{da}{dt} = \frac{i}{2}\left[\nabla^2 a - a\right] + ia\,|a|^2 . \tag{154}$$

Together with the initial condition that $a(0,x) = sech(x)$, this has a soliton [85], an exact solution that doesn't change in time:

$$a(t,x) \;\;=\;\; sech(x). \tag{155}$$

The Fourier transform at $k = 0$ is simply:

$$\tilde{a}(t,0) \;\;=\;\; \frac{1}{\sqrt{2\pi}}\int sech(x)dx = \sqrt{\frac{\pi}{2}}. \tag{156}$$

**Exercises**

- **Solve the NLSE for a soliton using a function instead of a script, then include an additive complex noise of** $0.01(w_1 + iw_2)$ **to the differential equation, and plot again with an average over** $1000$ **samples.**

### 5.9.3 Planar noise

The next example is growth of thermal noise of a two-component complex field in a plane, given by the equation

$$\frac{d\mathbf{a}}{dt} = \frac{i}{2}\nabla^2\mathbf{a} + \mathbf{w}(t, x). \tag{157}$$

where $\zeta$ is a delta-correlated complex noise vector field:

$$w_j(t, \mathbf{x}) = \left[ w_j^{re}(t, \mathbf{x}) + i\zeta_j^{im}(t, \mathbf{x}) \right]/\sqrt{2}, \tag{158}$$

with the initial condition that the initial noise is delta-correlated in position space

$$a(0, \mathbf{x}) = \zeta^{(in)}(\mathbf{x}) \tag{159}$$

where:

$$\zeta^{(in)}(\mathbf{x}) = \left[ \zeta^{re(in)}(\mathbf{x}) + i\zeta^{im(in)}(\mathbf{x}) \right]/\sqrt{2} \tag{160}$$

This has an exact solution for the noise intensity in either ordinary space or momentum space:

$$
\begin{aligned}
\left\langle \left| a_j(t, \mathbf{x}) \right|^2 \right\rangle &= (1+t)/dV \\
\left\langle \left| \tilde{a}_j(t, \mathbf{k}) \right|^2 \right\rangle &= (1+t)/dV_k \\
\left\langle \tilde{a}_1(t, \mathbf{k})\tilde{a}_2^*(t, \mathbf{k}) \right\rangle &= 0.
\end{aligned}
\tag{161}
$$

Here, the noise is delta-correlated, and $dV$, $dV_k$ are the cartesian space and momentum space lattice cell volumes, respectively. Suppose that $n = n_x n_y$ is the total number of spatial points, and there are $n_{x(y)}$ points in the x(y)-direction, so then:

$$
\begin{aligned}
dV &= dxdy \\
dV_k &= dk_x dk_y = \frac{(2\pi)^2}{ndV}.
\end{aligned}
\tag{162}
$$

In the simulations, two planar noise fields are propagated, one using delta-correlated noise, the other with noise transformed to momentum space to allow filtering. This allows use of finite correlation lengths when needed, by including a frequency filter function that is used to multiply the noise in Fourier-space. The Fourier-space noise variance is the square of the filter function.

The first noise index, $p.noises(1)$, indicates how many noise fields are generated, while $p.noises(2)$ indicates how many of these are spatially correlated, via Fourier transform, filter and inverse Fourier transform. These appear to the user as additional noises, so the total is $p.noises(1)+p.noises(2)$. The filtered noises have a finite correlation length. They are correlated with the first $p.noises(1)$ x-space noises they are generated from, as this can be useful.

**Exercises**

- **Solve the planar noise growth equation**

```
function [e] = Planar()
p.name = 'Planar noise growth';
p.dimensions = 3;
p.fields = 2;
p.ranges = [1,5,5];
p.steps = 2;
p.noises = [2,2];
p.ensembles = [10,4,4];
p.initial = @Initial;
p.deriv = @Da;
p.linear = @Linear;
p.observe = @(a,p) a(1,:).*conj(a(1,:));
p.olabels = '<|a_1(x)|^2>';
p.compare = @(p) [1+p.t]/p.dv;
p.images = 4;
e = xspde(p);
end

function a0 = Initial(v,p)
a0(1,:)  = (v(1,:)+1i*v(2,:))/sqrt(2);
a0(2,:)  = (v(3,:)+1i*v(4,:))/sqrt(2);
end

function da = Da(a,w,p)
da(1,:)  = (w(1,:)+1i*w(2,:))/sqrt(2);
da(2,:)  = (w(3,:)+1i*w(4,:))/sqrt(2);
end

function L = Linear(p)
lap = p.Dx.^2+p.Dy.^2;
L(1,:)  = 1i*0.5*lap(:);
L(2,:)  = 1i*0.5*lap(:);
end
```

- **Add a decay rate of $-a$ to the differential equation, then plot again**

- **Add growth and nonlinear saturation terms**

### 5.9.4   Gross-Pitaevskii equation

The next example is a stochastic Gross-Pitaevskii (GP) equation [86] in two dimensions,

$$\frac{da}{dt} = \frac{i}{2}\nabla^2 a - ia(V(r) - i\kappa(r) + |a|^2) + \epsilon\eta \tag{163}$$

where $\eta$ is a correlated complex noise vector field:

$$\eta(t,\mathbf{x}) = w_1(t,\mathbf{x}) + iw_2(t,\mathbf{x}), \tag{164}$$

with the initial condition that the initial random field and the noise are both filtered in momentum space

$$a(0, \mathbf{x}) = a_0(\mathbf{x}) + \epsilon \zeta^{(in)}(\mathbf{x}) \tag{165}$$

where:

$$\zeta^{(in)}(\mathbf{x}) = v_1(\mathbf{x}) + i v_2(\mathbf{x}) \tag{166}$$

We add a Gaussian filter in momentum space for both the initial random field and noise so that, if $\tilde{w}(\mathbf{k})$ is a delta-correlated noise in momentum space:

$$w(\mathbf{k}) = \tilde{w}(\mathbf{k}) \exp\left(-|\mathbf{k}|^2\right)$$
$$v(\mathbf{k}) = \tilde{v}(\mathbf{k}) \exp\left(-|\mathbf{k}|^2\right) \tag{167}$$

This allows use of finite correlation lengths when needed, by including a frequency filter function that is used to multiply the noise in Fourier-space. The Fourier-space noise variance is the square of the filter function.

The first noise index, $p.noises(1)$, indicates how many noise fields are generated that are delta-correlated in $x$, while $p.noises(2)$ indicates how many of these are spatially correlated, via Fourier transform, filter and inverse Fourier transform. These appear to the user as additional noises, so the total is $p.noises(1) + p.noises(2)$. The filtered noises have a finite correlation length.

**Exercises**

- **Solve the stochastic GP equation** (163)**, with a noise coefficient of** $b = 0.1$, $V = 0.01 |\mathbf{x}|^2$, $\kappa = 0.001 |\mathbf{x}|^4$**, and a stored output data file.**

```
function [e] = GPE()
p.name = 'GPE';
p.dimensions = 3;
p.points = [101,64,64];
p.ranges = [1,20,20];
p.noises = [0,2];
p.rfilter = @(w,p) w.*exp(-p.kx.^2-p.ky.^2);
p.nfilter = @(v,p) v.*exp(-p.kx.^2-p.ky.^2);
b = @(xi) .1*(xi(1,:,:)+1i*xi(2,:,:));
p.initial = @(v,p) (p.x+1i*p.y)./(1+10*(p.x.^2 +
p.y.^2))+b(v);
V = @(p) 0.01*(p.x.^2 + p.y.^2)-0.001*1i*(p.x.^2 +
p.y.^2).^2;
p.deriv = @(a,w,p) -1i*a.*(V(p)+conj(a).*a)+b(w);
p.linear = @(p) 0.5*1i*(p.Dx.^2+p.Dy.^2);
p.observe{1} = @(a,p) a.*conj(a);
p.images = {2};
p.imagetype = {2};
p.olabels = {'|a|^2'};
p.file = 'GPE.mat';
e = xsim(p);
xgraph(p.file,p);
end
```

### 5.9.5 Characteristic equation

The next example is the characteristic equation for a traveling wave at constant velocity [87]. It is included to illustrate what happens at periodic boundaries, when Fourier-transform methods are used for propagation. There are a number of methods known to prevent this effect, including addition of absorbers - called apodization - at the boundaries. The equation is:

$$\frac{da}{dt} + \frac{da}{dx} = 0. \tag{168}$$

Together with the initial condition that $a(0,x) = sech(2x+5)$, this has an exact solution that propagates at a constant velocity:

$$a(t,x) = sech(2(x-t)+5). \tag{169}$$

The time evolution at $x = 0$ is simply:

$$a(t,0) = sech(2(t-5/2)). \tag{170}$$

**Exercises**

- **Solve the characteristic equation given above, noting the effects of periodic boundaries.**

```
function [e] = Characteristic()
p.name = 'Characteristic';
p.dimensions = 2;
p.initial = @(v,p) sech(2.*(p.x+2.5));
p.deriv = @(a,z,p) 0*a;
p.linear = @(p) -p.Dx;
p.olabels = {'a_1(x)'};
p.compare = @(p) sech(2.*(p.t-2.5));
e = xspde(p);
end
```

- **Recalculate with the opposite velocity, and a new exact solution.**

### 5.9.6 Nonlinear Anderson localization

A random potential *prevents* normal wave-packet spreading in quantum-mechanics. This is Anderson localization [88]: a famous property of quantum mechanics in a random potential. A typical experimental method is to confine an ultra-cold Bose-Einstein condensate (BEC) in a trap, then release the BEC in a random external potential produced by a laser [89]. The expansion rate of the BEC is reduced by the Anderson localization due to the random potential. Physically, the observable quantity is the particle density $n = |\psi|^2$, but there is a complication, which is that there are nonlinearities from atomic scattering [90].

This can be treated either using a Schrodinger equation with a random potential, at low density, or using the Gross-Pitaevskii (GP) equation to include atom-atom interactions at the mean field level. In this example of a problem where strong localization occurs, the general equations are:

$$\frac{\partial \psi}{\partial t} = \frac{1}{i\hbar} \left[ -\frac{\hbar^2}{2m} \nabla^2 + V(\mathbf{r}) + g |\psi|^2 \right] \psi. \tag{171}$$

In calculations, it is best to use a dimensionless form by rescaling coordinates and fields. A simple way to simulate this with xSPDE is to treat $\psi$ as a scaled field $a(1)$, and to assume the random potential field $V(\mathbf{r})$ as caused by interactions with second random field $|a(2)|^2$. This has the advantage that it is similar to the actual experiment and allows one to treat time-dependent potentials as well, if desired.

With the rescaling, this simplifies to:

$$\frac{\partial a_1}{\partial \tau} = i\left[\frac{\partial}{\partial \zeta^2}^2 - |a_2|^2 - |a_1|^2\right] a_1. \tag{172}$$

A convenient initial condition is to use:

$$
\begin{aligned}
a_1 &= a_0 \exp(-\zeta^2)\\
\left\langle a_2(\zeta) a_2(\zeta')\right\rangle &= \nu \delta\left(\zeta - \zeta'\right).
\end{aligned}
\tag{173}
$$

**Exercise**

- **Solve Schrodinger's equation without a random potential, to observe expansion.**

- **Include a random potential $\nu$, to observe localization.**

- **Experiment with nonlinear terms and higher dimensions.**

Note that the GP equation is a mean field approximation; this is still not a full solution of the many-body problem! Also, the experiments are somewhat more complicated than this, and actually observe the momentum distribution.

# 6   xSIM and xGRAPH

*This section describes how to use xSPDE to run in a batch mode, as well as details of data storage and methods for graphing scanned parameters.*

## 6.1   Batch job workflow

An xSPDE session can either run simulations interactively, described in section 4, or else using a function file called a project file. In either case, the Matlab path must include the xSPDE folder. For generating graphs automatically, the script input or project function should end with the combined function **xspde**.

Alternatively, it can be useful to divide xSPDE into its simulation function, xSIM, and its graphics function, xGRAPH, to allow graphs to be made at a later time from the simulation. In this case the function *xsim* runs the simulation, and *xgraph* makes the graphs. The two-stage option is better for running batch jobs which you can graph at a later time.

### 6.1.1   Batch input template

To create a data file, you must enter the filename when running the simulation, using the $p.file = filename$ input. A typical xSPDE project function of this type, where all the data is stored is as follows:

```
function e = project.m
p.[label1] = [parameter1];
p.[label2] = ...;
p.file = '[myfile].mat'
[e,~,p] = xsim(p);
xgraph(p.file);
end
```

Alternatively, for an interactive session one can use the commands:

```
...
[e,data,p] = xsim(p);
xgraph(data,p);
...
```

This is specially useful if one wishes to have direct access to the data and graphics options, with possible multiple trials. When preparing a project file using the editor, click on the Run arrow above the editor window to run the job.

A batch job workflow is as follows:

- Create the metadata $p$, including a file name, eg, *p.file='myfile.mat'*.

- Change the Matlab directory path to your preferred directory.

- Run the simulation with *[e,data, p] = xsim(p)*, or just *xsim(p)*.

- Run *xgraph(p.file)*, and the data will be graphed.

- Alternatively, *xgraph(p.file,p)* allows you to change the inputs in the structure $p$.

- Graph outputs can be stored using the *p.saveeps=1* and/or *p.savefig=1* options.

You can use either Matlab (.mat) or standard HDF5 (.h5) file-types for data storage. If raw data is generated it will be stored too, but the files can be large. For stored graphics files the options are encapsulated postscript (.eps) files or Matlab graphics (.fig) files, obtained using the graphics input switches *p.saveeps* and/or *p.savefig*.

## 6.2 Graphical data

The following table show how xSPDE output data is stored, which helps customize and extend the code. There are several different types of arrays used. The observed averages are generated internally from the observe functions, *p.observe*. These are then modified by user functions *p.function*, and exported as graphics data.

The internal averages and the exported graphics data are as follows:

| Label | Indices | Description |
|:---:|:---:|:---:|
| av | $\{n\}(\ell, \mathbf{j})$ | Observed averages |
| d | $\{s\}\{n\}(\ell, \mathbf{j}, c)$ | Graph data |

Here:

- $s$ is the sequence index

- $n$ is the graph index

- $\ell$ is the graphics line index

- $j_1$ is the time index

- $\mathbf{j} = j_1, j_2, \ldots j_d$ is the space-time index

- $c$ is the check index

### 6.2.1 Check index uses

There are multiple uses for the last index, $c$. It can be omitted if needed. If present, it stores data for errors and comparisons. This is indicated by the input parameter field $p.errors > 0$, which is the index of the largest error field. If there are no input parameters, or $p.errors = 0$, there is no error or comparison index. The standard value that xSIM outputs is $p.errors = 3$.

When the check index present, the usual index values are defined as follows:

$c = 1$ for the average of the $n$-th output function

$c = 2$ for the time-step error,

$c = 3$ for the sampling error.

$c = 4$ for (optional) comparisons

$c = 5$ for (optional) systematic comparison errors

$c = 6$ for (optional) statistical comparison errors

Finally, if xGRAPH is used with data from an other source, with no simulation error fields, but with comparisons, then one simply puts $p.errors = 1$, or if there is just one input error field $p.errors = 2$.

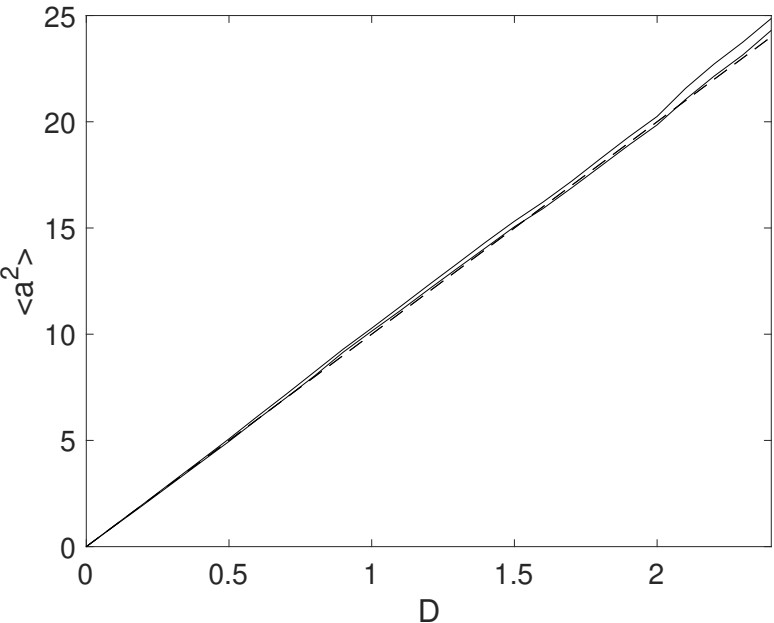

Figure 10: Scanned parameter output with a variable diffusion, for the case of a pure Wiener process, $\dot{a} = Bw(t)$. Exact value is the dashed line.

## 6.3 Scanned parameter plots

Since xSIM is a function that can be called, plots of results against simulation parameters are possible. This requires repeated calls to xSIM with different parameter values, together with data storage in an xGRAPH compatible form, and a call to xGRAPH. If different random seeds are required, the seed needs to be reset in each call. The relevant axes points plotted, labels and the values of scanned parameters also need to be input.

The simulation function xSIM uses the last data array index, $c$, to store the data values and up to two corresponding errors. This takes up three index values. A value of $c = 4$ is used to store comparison data, and its errors if there are any in $c = 5, 6$. This can be used for exact results, approximations, or experimental data.

### 6.3.1 Example: Scanned diffusion

As an example, consider the simplest possible stochastic equation, with a scanned diffusion:

$$\dot{a} = Bw(t). \tag{174}$$

The equation is integrated over the interval $t = 0 : 10$, with $a = 0$ initially, using $10^4$ trajectories to give an expected error of around $\pm 1\%$. The variance of $a$ at $t = 10$ is plotted as a function of $D = B^2$, then compared to an exact value. The result is in Fig (10). The corresponding code is given as well.

```
function e = WienerScan()
p.name = 'Wiener process';
p.ensembles = [1000,10];
p.points = 12;
p.deriv = @(a,z,p) z*p.B;
p.observe = @(a,p) a.^2;
p.olabels = {'<a^2>'};
p.glabels{1} = {'D'};
scanpoints = 25;
data{1}{1} = zeros(1,scanpoints,4);
for j = 1:scanpoints
  p.seed = j;
  p.B = sqrt((j-1)*0.1);
  [e,data1,input,~] = xsim(p);
  data{1}{1}(1,j,1:3) = data1{1}{1}(1,p.points,:);
  xk{1}{1}(j) = p.B^2;
  D(j) = p.B^2;
end
data{1}{1}(1,:,4) = input.ranges(1)*D(:);
input.xk = xk;
input.axes{1}{1} = 1:scanpoints;
xgraph(data,input);
end
```

Here $p.deriv$ defines the time derivative function $\dot{a}$, with $w$ being the delta-correlated Gaussian noise that is generated internally.

## 6.4   Project examples

### 6.4.1   Kubo project

To get started on more complex programs, we next simulate the Kubo oscillator, which is an oscillator with a random frequency:

$$\dot{a} = iaw. \tag{175}$$

**Exercises**

- Simulate the Kubo oscillator using a file, *Kubo*.*m*, with two ensemble levels to allow sampling error estimates. The error vector *error* gives the total time-step error plus the sampling error.

- Increase the first ensemble size to check how it modifies the sampling errors.

```
function [error] = Kubo()
p.name = 'Kubo oscillator';
p.ensembles = [400,16];
p.initial = @(v,p) 1;
p.deriv = @(a,w,~) 1i*a.*w;
p.olabels = {'<a_1>'};
p.file = 'kubo.mat';
[error,~,~,~] = xsim(p);
xgraph(p.file);
end
```

This function generates a data file, `kubo.mat`. If you run this twice without deleting the earlier file, you will get a warning and the old file will be moved to a backup file-name, `kubo_1.mat`, to protect the earlier data. Note that xGRAPH will graph the data in the most recent file saved.

You can also include modified graphics parameters as a second input when running `xGRAPH`, just in case the first graphs you generate need further changes.

### 6.4.2 Gaussian diffraction

Free diffraction and absorption of a Gaussian wave-function in $d - 1 = s$ space dimensions, is given by the partial differential equation (PDE):

$$\frac{da}{dt} = -\frac{\gamma}{2}a + \frac{i}{2}D\nabla^2 a. \tag{176}$$

The corresponding stochastic partial differential equation (SPDE) includes additional noise, so that:

$$\frac{da}{dt} = -\frac{\gamma}{2}a + \frac{i}{2}D\nabla^2 a + bw(t,x). \tag{177}$$

The xSPDE spectral definition in space is:

$$\tilde{a}(t,\mathbf{k}) = \frac{1}{[2\pi]^{s/2}} \int e^{i\mathbf{k}\cdot\mathbf{x}} a(t,\mathbf{x}) d\mathbf{x}. \tag{178}$$

Together with the initial condition that $a(0,x) = exp(-|\mathbf{x}|^2/2)$, this has an exact solution for the diffracted intensity with $b = 0$, in either ordinary space or momentum space:

$$
\begin{aligned}
|a(t,\mathbf{x})|^2 &= \frac{1}{\left(1+(Dt)^2\right)^{s/2}} exp\left(-|\mathbf{x}|^2/\left(1+(Dt)^2\right) - \gamma t\right) \\
|\tilde{a}(t,\mathbf{k})|^2 &= exp\left(-|\mathbf{k}|^2 - \gamma t\right).
\end{aligned}
\tag{179}
$$

**Exercises**

- Simulate Gaussian diffraction in three dimensions using an xSPDE function

- Check your results against the exact solution

- The example below stores data in a standard HDF5 file.

```
function [e] = Gaussian()
p.dimensions = 4;
p.initial = @(v,p) exp(-0.5*(p.x.^2+p.y.^2+p.z.^2));
p.linear = @(p) 1i*0.05*(p.Dx.^2+p.Dy.^2+p.Dz.^2);
p.observe = @(a,p) a.*conj(a);
p.olabels = '|a(x)|^2';
p.file = 'Gaussian.h5';
p.images = 4;
e = xsim(p);
xgraph(p.file);
end
```

- **Add an additive complex noise of** $0.01(w_1+iw_2)$ **to the Gaussian differential equation, then replot with an average over** 100 **samples.**

- Work out the exact solution and repeat the comparisons.

Note that for this, you'll need to add: $p.deriv = @(a,w,p) .. + 0.01*(w(1,:)+i*w(2,:))$

## 6.5 Hints

- When first using xSPDE, it is a good idea to run the batch test script, *Batchtest*.

- *Batchtest* uses the Matlab parallel toolbox installation. If you have no license for this, omit the third ensemble setting.

- To create a project file, it is often easiest to start with an existing example function using a similar equation: see the xAMPLES folder.

- Graphics parameters can be included in the xSIM inputs to modify graphs.

- Comparison functions can be included if you want to compare with analytic results.

- Sections 9 and 10 list the input parameters.

# 7 Stochastic methods

*This section describes the general background to the choices of methods available, and how to add custom numerical algorithms if required.*

## 7.1 Introduction to algorithms

Stochastic, partial and ordinary differential equations are central to numerical mathematics. Ordinary differential equations have been known in some form ever since calculus was invented. There are a truly extraordinary number of algorithms used to solve these equations. One program cannot possibly provide all of them. This section provides an overview of the included algorithms, for the more advanced and expert user.

xSPDE has six built-in choices of algorithm, with defaults. All built-in methods have an interaction picture and can be used with any space dimension, including $dimensions = 1$, which is an ordinary stochastic equation. All can be used with stochastic or with non-stochastic equations, and with order extrapolation.

For stochastic equations, the Euler method requires an Ito form of stochastic equation, the implicit Euler method requires an implicit Ito form, while the others should be used with the Stratonovich form of calculus. Each uses the interaction picture to take care of exactly soluble linear terms.

### 7.1.1 Standard methods

The standard xSIM algorithms given below are available for ODEs, PDEs, SDEs and SPDEs. More advanced algorithms for specialized cases are described in section 7.

For stochastic differential equations, which are non-differentiable, the usual rules of calculus do not apply because stochastic noise is non-differentiable. It has fluctuations proportional to $1/\sqrt{dt\,dV}$, for noise defined on a lattice with temporal cell-size $dt$ and spatial cell-size $dV$. Hence, the usual differentiability and smoothness properties required to give high-order convergence for standard Runge-Kutta methods are simply not present. Instead, xSPDE has a built-in extrapolation to zero step-size for high-order stochastic convergence.

Many more complex higher order algorithms for stochastic integration exist but are not included in the current xSPDE distribution, and users are encouraged to contribute their favorite methods.

We note here that there are multiple error sources possible. SDE/SPDE errors are often dominated by the sampling error, not discretization. In addition, all convergence theorems only apply to the limit of zero step-size. One may be very far from this regime in a given practical calculation. Analytic error estimates also have prefactors which are hard to calculate. However, xSPDE can numerically estimate both the discretization and sampling error for any given average observable.

## 7.2 General differential form

The general equation treated is given in differential form as

$$\frac{\partial a}{\partial t} = A[\nabla, a, t] + \underline{\mathbf{B}}[\nabla, a, t] \cdot \zeta(t) + \underline{\mathbf{L}}[\nabla] \cdot a. \tag{180}$$

It is convenient for the purposes of describing interaction picture methods, to introduce an abbreviated notation as:

$$\mathcal{D}[\mathbf{a}, t] = A[a, t] + \underline{\mathbf{B}}[a, t] \cdot \zeta(t). \tag{181}$$

Hence, we can rewrite the differential equation in the form:

$$\frac{\partial \boldsymbol{a}}{\partial t} = \mathcal{D}[\mathbf{a}, t] + \underline{\mathbf{L}}[\boldsymbol{\nabla}] \cdot \boldsymbol{a}. \tag{182}$$

### 7.2.1 Linear propagator

Next, we define a linear propagator. This is given formally by:

$$\mathcal{P}(\Delta t) = \exp\left(\Delta t \underline{\mathbf{L}}[\boldsymbol{\nabla}]\right). \tag{183}$$

Typically, but not necessarily, this is evaluated in Fourier space, where it should be just a diagonal term in the momentum vector conjugate to the transverse space coordinate. It will then involve a Fourier transform, multiplication by an appropriate function of the momentum, and then an inverse Fourier transform afterwards. For simplicity, the stochastic noise is assumed constant throughout the interval $dt$. The reader is referred to the literature for more details.

It is simple to add your own algorithm if you prefer a different one. Note that if they use an interaction picture, then *ipsteps* must be given explicitly to specify the interaction picture duration, where *ipsteps* gives the number of sequential propagator steps in time required for the method.

## 7.3 Standard methods

The standard methods are listed below. All of these can be used with any equation: ODE, SDE, PDE or SPDE, either with or without a linear interaction picture term.

### 7.3.1 *Euler*: Ito-Euler

This is an explicit Ito-Euler method using an interaction picture. While traditional, it is not generally recommended. If it is used, very small step-sizes will generally be necessary to reduce errors to a usable level. This is because it is is only convergent to first order deterministically and tends to have large errors.

It is designed for use with an Ito form of stochastic equation. It requires one IP transform per step ($p.ipsteps = 1$). Starting from time $t = t_n$, to get the next time point at $t = t_{n+1} = t_n + \Delta t$, one calculates:

$$\begin{aligned} \Delta \mathbf{a}_n &= \Delta t \mathcal{D}[\mathbf{a}_n, t_n] \\ \mathbf{a}_{n+1} &= \mathcal{P}(\Delta t) \cdot [\mathbf{a}_n + \Delta \mathbf{a}_n] \end{aligned} \tag{184}$$

### 7.3.2 *Implicit*: implicit Ito-Euler

This is a fully implicit Ito-Euler method using an interaction picture. It is more robust, though slower, than the explicit form. If it is used, very small step-sizes will generally be necessary to reduce errors to a usable level.

This is because it is is only convergent to first order, and therefore tends to have large errors. It is designed for use with an implicit Ito form of stochastic equation. Note that this implies double the usual Stratonovich correction!

It requires one IP transform per step ($p.ipsteps = 1$). Starting from time $t = t_n$, to get the next time point at $t = t_{n+1} = t_n + \Delta t$, one calculates, using iteration to get the implicit result of the next time-point:

$$
\begin{aligned}
\bar{\mathbf{a}}^{(0)} &= \mathcal{P}(\Delta t) \cdot [\mathbf{a}_n] \\
\bar{\mathbf{a}}^{(i)} &= \bar{\mathbf{a}}^{(0)} + \Delta t \mathcal{D}\big[\bar{\mathbf{a}}^{(i-1)}, t_{n+1}\big] \\
\mathbf{a}_{n+1} &= \bar{\mathbf{a}}^{(iter)}
\end{aligned}
\tag{185}
$$

$$
\begin{aligned}
\tilde{\mathbf{a}}_n &= \mathcal{P}(\Delta t) \cdot [\mathbf{a}_n] \\
\Delta \mathbf{a}_n &= \Delta t \mathcal{D}\big[\tilde{\mathbf{a}}_n + \Delta \mathbf{a}_n, t_n\big] \\
\mathbf{a}_{n+1} &= \tilde{\mathbf{a}}_n + \Delta \mathbf{a}_n
\end{aligned}
\tag{186}
$$

### 7.3.3 *MP*: **Midpoint**

This is a semi-implicit midpoint method using an interaction picture. It gives good results for stochastic and stochastic partial differential equations. It is convergent to second order in time for deterministic equations and for stochastic equations with commuting noise. It is strongly convergent and robust. It requires two half-length IP transforms per step ($p.ipsteps = 2$).

To get the next time point, one calculates a midpoint derivative iteratively at time to get the next time point at $t = t_{n+1/2} = t_n + \Delta t/2$, to give an estimated midpoint field $\bar{\mathbf{a}}^{(i)}$, usually with four iterations. The number of iterations can be changed:

$$
\begin{aligned}
\bar{\mathbf{a}}^{(0)} &= \mathcal{P}\left(\frac{\Delta t}{2}\right) \cdot [\mathbf{a}_n] \\
\bar{\mathbf{a}}^{(i)} &= \bar{\mathbf{a}}^{(0)} + \frac{\Delta t}{2} \mathcal{D}\big[\bar{\mathbf{a}}^{(i-1)}, t_{n+1/2}\big] \\
\mathbf{a}_{n+1} &= \mathcal{P}\left(\frac{\Delta t}{2}\right) \cdot \big[2\bar{\mathbf{a}}^{(iter)} - \bar{\mathbf{a}}^{(0)}\big]
\end{aligned}
\tag{187}
$$

This is the default method for stochastic cases.

### 7.3.4 *MPadap*t: **adaptive midpoint**

This is an implicit midpoint method using an interaction picture, together with an adaptive technique for integrating highly nonlinear equations. At low amplitudes it is identical to the standard midpoint method. For amplitudes $|a_i|^2$ above a critical value, $r.adapt$, the amplitude is inverted and propagated using the differential equation for its inverse.

Initially a switch $p$ is set to 1 for low amplitudes, and $-1$ for high amplitudes. To get the next time point, one calculates a midpoint derivative iteratively at time to get the next time point at $t = t_{n+1/2} = t_n + \Delta t/2$, to give an estimated midpoint field $\bar{\mathbf{a}}^{(i)}$, as above, but with the derivative

modified to give the derivative of $a_i^p$:

$$
\begin{aligned}
\bar{\mathbf{a}}^{(0)} &= \mathcal{P}\left(\frac{\Delta t}{2}\right) \cdot [\mathbf{a}_n] \\
\tilde{\mathbf{a}}^{(0)} &= \mathbf{a}_n^p \\
\tilde{\mathbf{a}}^{(i)} &= \tilde{\mathbf{a}}^{(0)} + \frac{\Delta t}{2} p \left[\tilde{\mathbf{a}}^{(i-1)}\right]^{1-p} \left(\mathcal{D}\left[[\tilde{\mathbf{a}}^{(i-1)}]^p, t_{n+1/2}\right]\right) \\
\mathbf{a}_{n+1} &= \mathcal{P}\left(\frac{\Delta t}{2}\right) \cdot \left[2\tilde{\mathbf{a}}^{(iter)} - \tilde{\mathbf{a}}^{(0)}\right]^p
\end{aligned}
\tag{188}
$$

### 7.3.5 *RK2*: second order Runge-Kutta

This is a second order Runge-Kutta method using an interaction picture. It is convergent to second order in time for non-stochastic equations, and for stochastic equations with additive noise, but otherwise it is first order. It often has higher errors than midpoint methods. It requires two IP transforms per step, but each is a full time-step long (*p.ipsteps* = 1).

To get the next time point, one calculates:

$$
\begin{aligned}
\bar{\mathbf{a}} &= \mathcal{P}(\Delta t) \cdot [\mathbf{a}_n] \\
\mathbf{d}^{(1)} &= \Delta t \mathcal{P}(\Delta t) \cdot \mathcal{D}[\mathbf{a}_n, t_n] \\
\mathbf{d}^{(2)} &= \Delta t \mathcal{D}\left[\bar{\mathbf{a}} + \mathbf{d}^{(1)}, t_{n+1}\right] \\
\mathbf{a}_{n+1} &= \bar{\mathbf{a}} + \left(\mathbf{d}^{(1)} + \mathbf{d}^{(2)}\right)/2.
\end{aligned}
\tag{189}
$$

### 7.3.6 *RK4*: fourth order Runge-Kutta

This is a fourth order Runge-Kutta method using an interaction picture. It is convergent to fourth order in time for non-stochastic equations, but for stochastic equations it can be more slowly convergent than the midpoint method. It requires four half-length IP transforms per step (*ipsteps* = 2). To get the next time point, one calculates four derivatives sequentially:

$$
\begin{aligned}
\bar{\mathbf{a}} &= \mathcal{P}\left(\frac{\Delta t}{2}\right) \cdot [\mathbf{a}_n] \\
\mathbf{d}^{(1)} &= \frac{\Delta t}{2} \mathcal{P}\left(\frac{\Delta t}{2}\right) \cdot \mathcal{D}[\mathbf{a}_n, t_n] \\
\mathbf{d}^{(2)} &= \frac{\Delta t}{2} \mathcal{D}\left[\bar{\mathbf{a}} + \mathbf{d}^{(1)}, t_{n+1/2}\right] \\
\mathbf{d}^{(3)} &= \frac{\Delta t}{2} \mathcal{D}\left[\bar{\mathbf{a}} + \mathbf{d}^{(2)}, t_{n+1/2}\right] \\
\mathbf{d}^{(4)} &= \frac{\Delta t}{2} \mathcal{D}\left[\mathcal{P}\left(\frac{\Delta t}{2}\right)\left[\bar{\mathbf{a}} + 2\mathbf{d}^{(3)}, t_{n+1}\right]\right] \\
\mathbf{a}_{n+1} &= \mathcal{P}\left(\frac{\Delta t}{2}\right) \cdot \left[\bar{\mathbf{a}} + \left(\mathbf{d}^{(1)} + 2\left(\mathbf{d}^{(2)} + \mathbf{d}^{(3)}\right)\right)/3\right] + \mathbf{d}^{(4)}/3
\end{aligned}
\tag{190}
$$

This might seem the obvious choice, having the highest order. However, it can converge at a range of apparent rates, depending on the relative importance of stochastic and non-stochastic terms. Due to its use of differentiability, it may converge more slowly than the midpoint method with stochastic terms present. It is the default for ODE and PDE cases.

## 7.4 Advanced methods

Three more advanced method libraries are included here, namely *weighted*, *projected* and *forward-backward* stochastic differential equations. If you have a favorite algorithm that is not included, user-defined algorithms and libraries can be added. The existing methods are listed below, and the corresponding .m-files can be used as a model.

Define the routine, for example "myalgorithm.m", set $p.method = @myalgorithm$, then adjust the input value of *ipsteps* and *order* if these need be changed to a new value. The interaction-picture transform, *prop*, can also be changed if the built-in choice is not sufficient. The xSPDE algorithms available currently treat

- ordinary (and partial) differential equations

- stochastic differential equations

- stochastic partial differential equations

- weighted stochastic differential equations

- projected stochastic differential equations,

- forward-backward stochastic differential equations

The first three have already been treated. In this section, we explain the last three cases, which involve more specialized libraries of functions.

### 7.4.1 Additional inputs

Some of the more advanced features of the libraries require additional input parameters. In particular:

**backfields** is used for forward-backward stochastic equations, describing backward time components. These are described in the *Forward-backward* section. Note that **fields** is still used, and it gives the total number of forward+backward fields.

**auxfields** gives the number of auxiliary fields. These have a functional definition (*defines*) that includes both a field and noise variable, as needed for spectral observables. Field index numbers $i$ greater than $fields$ access the auxiliary fields in the *observe* function.

## 7.5 Weighted library

In some types of stochastic equation, there is a weight associated with each trajectory, which is used to weight the probability of the trajectory [91]. This type of equation is sometimes found when dealing with quantum trajectories [92, 93] and feedback [94].

The equations still have the standard form of Eq (1), with an extra weight equation, Eq (4). However, the results for mean values are weighted by a term $\exp(\Omega(t))$, so that:

$$\langle \mathbf{O} \rangle_\Omega = \frac{\sum_n \mathbf{O}(\mathbf{a}^{(n)}) \exp(\Omega^{(n)}(t))}{\sum_n \exp(\Omega^{(n)}(t))}. \tag{191}$$

This reduces to the standard expression of Eq (3) in the case that $\Omega(t) = 0$. To simulate these equations automatically, the weight exponent $\Omega$ is integrated as the *last* field in the vector $\mathbf{a}$, which

must have at least two components. A nonzero threshold weight, *thresholdw*, must be entered to allow calculation of breeding.

With these changes, averages in each vector ensemble are calculated using Eq (191). Before each plotted step in the calculation, a breeding calculation is carried out. There are $p.steps(1)-1$ of these in total. During breeding, any weight such that $\exp\left(\Omega^{(n)}\right) < thresholdw/\langle\exp(\Omega)\rangle$ is removed.

The most probable trajectory is then duplicated to replace the low-weight trajectory. Both exponential weights are halved, so the total weight of the remaining trajectories is unchanged. If they are complex, weights such that $\exp\left(Re\left(\Omega^{(n)}\right)\right) < thresholdw/\langle\exp(Re(\Omega))\rangle$ are removed, and the real weight of the bred trajectory is reduced, which removes any low-weight trajectories that don't contribute. When used, the internal variable *p.breedw* is set to allow the fraction of trajectories that are bred per step to be monitored.

### 7.5.1 Example

The following example shows how weights are implemented.

```
function [e] = Weightcheck()
p.name = 'Weightcheck';
p.ensembles = [10000,10,1];
p.fields = 2;
p.points = 6;
p.order = 2;
p.thresholdw = 0.1;
p.diffplot = 1;
p.initial = @(w,p) [1+w(1,:);0*w(2,:)];
p.deriv = @(a,z,p) [-a(1,:)+ z(1,:);-a(2,:)+ z(2,:)];
p.observe{1} = @(a,p) a(1,:);
p.observe{2} = @(a,p) p.breedfrac;
p.compare{1} = @(p) exp(-p.t);
p.olabels{1} = '<a>';
p.olabels{2} = '<fractional breeds per step>';
e = xcheck(2,p);
end
```

This algorithm converges with second-order accuracy for this exercise, due to the structure of the equation. The example also demonstrates how to use the *xcheck* function instead of *xspde*, to check convergence.

## 7.6 Projection library

It is sometimes necessary to constrain an equation to a sub-manifold [95], with an equation of form:

$$\mathbf{f}(\mathbf{a}) = 0, \tag{192}$$

where $\mathbf{f}(\mathbf{a})$ is a scalar or vector function that defines the relevant manifold in Euclidean space. The projected SDE then has the form of a Stratonovich SDE, where:

$$\frac{\partial\mathbf{a}}{\partial t} = \mathcal{P}_{\mathbf{a}}^{\parallel}\left[\mathbf{A}[\mathbf{a}]+\underline{\mathbf{B}}[\mathbf{a}]\cdot\mathbf{w}(t)\right], \tag{193}$$

where $\mathcal{P}_{\mathbf{a}}^{\parallel}$ is a tangential projection operator at location $\mathbf{a}$ on the sub-manifold, and as usual, $\mathbf{A}$ is a vector, $\underline{\mathbf{B}}$ a matrix and $\mathbf{w}$ is a real Gaussian noise vector, delta-correlated in time. Similarly, the general stochastic partial differential equation can be written in projected form as

$$\frac{\partial \mathbf{a}}{\partial t} = \mathcal{P}_{\mathbf{a}}^{\parallel}\left[\mathbf{A}[\mathbf{a}] + \underline{\mathbf{B}}[\mathbf{a}] \cdot \mathbf{w}(t,\mathbf{x}) + \underline{\mathbf{L}}[\nabla,\mathbf{a}]\right]. \tag{194}$$

When numerically integrating these, it is also useful to have a normal projection $\mathcal{P}^{\perp}$ available. This is used to normally project to the nearest point on the manifold, to eliminate constraint errors. These are solved using functions collected in a projection library, to provide the specialized methods that are needed for this purpose.

The projection library has three predefined algorithms,

- *Enproj*,

- *MPproj*,

- *MPnproj*.

Here the capital E stands for Euler, MP for midpoint. All use tangential projection. The letter *n=normal* indicates if an additional normal projection is used. In all cases, if it is present, a normal projection is used last. The recommended type is ***MPnproj***, due to its much lower errors.

Tangential and normal projections are needed to define the geometry of any sub-manifold. These are input by setting the variable *project* equal to a function handle that defines the projection. These can be user provided if required. There are three different predefined manifold geometry types, which need different inputs, given below.

### 7.6.1   Calling the *project* function

The calling arguments for the *project* function are: *(d,a,n,p)*, where d is a vector to be tangentially projected at location a, a is the current (near)-manifold location, n is an option switch, and p is the parameter structure.

The options available in any *project* implementation are defined as:

- $n = 0$ returns the tangent vector for testing

- $n = 1$ returns the tangential projection of $d$ at $a$

- $n = 2$ returns the normal projection of $a$, where $d$ is not used

- $n = 4$ returns the constraint function at $a$ for testing

The projections defined in an xSPDE *project* function can be of any type. Arbitrary dimension reduction and manifold geometry is possible. Currently in the examples, dimensionality is reduced by 1, and normal projections use fixed point iterations, defined by *iterproj*.

### 7.6.2   The predefined manifold geometries

The current manifolds, by setting p.project = @Quadproj ..., are as follows:

1. Quadratic - *Quadproj* - needs: *qcproj* defined by $f = \sum q_{ij} x^i x^j - 1 = 0$

2. Polynomial - *Polproj* - needs: *vcproj* defined by $f = \sum v_i (x^i)^p - 1 = 0$

3. Catenoid - *Catproj* - uses fixed coefficients defined by $f = (x_1)^2 + (x_2)^2 - (sinh(x_3))^2 - 1 = 0$

Any other manifold can be used by replacing these predefined manifolds with an appropriate *project* function.

## 7.7 Forward-backward library

The xSPDE forward-backward library implements an iterative stochastic method which propagates an SDE or PSDE forward and backward in time. This is used to treat Q-function phase-space methods, which do not have a positive-definite diffusion [96]. The iteration converges in simple cases, typically with no cross-coupling apart from boundary conditions. It uses the algorithm *MPfb*.

The general FB equations have the following structure, written as an integral equation to make it clear what the relevant boundary conditions are:

$$
\mathbf{p}(t) = \mathbf{p}(0, \mathbf{q}(0)) + \int_0^t \left\{ \mathbf{A}^p \left[ \mathbf{a}(t') \right] dt' + \underline{\mathbf{B}}^p \left[ \mathbf{a}(t') \right] \cdot d\mathbf{w}^p(t') \right\}
$$
$$
\mathbf{q}(t) = \mathbf{q}(T, \mathbf{p}(T)) + \int_t^T \left\{ \mathbf{A}^q \left[ \mathbf{a}(t') \right] dt' + \underline{\mathbf{B}}^q \left[ \mathbf{a}(t') \right] \cdot d\mathbf{w}^q(t') \right\}. \tag{195}
$$

Here, $\mathbf{a} = [\mathbf{p}, \mathbf{q}]$ includes forward components $\mathbf{p}$ and backwards components $\mathbf{q}$. These have "initial" conditions in the past and the future, respectively, and can depend on random inputs, just as with ordinary stochastic equations.

The library includes the *xpathfb* function which replaces the *xpath* function, which is used automatically. However, the user must specify a modified step integrator, either *Eulerfb* or *MPfb*. The *initial* and deriv routines require additional arguments, which are described in the table below, and are used during the iteration scheme.

The noise terms $\mathbf{w} = [\mathbf{w}^p, \mathbf{w}^q]$ are uncorrelated real Gaussian noises:

$$
\left\langle dw_i^\alpha(\mathbf{x}) \, dw_j^\beta(\mathbf{x}') \right\rangle = \delta_{ij} \delta_{\alpha\beta} dt. \tag{196}
$$

This is solved in differential form, where $t_- = T - t$, as:

$$
\frac{\partial \mathbf{p}}{\partial t} = \mathbf{A}^p[\mathbf{a}] + \underline{\mathbf{B}}^p[\mathbf{a}] \cdot \mathbf{w}^p(t)
$$
$$
\frac{\partial \mathbf{q}}{\partial t_-} = \mathbf{A}^q[\mathbf{a}] + \underline{\mathbf{B}}^q[\mathbf{a}] \cdot \mathbf{w}^q(t). \tag{197}
$$

Each equation is solved by iteration. The previous value of the counter-propagating field, ie $\mathbf{a}^{(n-1)}$, is used to solve for $\mathbf{a}^{(n)}$ in step $n$, since the current value is not yet known. That is, the algorithm is:

$$
\frac{\partial \mathbf{p}^{(n)}}{\partial t} = \mathbf{A}^p \left[ \mathbf{p}^{(n)}(t), \mathbf{q}^{(n-1)}(t) \right] + \underline{\mathbf{B}}^p \left[ \mathbf{p}^{(n)}(t), \mathbf{q}^{(n-1)}(t) \right] \cdot \mathbf{w}^p(t) \tag{198}
$$
$$
\frac{\partial \mathbf{q}^{(n)}}{\partial t_-} = \mathbf{A}^q \left[ \mathbf{p}^{(n-1)}(t_-), \mathbf{q}^{(n)}(t_-) \right] + \underline{\mathbf{B}}^q \left[ \mathbf{p}^{(n-1)}(t_-), \mathbf{q}^{(n)}(t_-) \right] \cdot \mathbf{w}^q(t_-),
$$

Convergence is the responsibility of the user, and the algorithm has a fixed number of iterations. The starting point of the iteration is the path function $fbfirst$. The simulation requires the following additional inputs, including $backfields$, defining the backward components.

| Label | Type | Typical value | Description |
|---|---|---|---|
| *backfields* | integer vector | 1 | Number of backward variables |
| *initialfb* | function handle | $@(a0, a1, w, p)$ | Initial value for $a$ |
| *firstfb* | function handle | $@(a0, nc, p)$ | First trajectory estimate |
| *iterfb* | integer | 2 | Forward-backward iterations |
| *method* | function handle | $@xMPfb$ | Forward-backward algorithm |
| *deriv* | function handle | $@(a, a_-, w, p)$ | Derivative function |

In *initialfb*, the $a0$ fields from the previous iteration are at the *first* times computed previously, so $\mathbf{a}0 = \left[ \mathbf{p}^{(n-1)}(0), \mathbf{q}^{(n-1)}(T) \right]$, while the $a1$ fields are evaluated at the *last* times computed from the previous iteration, so $\mathbf{a}1 = \left[ \mathbf{p}^{(n-1)}(T), \mathbf{q}^{(n-1)}(0) \right]$.

On the first call to *initialfb, with $p.iter = 1$*, a startup procedure is used. In the startup procedure, *a0* is generated internally by *initialfb*. However, *a1* is obtained in the internal calling function *xpathfb* using the output of *firstfb,* which gives an initial iterative path estimate of $a$. It returns a default path equal to the initial boundary value $a0$, if not defined by the user. More generally, it should be set to a value to allow iterations to converge. The calling arguments of *firstfb* include the initial boundaries $a0$ and the usual check index ($nc = 1, 2$).

The *initial* function returns $a0$, giving the current initial values. On the first iteration *initial* returns an internally defined $a0$. Subsequently it requires $a0$, the stored first iteration boundaries as well as $a1$, the previous iteration end-points. The estimate for the previous path in the deriv function is obtained from the iteration starter function *firstfb* on the first iteration, and subsequently from a stored value.

Internally, the raw fields $\mathbf{a} = [\mathbf{p}, \mathbf{q}]$ are stored in complementary time-orders, with $\mathbf{p}$ solved normally in forward time, and $\mathbf{q}$ solved in reverse temporal order.

When the previous iteration field is passed to $deriv$, the time-orders of the previous iteration are reversed so that previous iteration $\mathbf{q}$ times are the same as $\mathbf{p}$ times, and vice-versa. The previous fields are therefore at the same time as those of the complementary present field. When passed to $observe$, both $\mathbf{p}, \mathbf{q}$ are given in time-increasing order to allow synchronized observations.

### 7.7.1  Example:

This is a trivial example, to illustrate the code structure. It has two counter-propagating stochastic processes, one decaying in the forward time direction, and one decaying in the backward time direction.

```
function [e] = Fbcheck()
p.ranges = 1;
p.fields = 2;
p.backfields = 1;
p.initialfb = @(~,~,w,p) 1+0.5*w;
p.ensembles = [400,1,1];
p.method = @MPfb;
p.deriv = @(a,~,w,p) -a + w;
e = xspde(p);
end
```

# 8 Integration errors

*This section describes how xSPDE estimates errors from time discretization and statistical sampling. Other numerical errors require manual checks.*

Errors and the need for error-checking are an integral part of numerical calculations. This is more subtle in stochastic equations, because there are both multiple sources of errors and multiple outputs. The xSPDE philosophy is to compute the most relevant errors for every average output, since each output average may have quite different errors.

## 8.1 Discretization errors

To check convergence, xSPDE repeats the calculations at least twice for checking step-sizes, and many times more in stochastic cases to estimate sampling errors. These checks can be turned on and off. *If you think the checks make xSPDE slow, turn them off - but you won't get any error-estimates.* Whatever the application, you will find the error-estimates useful.

If the errors are too large relative to the application, you should decrease the time-steps or increase the number of samples. Which is needed depends on the type of error.

Errors caused by the finite time-domain step-size are checked automatically provided that $checks = 1$ is specified, which is the default option. If $checks = 0$ is used, there is no time-domain error check.

Errors due to a finite step-size are estimated by running a check simulations with half the initial step-size and the same random sequence, extrapolating to zero step-size if $order > 0$ is specified, then returning an error bound as the difference of the two most accurate results. Any 2D output graphs plot error-bars if $checks = 1$ was specified, provided they are large enough to plot. RMS output errors are also reported. Individual error bounds $e(o)$ are given in the output data, and the plots give $\bar{o} \pm e(o)$.

Error-bars below a minimum relative size compared to the vertical range of the plot, specified by the graphics variable $minbar$, are not plotted. The default for this is $minbar = 0.01$. All error bars are calculated individually for each type of data average. Minbar is a cell array that can can be set for each type of average or graph. If the cell argument is omitted, it applies globally. Error estimates are also given for functional transforms of averages.

If the errors are too large, one can either increase the *points*, which gives more plotted points and lower errors, or increase the *steps*, which reduces the step size without changing the data resolution. The default algorithm and extrapolation order can also be changed. Error bars on graphs can be removed by setting $checks = 0$ or increasing $minbar$.

Discretization errors caused by the finite spatial lattice are not currently checked in the xSIM code. They must be checked by comparing results with different transverse lattice ranges and step-size. Similarly, errors from probability binning are not checked.

### 8.1.1 Discretization error outputs

In xSPDE, the discretization or step-size error due to finite time-step sizes is called the "step" error. For checking step errors, xSPDE allows the user to specify $checks = 1$, which is the default option. This gives one integration at the specified step-size, and one at half the specified step-size. The data is plotted using the more accurate fine step-size results, but with the coarse time lattice in order to calculate the estimated discretization errors.

The RMS value of the step error for each computed function, normalized by the maximum modulus of the observable, is printed out after each xSPDE simulation. If the expected comparison value is zero, the absolute value is given.

Both fine and coarse time-step results employ identical underlying random noise processes, from the same initial random seed. To compensate for the grid size, the coarse time-step uses a sum of two successive fine noise increments. This has the advantage that any differences are only from the effects of the time-step on the integration accuracy.

If different noises were used, part of the measured error-bar would be from sampling errors. Where there is 2D graphical output, the error bars give the step error, if you set $p.checks = 1$. The standard error-bar, with no extrapolation, has a half-size equal to the difference of the two most accurate results.

If computed, the discretization error is included in the graphical data outputs for all observables. It is accessed by setting the last index for the output data equal 2. The raw discretization error is generally a very cautious estimate, and may overestimate the errors. This estimate can be improved using extrapolation, explained next.

## 8.2 Higher order convergence

xSPDE uses extrapolation to improve convergence, which requires an input of the *order*. If this is non-zero, and *checks* are set to 1 to allow successive integration with two different step-sizes, the output of all data graphed will be extrapolated by assuming the method has the specified order.

### 8.2.1 Extrapolation

Extrapolation is valuable for improving the accuracy of a differential equation solver. It is valid for small time-steps. Suppose an algorithm has a correct solution $R_0$, but returns a numerical result $R$ with an error order $n$. For small step-size, integration results $R(dt)$ with step-size $dt$ have an error of order $dt^n$, that is:

$$R(dt) = R_0 + e(R) = R_0 + k.dt^n. \tag{199}$$

Hence, from two results at different values of $dt$, differing by a factor of 2, one would obtain

$$\begin{aligned} R_1 &= R(dt) = R_0 + k.dt^n \\ R_2 &= R(2dt) = R_0 + 2^n k.dt^n. \end{aligned} \tag{200}$$

The true result, extrapolated to the small-step size limit, is obtained by giving more weight to the fine step-size result, while *subtracting* from this a correction due to the coarse step-size calculation, to cancel the leading error term:

$$R_0 = \frac{\left[R_1 - R_2 2^{-n}\right]}{\left[1 - 2^{-n}\right]}. \tag{201}$$

Thus, if we define a factor $\epsilon$ as

$$\epsilon(n) = \frac{1}{[2^n - 1]} = \left(1, \frac{1}{3}, \frac{1}{7} \ldots\right), \tag{202}$$

the true results are obtained from extrapolation to zero step-size as:

$$R_0 = (1 + \epsilon)R_1 - \epsilon R_2. \tag{203}$$

The built-in algorithms have an order as ordinary differential equation integrators of 1, 1, 2, 2, 2, 4 respectively and will converge to this order at small step-sizes. Weak first order convergence is always obtainable for these single noise-step SDE methods [30]. Second order convergence is obtained in some cases with midpoint, RK2 and RK4 algorithms.

Higher order convergence for the raw data is not guaranteed for the built-in SDE algorithms. The algorithms used do **not** always converge to the standard ODE order when used for stochastic equations. Hence extrapolation higher than first order should be used with caution in stochastic calculations, unless more complex methods are used [28].

### 8.2.2 Extrapolated error-bars

If extrapolation is used, the error bar half-size is the difference of the best raw estimate and the extrapolation. Extrapolated results are usually inside those given by the error-bars, however, note that:

- **extrapolation with too high an order may under-estimate error bars**

- **extrapolation with too low an order reduces the accuracy**

Hence, xSPDE assumes a default order of $order = 1$ for all SDE and SPDE cases. This gives an extrapolated weak order of 2 for stochastic cases. One can set $order = 0$ to remove the default, or use a higher order if preferred, although, as explained above, it requires some caution. For an ODE or PDE the default order is the usual deterministic order. For the default RK4 deterministic method, the default is $order = 4$. All orders are improved by one with extrapolation.

High-order convergence *without* extrapolation can also be obtained, either in special cases using the xSPDE methods, or by adding user-specified techniques. The xSPDE libraries can be readily extended by the user to include these, through defining a modified *method* function appropriately.

## 8.3 Statistical errors

Sampling error estimation in xSIM uses three different techniques.

- xSIM uses sub-ensemble averaging, requiring high-level ensembles.

- For probability estimates, a Poissonian sampling error is used, based on counts.

- If there is a comparison probability, this is used for sampling error estimates.

This procedure leads to reliable sampling error estimates, and makes efficient use of the vector instruction sets used by Matlab. Ensembles are specified in three levels. The first, *ensembles(1)*, is called the number of samples for brevity. All computed quantities returned by the **observe** functions are first averaged over the samples, which are calculated efficiently using a parallel vector of trajectories. By the central limit theorem, these low-level sample averages are distributed as a normal distribution at large sample number.

Next, the sample averages are averaged again over the two higher level ensembles, if specified. This time, the variance is accumulated. The variance of these distributions is used to estimate a standard deviation in the mean, since each computed quantity is now a normally distributed result. This method is applied to all the observables. The two lines generated represent $\bar{o} \pm \sigma(o)$, where $o$ is the observe function output, and $\sigma$ is the standard deviation in the mean.

Here, *ensembles(2)* specifies ensembles computed in series. The highest level ensemble, *ensembles(3),* is used for parallel simulations. This is faster for a multiple core CPU or when the codes are run in a supercomputing environment, which requires the Matlab parallel toolbox. Either type of high-level ensemble, or both together, can be used to calculate sampling errors.

If $ensembles(2) > 1$ or $ensembles(3) > 1$, which allows xSPDE to calculate sampling errors, it will plot upper and lower limits of one standard deviation. If the sampling errors are too large, try increasing $ensembles(1)$, which increases the trajectories in a single thread. An alternative is to increase $ensembles(2)$, which is slower, but is only limited by the compute time, or else to increase $ensembles(3)$, which gives higher level parallelization.

Each is limited in different ways: the first by memory, the second by time, the third by the number of cores. Sampling error control helps ensures accuracy.

### 8.3.1 Sampling error

Quantitative sampling error estimation in xSPDE uses sub-ensemble averaging. Ensembles are specified in three levels, using vector, serial and parallel methods, respectively. The vector ensemble length, *p.ensemble(1)*, is called the number of samples for brevity. All quantities returned by the observe functions are averaged over the samples, which are calculated efficiently using a vector of trajectories.

By the central limit theorem, the sample averages are distributed as a normal distribution at large sample number. Next, the sample averages are averaged over the two higher level ensembles, if specified. The variance of this data is used to estimate a standard deviation in the mean, since each is normally distributed.

The highest level ensemble, $p.ensemble(3)$, is used for parallel simulations. This requires the Matlab parallel toolbox. Either type of high-level ensemble, or both together, can be used to calculate sampling errors.

Note that one standard deviation is not a strong bound; errors are expected to exceed this value in 32% of observed measurements. Another point to remember is that stochastic errors are often correlated, so that a group of points may all have similar errors due to statistical sampling.

The statistical error due to finite samples of trajectories is called the sampling error. The RMS value of the relative sampling error for each computed function, normalized by the maximum modulus of the observable, is printed out after each xSPDE simulation. If the expected comparison value is zero, the absolute value is given.

Averages over stochastic ensembles are the specialty of xSPDE, which requires specification of the ensemble size. A hierarchy of ensemble specifications in three levels allows maximum resource utilization, so that:

$$p.ensembles = [ensembles(1), ensembles(2), ensembles(3)].$$

The local ensemble, $ensembles(1)$, gives within-thread parallelism, allowing vector instruction use for single-core efficiency. The serial ensemble, $ensembles(2)$, gives the number of independent sub-ensembles of trajectories calculated serially.

The parallel ensemble, $ensembles(3)$, gives multi-core parallelism, and requires the Matlab parallel toolbox. This improves speed when there are multiple cores. One should optimally put $ensembles(3)$ equal to the available number of CPU cores.

The *total* number of stochastic trajectories or samples is

$$ensembles(1) \times ensembles(2) \times ensembles(3).$$

Either *ensembles*(2) or *ensembles*(3) are required if sampling error-bars are to be calculated, owing to the sub-ensemble averaging method used in xSPDE to calculate sampling errors accurately.

Two lines are graphed for an upper and lower standard deviation departure from the mean. This is only plotted if the total number of serial or parallel ensembles is greater than one, preferably at least 10–20 to give reliable estimates. The sampling error is reasonably accurate, but may underestimate errors for unusual distributions. These estimates are available for all observables in any dimension. The two lines generated in the graphs represent $\bar{o} \pm \sigma$, where $o$ is the mean output, and $\sigma$ is the computed standard deviation in the mean.

## 8.4 Convergence tests

### 8.4.1 Comparisons: *compare*

Every *observe* function can be accompanied by a comparison function, with a function handle $compare\{n\}$. This generates a vector of analytic solutions or experimental data-points which is compared to the average of the stochastic results. Results are plotted as additional lines on the two-dimensional graphical outputs, and a summary of comparison differences is printed.

*A* cell array of functions is used to obtain comparison results. These are calculated from the user-specified **compare{n}(p)** handle where the function argument is the parameter structure p, giving a extra dashed line on the two-dimensional graphs. Other graphics options are available as well. These optional comparisons can be input in all dimensions. When there are error estimates, a chi-squared test is carried out to determine if the difference is within the expected step-size and sampling error bars. If the comparison has errors, for example from experimental data, the chi-squared test will include the experimental errors.

### 8.4.2 Convergence: *xcheck*

The convergence checker, *xcheck(checks,p),* is designed for use where there are analytic results available for comparisons. This will automatically run xSIM a total of *checks* times, increasing the initial *steps* by 2 after each run, to reduce the step-size by 2. It then runs xGRAPH to display the most accurate result. It prints the time-step, the maximum difference with an input *compare* and the estimated errors found at the relevant point.

**Exercise**

- **Simulate the Kubo oscillator using the file,** *Kubocheck.m,* **with xcheck.**

```
function [e] = Kubocheck()
p.name = 'Kubo with convergence checks';
p.ensembles = [1000,10];
p.initial = @(w,p) 1;
p.range = 2;
p.deriv = @(a,xi,p) 1i*xi.*a;
p.observe{1} = @(a,p) real(a(1,:));
p.observe{2} = @(a,p) a(1,:).*conj(a(1,:));
p.olabels = {'<a> ','< a^2> '};
p.xlabels = {'\tau'};
p.compare{1} = @(p) exp(-p.t/2);
p.compare{2} = @(p) 1;
e = xcheck(2,p);
end
```

## 8.5   Chi-squared estimates

Chi-squared error estimates are reported in cases that have statistical sampling errors and comparison functions. These allow estimates of goodness of fit for probabilities. For $N_p$ independent points graphed or measured, if $O_i$ is an observable with measured mean $\bar{O}_i$ and statistical fluctuations $\Delta O_i$, one has that:

$$\chi^2/N_p = \frac{1}{N_p} \sum_i \frac{\left\langle \left[ (\bar{O}_i + \Delta O_i) - O_i^a \right]^2 \right\rangle}{\sigma_i^2} \tag{204}$$

Here $\sigma_i^2$ is an estimated variance. Provided that $\left\langle \Delta O_i^2 \right\rangle = \sigma_i^2$ and $\bar{O}_i = O_i^a$, one should obtain the expected result of $\chi^2/N_p \approx 1$. The exact distribution is known in special cases, but this requires that all data is independent and has a Gaussian distribution, which is not the case for stochastic trajectories.

Because of the variety of error-sources, and the lack of independence from point to point, these error sums are not identical to Pearson's original definition of $\chi^2$, and therefore should be used with caution. Nevertheless, the definition provides a way of evaluating goodness of fit that is useful.

Here, the value of $\sigma_i^2$ is obtained by including *all* known error sources, so

$$\sigma_i^2 = \sum_{n=1}^{4} \left( \sigma_i^{(n)} \right)^2. \tag{205}$$

These are:

1. If sub-ensemble measures are used, the estimated $\sigma_i^2$ includes sampling errors.

2. If checks are included, the estimated $\sigma_i^2$ includes discretization errors.

3. If comparisons have systematic errors, e.g. from experimental data,

4. If comparisons have known statistical errors.

In the case of sampled probabilities where there is a comparison probability, the estimated statistical variance in the data is obtained following Pearson's original method. That is, from *estimated counts given the comparison probability*, rather than a computed variance. This allows the use of standard $\chi^2$ comparisons.

### 8.5.1 Probability comparisons

Comparisons of trajectory probabilities and analytic probabilities do not always result in perfect agreement. This is because the limitations of memory and simulation time mean that trajectories have to be binned, which leads to an additional discretization error. Note that xSPDE approximates the comparison analytic probability of a bin by the central bin value of the probability, which is the simplest procedure.

To explain this, comparisons of probabilities ought to use the average probability density over the bin, which is different from the central value. Suppose one has a comparison distribution $p^a(x)$. Using Simpson's rule, the average analytic probability density integrated over a bin size $\Delta x$ is approximately:

$$
p_o^a = \frac{1}{\Delta x} \int_{x_0 - \Delta x/2}^{x_0 + \Delta x/2} p^a(x) dx \tag{206}
$$
$$
\approx \frac{1}{6} \left[ 4p^a(x_0) + p^a\left(x_0 + \frac{\Delta x}{2}\right) + p^a\left(x_0 - \frac{\Delta x}{2}\right). \right]
$$

This is equivalent to a cubic polynomial fit. It can be used to improve the analytic binning comparisons. It is especially important for multi-dimensional comparisons. It results in 9 distinct terms for two dimensions. This correction must be inserted manually in the comparison functions.

### 8.5.2 Scaling of $\chi^2$ errors

Because chi-squared probability tests are sensitive, it helps to understand how they scale with bin-size. With $N_s$ total samples, the estimated probability $P_i$ in a bin with probability density $p(\mathbf{a})$ and sampled counts of $N_i$ is given by $P_i = N_i/N_s = p_i A$ for a bin $b_i$ with area $A$, where:

$$
p_i = \frac{1}{A} \int_{b_i} p(\mathbf{a}) dA \tag{207}
$$

The Poissonian variance of the counts in the bin is $\langle \Delta N_i \rangle = \langle N_i \rangle$. The expected probability variance is therefore

$$
\left\langle \Delta P^2 \right\rangle = \left\langle \Delta N_i^2 / N_s^2 \right\rangle = \langle N_i \rangle / N_s^2. \tag{208}
$$

Let $\langle N_i \rangle = N_i^a$, the analytic or expected count number. The expected probability density variance at a point is therefore

$$
\left\langle \Delta p_i^2 \right\rangle = \left\langle \Delta N_i^2 / A^2 N_s^2 \right\rangle = N_i^a / A^2 N_s^2 = p_i^a / A N_s. \tag{209}
$$

Here $p_i^a$ is the analytic or comparison probability density, and $\left\langle \Delta p_i^2 \right\rangle^a = p_i^a / A N_s$ is the expected analytic variance. The $\chi^2$ variable, that follows the Pearson $\chi^2$ distribution, is defined as follows:

$$
\chi^2 / N_p = \frac{1}{N_p} \sum_i \frac{\left\langle \left[ p_i - p_i^a \right]^2 \right\rangle}{\left\langle \Delta p_i^2 \right\rangle} \tag{210}
$$

Here, $p_i^a$ is obtained by integrating over the $i$-th probability bin. It can be estimated by using the central value, $p_i^a \approx p(\mathbf{a}_i)$, although cubic interpolation is more precise.

This could lead to a fixed error in the analytic probability density $p_i^a$, so $p_i^a \to p_i^a + \epsilon_i$, possibly localized to some fraction of bins $f$ which may change with the bin size. Suppose, for simplicity, that $\epsilon$ is due to an integration error in integrating the exact distribution or any other error in the 'exact' distribution, and it does not change with changes to the bin area $A$.

From the definition of $\chi^2$, if the generated samples have negligible step-size errors:

$$\chi^2/N_p = \frac{1}{N_p} \sum_i \frac{\left\langle \left[ \left( p_i^a + \Delta p_i \right) - p_i^a - \epsilon_i \right]^2 \right\rangle}{\left\langle \Delta p_i^2 \right\rangle} \tag{211}$$

For simplicity, if we consider the large sample limit with uniform probabilities,

$$\chi^2/N_p = 1 + \frac{f\epsilon^2}{\langle \Delta p^2 \rangle} = 1 + \frac{f\epsilon^2 A N_s}{p^a} \tag{212}$$

Increasing the bin area $A$ will increase $\chi^2/N_p$ above its usual value of *1* by an amount proportional to $A$. This is simply because smaller bins have less intrinsic accuracy, due to a larger sampling error. As a result, it is often preferable to use more accurate probability estimates with larger bins having more counts, since these are much more sensitive to effects like this.

Often, simulated and comparison graphs may appear identical visually, but even if they have small errors they may still be very significant. Such comparison binning errors can be reduced by using cubic spline interpolations, as explained above.

## 8.6 Error outputs

There are six types of data outputs: data, errors, comparisons and the comparison errors. Summaries will appear in the printed outputs if available. Step errors and sampling errors, as well as comparison data are stored in all the output data arrays. These are also available graphically in two-dimensional graphs.

### 8.6.1 Numerical error outputs

The last data index $c$ is used to obtain errors and comparisons in data outputs. To obtain comparison data, a comparison function is defined for each output function. This can include, for example, experimental data, experimental errors or exact analytic comparisons where they are available.

1. Means are in $c = 1$ data, except if *scatters>1,* which gives individual trajectories.

2. If *checks>0*, the step errors are in $c = 2$ data.

3. If $ensembles(2,3) > 1$, the sampling errors are in $c = 3$ data.

4. Comparison values from *compare* functions are in $c = 4$ data.

5. Comparison systematic errors can be included in $c = 5$ data.

6. Comparison statistical errors can be included in $c = 6$ data.

### 8.6.2   Graphical error outputs

These are explained in detail in the xGRAPH reference section 10.

1. Mean values or trajectories are graphed as separate data lines.

2. Step errors generate graph error bars

3. Sampling errors are graphed as parallel solid lines

4. Dashed lines indicate comparison values from *compare* functions.

5. Comparison systematic errors give additional error bars

6. Comparison statistical errors can be included as parallel lines

Because multiple errors can generate very complex graphs, there is additional control of error bar generation, explained in the xGRAPH reference section. One can also obtain difference graphs with comparisons, which allow errors to be examined more closely, and error bars can be combined in different ways.

Graphics data is only available for two-dimensional graphs, and is subject to selection using the *axes* inputs.

### 8.6.3   Printed error outputs

Printed error summaries are generated in each xSIM run, in addition to the data outputs. These are normalized, root mean square (RMS) errors. Normalization is carried out using the modulus of the largest data value. If the comparison results are all zero for a function, there is no normalization carried out.

After computing RMS values over each graph function, a second RMS average is taken over all totals, weighting each total equally, and including all functions and sequence datasets where there are nonzero errors reported. Data with no errors are not included in the totals for each category.

There is a final RMS average taken over the step, sampling and comparison totals. This again ignores categories with no errors. The purpose is to allow a rapid comparison to ensure that there are no higher than expected errors, which might require a new simulation with more steps or increased trajectories.

Printed errors are summarized in three main categories

1. Discretization or step errors

2. Sampling errors

3. Comparison errors

Comparison data may not be available over an entire lattice. If this is the case, the *axes* point selections can be used to restrict the relevant datas points used for these comparisons. This also applies to the goodness of fit and error-vector outputs, since they make use of comparison data where it is available.

### 8.6.4  Goodness of fit ($\chi^2$)

The $\chi^2$ statistics are obtained by normalizing the comparison squared differences by the sum of squares of all the data and comparison errors at that point. These are summed over every data point with relevant data, and the number of relevant data points, $k$, is stored. The ratio of $\chi^2/k$ should be order 1 for statistical errors.

These are summarized for each functional data output type, as well as giving rise to an error total.

### 8.6.5  Error vector output

When used as a function call in batch mode, the first type of data returned by xSIM is a six-component error vector. This can be used for summarizing error data in a batch job, to determine if a specified error-threshold is reached, to allow an iterative increase in the number of time-steps or trajectories.

The error-vector components are:

1. Total error overall, including step, discretization and comparisons

2. Total step-size error

3. Total sampling error

4. Total comparison error

5. Total $\chi^2/k$ goodness of fit

6. Simulation elapsed time

### 8.6.6  Error summaries

There are six types of data outputs: data, errors, comparisons and comparison errors. Summaries will appear in the printed outputs, depending on the verbosity setting. Step errors and sampling errors, as well as comparison data are stored in output data arrays. These are also available graphically in two-dimensional graphs.

# 9    xSIM reference

*This section gives a reference guide to the xSIM parameters and functions.*

## 9.1    Overview

Simulations carried out by xSIM are performed by other specialized internal functions. Input parameters come from an **input** cell array of structures, while output is saved in a **data** array, and optionally in a file. During the simulation, global averages and error-bars are calculated for time-step and sampling errors. When completed, timing and errors are printed.

The *xsim* function call syntax is: *[error,data,output(,rawdata)] = xsim(input);*

### 9.1.1    Input and data structures

To explain xSPDE in full detail,

- Simulation parameters are stored in the **input** cell array.

- This describes a sequence of parameter structures, so that **input={p1,p2,...}**.

- Each structure **p1,p2,...** generates an output which is the input of the next.

- The main simulation function is called using **xsim(input).**

- The RMS errors and integration time are returned in the **error** vector

- Parameters including defaults are returned in the **output** cell array.

- Averages are recorded sequentially in the **data** cell array.

- Raw trajectory data is optionally stored in the **rawdata** cell array.

The sequence *input* defines a sequence of individual simulations, with parameters that specify the simulation functions and give the equations and observables. If there is only one simulation, just one data structure is needed, without a cell array. In addition, xSPDE can generates graphs with its own graphics program, xGRAPH.

For convergence checking, a useful alternative to xspde which repeats the calculation *checks* times while halving the time-step each time, and reports the resulting errors for averaged observables, is:

- xcheck (checks,p)

### 9.1.2    Parameters and functions

The xSIM input objects include parameters and functions, with an extensible object-oriented architecture. All xSIM functions are modular and replaceable. In many cases this is as easy as just defining a new function handle to replace the default value.

There are two types of functions:

- *User* functions define the simulation, and have default values. The defaults are usually obtained by adding 'x' in front of the name. In the special case of *method*, the default depends on the problem.

- *Helper* functions usually start with 'x'. In some cases these are defaults for user functions. In all cases they have well-defined roles, like the reserved functions in C, Python, Matlab or Julia.

- All arguments in square brackets are optional, but may be needed only in specific cases.

- The last argument, *p,* is the parameter structure.

For example, to define your own integration function, include in the xSPDE/xSIM input the line:

```
p.method = @Mystep;
```

Next, include anywhere on your Matlab path the function definition, for example:

```
function a = Mystep(a,w,p)
% a = Mystep(a,w,p) propagates a step my way.
..
a = ...;
end
```

## 9.2 Parameter table

Simulation parameters are stored in a parameter structure which is passed to the *xsim* program. Constants can be included, but must not be reserved names. Names starting with a capital letter like 'A...' - except the reserved '*D*' for derivatives - are always available. Globals are incompatible with the Matlab parallel toolbox. Graphics data is stored for the graphics program to use.

Standard inputs have default values, which are user-modifiable through the *xpreferences* function. Defaults can be checked by including the input $verbose = 2$. All the inputs are part of a structure passed to xSPDE. If a cell array of multiple structures are input, these are executed in sequence, with the output of the first simulation passed to the second, then the third, and so on.

Library functions inputs do not have defaults, as these are subject to change.

| Label | Default value | Description |
|---|---|---|
| *version* | *'xSIM3.xx'* | Current version number |
| *name* | *"* | Simulation name |
| *dimensions* | 1 | Space-time dimensions |
| *fields* | 1 | Total number of stochastic fields |
| *backfields* | 0 | Number of backward fields |
| *auxfields* | 0 | Number of auxiliary fields |
| *ranges* | *[10,..]* | Range of coordinates in *[t,x,y,z,..]* |
| *origins* | *[0,..]* | Origin of coordinates in *[t,x,y,z,..]* |
| *points* | *[51,...* | Output lattice points in *[t,x,y,z,..]* |
| *noises* | *[1, 0]* | Number of noise fields in *[x,k]* |
| *inrandoms* | *[1, 0]* | Initial random fields in *[x,k]* |
| *ensembles* | *[1, 1, 1]* | Size of *[vector, serial, parallel]* ensembles |
| *steps* | *1* | Integration steps per output point |
| *iterations* | *4* | Maximum implicit or midpoint iterations |
| *order* | *1* | Extrapolation order: *0,1,2,..* |
| *checks* | *1* | Check time-step errors: 0 or 1 |
| *seed* | *0* | Seed for random number generator |
| *file* | *"* | File-name: *'f.mat'* = Matlab, *'f.h5'* = HDF5 |
| *boundaries{n}* | $[0,0;0,0]$ | Boundary: '-1,0,1'=Neum, periodic, Dirichlet boundary. |
| *binranges{n}* | *{0,0,...}* | Observable binning ranges for probabilities |
| *cutoffs{n}* | 0 | Lower graph cutoff for chi-squared estimates |
| *mincount* | 0 | Lower count cutoff for chi-squared estimates |
| *ipsteps* | 1 | IP transforms per time-step |
| *numberaxis* | 0 | If 1, forces use of numerical axis labels |
| *verbose* | 0 | 0 for brief, 1 for informative, 2 for full output |
| $A, B, C, \ldots$ | - | User specified static parameters |
| *olabels* | *{'a_1',..}* | Observable labels |
| *transforms* | *{[0 0 0 0],..}* | Fourier transforms in [t,x,y,z,..] per observable |
| *rawdata* | 0 | Raw data switch: 1 for raw output |
| *scatters* | *{0,..}* | Specify to obtain scatter plots, not averages |
| *octave* | *0* | Force octave syntax: 1 for octave |
| *thresholdw* | 0 | Threshold for weighted simulation breeding |
| *iterfb* | 2 | Iterations of forward-backward algorithm |
| *iterproj* | 2 | Iterations of projector algorithm |
| *qcproj* | - | Quadratic projection coefficients |
| *vcproj* | - | Vector projection coefficients |

## 9.3 Function tables

### 9.3.1 User function table

The user-defined functions, calling arguments, and purpose, are:

| Label | Arguments | Purpose |
|-------|-----------|---------|
| *deriv* | $(a, [a_-,] w, p)$ | Stochastic derivative |
| *initial* | $(r, p)$ | Function to initialize fields |
| *linear* | $(p)$ | Linear derivative function |
| *rfilter* | $(r, p)$ | Random filter function in k-space |
| *nfilter* | $(w, p)$ | Noise filter function in k-space |
| *transfer* | $(a, p, a_0, p_0)$ | Transfer inside a sequence |
| *method* | $(a, w, p)$ | Algorithm defining a time-step |
| *grid* | $(p)$ | Grid calculator for the lattice |
| *prop* | $(a, p)$ | Interaction picture propagator |
| *propfactor* | $(nc, p)$ | Propagator array calculation |
| *observe* | $(a, p)$ | Observable function cell array |
| *function* | $(o, p)$ | Functions of average observables |
| *compare* | $(p)$ | Comparisons, for differences and $\chi^2$ |
| *define* | $(a, w, p)$ | Defines an auxiliary field value |
| *randomgen* | $(p)$ | Initial random generator |
| *noisegen* | $(p)$ | Noise generator |
| *rfilter* | $(v, p)$ | Initial random kspace filter |
| *nfilter* | $(w, p)$ | Noise kspace filter |
| *project* | $(d, a, n, p)$ | Defines the projection |
| *firstfb* | $(a0, nc, p)$ | First forward-backward path |

### 9.3.2  Internal function table

For details of the internal functions, see section 9.6 and sections 7 and 7. All xSPDE internal functions are capitalized. They are:

| Label | Arguments | Purpose |
|---|---|---|
| *Ave* | $(a, [av,] p)$ | Averages over a spatial lattice |
| *Bin* | $(a, [dx,] p)$ | Bins results onto an axis |
| *Catproj* | $(d, a, n, p)$ | Catenoid projector |
| *D1* | $(a, [dir,] p)$ | First derivative |
| *D2* | $(a, [dir,] p)$ | Second derivative |
| *Euler* | $(a, w, p)$ | Euler algorithm |
| *Implicit* | $(a, w, p)$ | Implicit Euler algorithm |
| *MP* | $(a, w, p)$ | Midpoint algorithm |
| *MPadapt* | $(a, w, p)$ | Midpoint adaptive algorithm |
| *RK2* | $(a, w, p)$ | Runge-Kutta (2) algorithm |
| *RK4* | $(a, w, p)$ | Runge-Kutta (4) algorithm |
| *MPfb* | $(a, w, p)$ | Midpoint forward-backward algorithm |
| *Quadproj* | $(d, a, n, p)$ | General quadratic projector |
| *Polproj* | $(d, a, n, p)$ | Diagonal polynomial projector |
| *Int* | $(a, [dx \, or \, dk], p)$ | Integrates over space or momentum |
| *Enproj* | $(a, w, p)$ | Euler normal projection algorithm |
| *MPproj* | $(a, w, p)$ | Midpoint projection algorithm |
| *MPnproj* | $(a, w, p)$ | Midpoint normal projection algorithm |

- Projection algorithms with a '*proj*' suffix require a *project* function.

- Forward-backward algorithms with an '*fb*' suffix require a second field in the user deriv function.

- For *Int*, one can integrate either with respect to $dx$ or $dk$, in either ordinary space or momentum space, by changing the second argument passed to $xint$ as required.

- For integration in momentum space, fields that are passed to *Int* are transformed if the *observe* function is used with Fourier transforms selected using *transforms*.

- For integrating functions like *function{n}* with transforms, the transform flags *transforms{n}* should be used both for the function and any *observe* averages used.

- Average data is *not* Fourier transformed after averaging. If this is required, it is best to output the data first.

## 9.4   Parameter reference

### 9.4.1   *auxfields*

**Default:** *0*

These are real or complex auxiliary fields stored at each lattice point, specified using *define*. They are useful for input/output spectral calculations, and can be functions of the noise.

**Example:** *p.auxfields = 2*

### 9.4.2  *axes{n}*

**Default:**  *{0,0,0,..}*

Gives the axis points used for comparisons in the *n*-th output function, in each dimension. For each function, the axes can be individually specified in each dimension. Each entry value is a vector range for a particular dimension, for *d=1,...p.dimension*s. Thus, *5* gives the fifth point only in that dimension, and an input *1:4:41* plots every fourth point. Zero or negative values are shorthand: *-1* generates a default point at the midpoint, *-2* the endpoint, and *0* is the default value that gives the vector for the every axis point. This data is also used to control graphics outputs. It can be input separately for the graphs if required.

**Example:**  *p.axes{4} = {1:2:10,0,0,-1}*

### 9.4.3  *backfields*

**Default:**  *0*

The optional input **backfields** is the number of backward-time stochastic fields that are integrated, as part of the overall vector of integrated *fields* components. Requires a forward-backward method like *MPfb*.

**Example:**  *p.backfields = 2*

### 9.4.4  *binranges{n}*

**Default:**  *{}*

Nested cell array, $binranges\{n\}\{m\}$, that defines the probability plotted for observable $n$. If null or zero, the mean of the observable is calculated as usual. The second cell index, $m = 1,\ldots M$, corresponds to the line index returned by the corresponding $n$-th *observe* function. When nonzero, the probability of the $n$-th observable is calculated and plotted according to the specified vector of axis points. This sets extra dimensions in the data, depending on the range of $m$ values, with $[o_1, o_2, \ldots o_K]$, being the start and end of each of the bins used to accumulate probabilities. The $k-th$ bin is centered at $(o_k + o_{k+1})/2$. In this version of xSPDE, each bin must have the same width for an observable and line number. The output is the average probability density versus the (vector) value of the observable. Hence $M$ extra output dimensions are added to the generated probability data.

**Example:**  *p.binranges{n}{1} = {-5:0.1:5,-2:0.1:2}*

### 9.4.5  *boundaries{dir}*

**Default:**  *[0, 0]*

Cell array for type of spatial boundary conditions used, set for each dimension and field component independently, and used in the equation solutions. The cell index is $dir = 2, 3, ..$, indicating the dimension. The boundary conditions are defined as a matrix. The first index is the field index $i$ and the second index the boundary $j$, with $j = 1$ for the lower and $j = 2$ for the upper boundary. The options are $b = -1, 0, 1$.

- The default option, or 0, is periodic.

- If -1, Robin/Neumann boundaries are used, with derivatives set to prescribed values.

- If 1, Dirichlet boundaries are used, with fields set to prescribed values.

In the current code, only default boundaries are available using spectral (*linear*) methods. Using arbitrary non-periodic boundaries *requires* the use of finite difference derivatives, without the option of an interaction picture derivative. In such general cases, arbitrary boundary values are set by *boundfun(a,d,p)*.

**Example:** *p.boundaries{d} = [-1,1;0,0;1,-1]*

### 9.4.6   *c...*

The starting letter *c* is always reserved to store user-specified constants and parameters. It is passed to user functions and can be any data. All inputs — including *c* data — are copied into the stored data files via the lattice structure p, to give a permanent record of simulation parameter values along with the output data.

**Example:** *p.constant = 2\*pi*

### 9.4.7   *checks*

**Default:** *1*

This defines if a repeat integration is carried out for error-checking purposes. If *p.checks = 0*, there is one integration, with no checking at smaller time-steps. For error checking, set *p.checks = 1*, which repeats the calculation at half the time-step — but with identical noise — to obtain error bars. This is the default value, taking three times longer overall, but with increased accuracy and error-estimates.

Also see the *order* parameter, below.

**Example:** *p.checks = 0*

### 9.4.8   *dimensions*

**Default:** *1*

This is the space-time dimension for an SPDE. If omitted, *dimensions=1*, giving an SDE. It is arbitrary apart from the obvious memory requirements at large dimensionality.

**Example:** *p.dimensions = 4*

### 9.4.9   *ensembles*

**Default:** *[1, 1, 1]*

Number of independent stochastic trajectories simulated. This has three levels to maximize efficiency. The first is within-thread parallelism, allowing vector instructions. The second gives a number of independent trajectories calculated serially. The third gives multi-core parallelism and requires the Matlab parallel toolbox. Either *p.ensembles(2)* or *p.ensembles(3)* are required to obtain sampling error-bars. The total number of stochastic trajectories or samples is *ensembles*(1)×*ensembles*(2)×*ensemble* The second and third *ensembles* cannot be changed during a sequence of simulations.

**Example:** *p.ensembles = [1000,100,10]*

### 9.4.10 *fields*

**Default:** *1*

These are real or complex variables stored at each lattice point that are the independent variables for integration. The fields are vectors that can have any number of components and any number of dimensions. The *fields* input is the number of real or complex components that are initialized by the *initial* function and integrated using the deriv derivative.

**Example:** *p.fields = 2*

### 9.4.11 *file*

**Default:** ' '

Matlab or HDF5 file name for output data. Includes all data and parameter values, including raw trajectories if $p.rawdata = 1$. If not needed just omit this. A Matlab filename should end in .mat, while an HDF5 file requires the filename to end in .h5. For a sequence of inputs, the filename should be given in the first structure of the sequence, and the entire sequence is stored. This cannot be changed for successive parts of the overall sequence.

**Example:** *p.file = 'file-name'*

### 9.4.12 *functions*

**Default:** number of defined functions or observables

This gives the maximum number of output datasets which are functions of the observables. The default number of functional transformations is the greater of the length of the cell arrays of *observe* and *function* definitions. Normally, this is not initialized, as the default is typically used, unless one wishes to reduce the data output without changing an input script.

**Example:** *p.functions = 1*

### 9.4.13 *ipsteps*

**Default:** 1 for *Euler, Implicit* and *RK2*; 2 for *MP, MPadapt* and *RK4; 0 otherwise*

This specifies the number of interaction picture time-steps needed in an integration time-step. Default values are chosen according to the setting of *method*. Can be changed for custom integration methods. This must be initialized if a non-standard integration method is used that requires an interaction picture as well.

**Example:** *p.ipsteps = 1*

### 9.4.14 *iterations*

**Default:** 4

For iterative algorithms like the implicit midpoint method, the iteration count is set here, typically around 3-4. Will increase the integration accuracy if set higher, but it may be better to increase steps if this is needed. With non-iterated algorithms, this input is not used:

**Example:** *p.iterations = 3*

### 9.4.15 *name*

**Default:** ''

Name used to label simulation, usually corresponding to the equation or problem solved. This can be removed from graphs using *headers* equal to a single blank space when running *xgraph*.

**Example:** *p.name = 'your project name'*

### 9.4.16 *noises*

**Default:** *fields (1)*

This gives the number of stochastic noises generated per lattice point, in coordinate and momentum space, respectively. Set to zero (*noises* = 0) for no noises. This is the number of rows in the noise-vector. Noises can be delta-correlated in x-space or in k-space. The second input is the dimension of noises in k-space. It can be omitted if zero. This allows use of finite correlation lengths, by including a frequency filter function that is used to modify the noise in Fourier-space. The Fourier-space random variance is defined by the filter function. This takes the noises in Fourier space and returns a filtered version, which is inverse Fourier transformed before use. The first noise index, *noises*(1), indicates how many independent noise fields are generated, while *noises*(2) indicates how many noises are Fourier-transformed, filtered and then inverse Fourier transformed to give correlations. These are extra noises, so the total is *noises*(1) + *noises*(2). Filtered noises have a finite correlation length.

**Example:** *p.noises = [2,4]*.

### 9.4.17 *order*

**Default:** 1 for *ensembles* $\neq$ 1, otherwise the deterministic order.

This is the extrapolation order, which is only used if *p.checks* = 1. The program uses the estimated convergence order to extrapolate to zero step-size, with reduced errors. If *p.order = 0*, no extrapolation is used, which is the most conservative input. The default order is usually acceptable, especially when combined with the default midpoint algorithm.

The extrapolation order cannot be changed during a sequence. The default deterministic orders of the six preset methods used *without* stochastic ensembles are:

**1** for *Euler* and *Implicit;*

**2** for *RK2*, *MP* and *MPadapt*;

**4** for *RK4*.

**Example:** *p.order = 0*

### 9.4.18 *origins*

**Default:** *[0, -p.ranges/2]*

This displaces the graph origin for each simulation to a user-defined value. If omitted, all initial times in a sequence are zero, and the space origin is set to *-p.ranges/2* to give results that are symmetric about the origin. As an example, for the x-dimension, the problem is solved on an interval of $x = [O_2, O_2 + R_2]$, with a default origin of $-R_2/2$, so that $x = [-R_2/2, R_2/2]$.

**Example:** p.origins $= [0,-20,-20]$

### 9.4.19 *points*

**Default:** *[51, 35, ..., 35]*

The rectangular lattice of points plotted for each dimension are defined by a vector giving the number of points in each dimension. The default values are given as a rough guide for initial calculations. Large, high dimensional lattices take more time to integrate. Increasing points improves graphics resolution and gives better accuracy in each relevant dimension as well, but requires more memory. Speed is improved when the lattice points are a product of small prime factors. In order to discretize the problem, the $p_i$ lattice *points* are fitted into the range $R_i$ so that $dx_i = R_i/(p_i-1)$, ie:

$$x_i = O_i + (i-1)dx_i. \tag{213}$$

**Example:** *p.points = [30,40,40]*

### 9.4.20 *inrandoms*

**Default:** noises

This gives the number of initial random fields generated per lattice point in coordinate and momentum space. Set to zero ($p.inrandoms = 0$) for no random fields. Random fields can be delta-correlated in x-space or in k-space. The second input is the dimension of random fields that are delta-correlated in momentum space. It can be left out if zero. The Fourier-space random variance is modified by the filter function. This takes the initial random fields in Fourier space and returns a filtered version, which is inverse Fourier transformed before use. The first noise index, *p.inrandoms(1)*, indicates how many independent random fields delta-correlated in space are generated, while *p.inrandoms(2)* indicates how many additional random fields are Fourier-transformed, filtered and then inverse Fourier transformed. These are additional random fields, so the total is *p.inrandoms(1)+p.inrandoms(2)*. The filtered random inputs have a finite correlation length.

**Example:** *p.inrandoms = [2, 0]*

### 9.4.21 *ranges*

**Default:** *[10, 10, ...]*

Each lattice dimension has a coordinate range. The default value is 10 in each dimension. In the temporal graphs, the first coordinate is plotted over $0 : p.ranges(1)$. All other coordinates are plotted over $-p.ranges(n)/2 : p.ranges(n)/2$. The starting value can be changed using the *origins* variable.

**Example:** *p.ranges = [1, 10]*

### 9.4.22  *rawdata*

**Default:** *0*

Flag for storing raw trajectory data. If this flag is turned on, raw trajectories are stored in memory. The raw data is returned in function calls and also written to a file on completion, if a file-name is included.

**Example:** *p.rawdata = 1*

### 9.4.23  *scatters{n}*

**Default:** 0

Cell array that defines the number of scatter trajectories plotted for observable $n$. If absent or zero, the mean of the observable is calculated as usual. If nonzero, a set of $s$ observables that correspond to independent stochastic fields are accumulated, with no averaging. This cannot be combined with probabilities or with parallel ensembles. There must be at least $s$ trajectories in *ensembles(1)*, otherwise the number of stored trajectories is reduced.

**Example:** *p.scatters{n} = 20*

### 9.4.24  *seed*

**Default:** *0*

Random noise generation seed, for obtaining reproducible noise sequences. Set to unique and distinct values for the different parallel ensembles. Used if $p.noises > 0$ or $p.inrandoms > 0$.

**Example:** *p.seed = 42*

### 9.4.25  *steps*

**Default:** *1*

Number of time-steps per plotted point. The total number of integration time-steps in a simulation is therefore *p.steps×(p.points(1)-1)*. Thus, steps can be increased to improve the accuracy, but gives no change in graphics resolution. Increasing the steps will give a lower time-discretization error.

**Example:** *p.steps = 1, 2, ...*

### 9.4.26 *transforms{n}*

**Default:** *[0,0,..]*

Cell array defining the Fourier transforms used for an observable $n$. There is one transform vector per observable. The $j$-th flag, *tr(j)*, indicates a Fourier transform on the $j$-th axis if set to one, starting with the time axis. The default value is zero, indicating no transform. The normalization of the Fourier transform is such that the $k = 0$ value in momentum space corresponds to the integral over space with a factor of $1/\sqrt{2\pi}$ in each transformed dimension. The Fourier transform that is graphed has $k = 0$ as the central value. The default is no Fourier transform. Must be set for any functional transform of a Fourier observable, to give the correct graph axes.

**Example:** *p.transforms{n} = [1,0,0,1]*

### 9.4.27 *verbose*

**Default:** 0

Print flag for output information while running xSIM. Print options are:

- Minimal if *verbose = -1*: Prints just the start-up time and hard error messages

- Brief if *verbose = 0*: Additionally prints the final, total integration errors

- Informative if *verbose = 1*: Also prints the individual function RMS errors and progress indicators

- Full if *verbose = 2*: Prints everything, including the internal parameter structure data.

In summary, if *verbose = 0*, most output is suppressed except the final data, while *verbose = 1* displays a progress report, and *verbose = 2* additionally generates a readable summary of the parameter input as a record.

**Example:** *p.verbose = 2*

### 9.4.28 *version*

**Default:** *'xSIM3.44'*

Sets the current version number of the simulation program. There is no need to input this except for project documentation for a customized version.

**Example:** *p.version = 'current version name'*

## 9.5 Function reference

### 9.5.1 User function reference

The following function inputs define the differential equation that is integrated or solved. They are specified in an xSPDE/xSIM input file using *p.(fun) = @(Myfun)*, either as inline or externally defined functions. Externally defined functions must be in the same file as the input parameters, or on the execution path.

### 9.5.2 *boundfun(a, d, p)*

**Default:** *xboundfun()*

The boundary function *boundfun(a,d,p)* is called for specified boundary conditions in the $d$-th dimension. This returns the boundary values used for the fields or their first derivatives in space dimension $d > 1$, as an array indexed as $b(f, \mathbf{i}, e)$ in the standard way. Here $f$ is the field index, $\mathbf{i} \equiv [j_2, \dots j_d]$ are the space indices, and $e$ is the ensemble index.

Only two values are needed for $j_d$, which is the index of the dimension whose boundary values are specified. These are $j_d = 1, 2$, for the lower and upper boundary values, which are either field values or derivatives. Boundary values may be constant or a function of the fields $a$ and space-time $t, \mathbf{x}$.

If boundary values have stochastic values which are calculated only once, they must be initialized. To allow for this, *boundfun(a,d,p)* is initially called with time $t = origins(1) - 1$, snd the input field $a$ set to random values from *randomgen,* which are independent of those that initialize the field at $t = origins(1)$.

They are reproducible for different *check* cycles, to allow noise-independent error-checking. The initial results for the boundaries are stored in an array *boundval{d}* for later use by *boundfun* if required.

The default boundary value is zero, set by the default boundary function *xboundfun(a,d,p)*.

### 9.5.3 *compare(p)*

**Default:** *compare{n}= []*

This is for comparisons to experimental or analytic data. The output is an array with $d + 2$ dimensions. The first dimension is the line index, the next $d$ dimensions are time and space, while the last index is an error index. This can have up to two additional entries for systematic and/or statistical error bars in the comparison data, from analytic or experimental results. Error-bars are optional if not available.

### 9.5.4 *define(a, w, p)*

**Default:** *xdefine()*

Calculates auxiliary fields, which are combinations of fields and noises. If used they are accessed in *observe* functions as $a(n, :)$, where $n > fields$. The default, *xdefine(),* sets the auxiliary fields to zero.

### 9.5.5 *deriv(a,w,p)*

**Default:** *deriv()= 0*

This defines the stochastic equation time derivative, given the current field $a$, delta-correlated noise terms $w,$ and parameters $p$. It is defined explicitly in (71). This is the right-hand-side of (1) or (57), *without* the linear term if it is specified separately.

### 9.5.6 *firstfb(a0,nc,p)*

**Default:** *xfirstfb()*

Returns the zero-th order field estimates in a forward-backward iteration. Here *nc* is the time-step check index. This is needed because the number of time-points to be initialized depends on *nc*. The default function is *xfirstfb*, which sets each field in either direction equal to its initial value at the time boundaries, given by *a*0. Other estimates may give faster convergence.

### 9.5.7 *function(o,p)*

**Default:** *xfunction{n}=@(o,p) o{n}*

This is a user-defined array of functions of the *observe* outputs after averaging over *ensembles(1)*, possibly involving combinations of several observed averages. The input to the *n*-th function is the cell array of all averages, and the output is the data for the *n*-th graph. This function is compatible with all error estimates. The default values generate all the *observe* averages that are in the data. The output data format is an array with $d + 1$ dimensions. The first dimension is the line index, the next $d$ dimensions are time and space.

### 9.5.8 grid

**default** *xgrid*

Calculates the spatial grid for specialized purposes like non-uniform grids. The default, *xgrid*, returns a homogeneous rectangular grid in both ordinary and momentum space.

### 9.5.9 *initial(rv, p)*

**Default:** *xinitial()*

This is used to initialize each integration in time. It is a user-defined function which can involve random numbers for an initial probability distribution. This creates a stochastic field on the lattice, called **a**. The returned first dimension should be *p.fields*. The initial Gaussian random field variable, *rv*, has unit variance if dimension is 1 or else is delta-correlated in space, with variance $1/p.dv = 1/(dx_2...dx_d))$ for $d$ space-time dimensions. If *inrandoms* is given in the input parameter structure, *rv* has a first dimension of *inrandoms(1) + inrandoms(2)*. If not specified, the default for *inrandoms* is *noises*. The default function is *xinitial,* which sets fields to zero, returning **a** $= 0$.

### 9.5.10 *linear(p)*

**Default:** *xlinear()*

A user-definable function for the linear response, which is a matrix for an SDE or ODE. For an SPDE or PDE, it includes transverse derivatives in space, returning the linear coefficients *L* in FFT/DST/DCT space, which are assumed diagonal in the field index. These are functions of differential terms *Dx, Dy, Dz*, which correspond to $\partial/\partial x, \partial/\partial y, \partial/\partial z$, respectively. Each component has a dimension the same as the coordinate lattice. For axes that are numbered, use *D{2}, D{3}* etc. The default, *xlinear,* sets *L* to zero.

### 9.5.11 *nfilter (w,p)*

**Default:** *xnfilter()*

Returns the momentum-space filter function for the propagation noise terms in momentum-space. Each component has an array dimension the same as the random noises in momentum space, that is, the return dimension is *[noises(2), d.lattice]*.

### 9.5.12 *noisegen(p)*

**Default:** *xnoisegen(p)*

Generates arrays of noise terms for each point in time. The default, *xnoisegen()* returns *noises(1) + noises(2)* Gaussian real noises that are delta-correlated in time, space and momentum space, unless *nfilter* is used to modify momentum space correlations.

### 9.5.13 *observe(a, p)*

**Default:** *xobserve{1}=@(a,p) a*

Cell array of function handles that take the current field and returns an observable o. Note the use of braces for cell arrays! One can input these as *p.observe{n} = @(a,p) f(a,p)*. An omitted function less than the maximum index is replaced by the default, which is the vector *a* of real field amplitudes.

### 9.5.14 *prop(a, p)*

**Default:** *xprop()*

Returns the fields propagated for one step in the interaction picture, given an initial field a, using the propagator array. The time-step used in propagator depends on the input time-step, the error-checking and the algorithm. The default, *xprop*, takes a Fourier transform of **a**, multiplies by propfactor to propagate in time, then takes an inverse Fourier transform.

### 9.5.15 *propfactor(nc, p)*

**Default:** *xpropfactor()*

Returns the interaction picture propagator used by the prop function. The time propagated is a fraction of the current integration time-step, *dt*. It is equal to $1/ipsteps$ of the integration time-step. It uses data from the **linear** function to calculate this.

### 9.5.16 *randomgen(xp)*

**Default:** *xrandomgen()*

Generates a set of initial random fields v to initialize the fields simulated. The default, **xrandomgen**, returns Gaussian real fields that are delta-correlated in space or momentum space. The default uses rfilter to modify spatial correlations in momentum space if specified.

### 9.5.17  *rfilter(w, p)*

**Default:** *xrfilter()*

Returns the momentum-space filter function for the momentum-space random terms. Each component has an array dimension the same as the input random fields in momentum space, that is, the return dimension is *[inrandoms(2), nlattice]*.

### 9.5.18  *method(a, w, p)*

**Default:** *MP* (stochastic); *RK4* (deterministic)

Specifies the stochastic integration method for the field $a$, noise $w$, parameters $p$. It returns the new field $a$. It uses the current reduced step in time p.dtr and current time p.t. This function can be set to any of the predefined stochastic integration routines provided with xSPDE, described in the Algorithms section. User-written functions can also be used. The default deterministic method, *RK4*, is a fourth-order interaction picture Runge-Kutta. The default stochastic method, *MP*, is an interaction picture midpoint integrator and is used if *ensembles* is not *[1,1,1]*.

### 9.5.19  *transfer(v, p, a0, p0)*

**Default:** *xtransfer()*

This function initializes sequential simulations, where the previous field *a0* and parameter structure *p0* are inputs to the next stage in the integration sequence. The default, *xtransfer()*, takes the output, *a0* of the previous simulation to initialize the fields $a$. Otherwise, this function is identical to *initial()*. The default set by *xtransfer* is $a = a0$.

## 9.6  Internal function reference

The following xSIM predefined functions are available to define the differential equations and averages. They all start with a capital letter. Algorithms are documented in section 7. Fields can be differentiated or integrated only in space, observables in space or time.

### 9.6.1  *Ave(o[, av ], p)*

This function takes a field or observable and returns an average over one or more dimensions. The input includes an optional averaging switch *av*. If $av(j) > 0$, an average is taken over dimension $j$. If the *av* vector is omitted, the average is taken over all space directions.

### 9.6.2  *Bin(o[, dx ], p)*

The *Bin* function takes a field o and returns probabilities on space axes that are defined by a vector dx. This allows binning of position probabilities if the observable is a mean position that is plotted on an axis. If $j$ is the first index with $dx(j) > 0$, the binning is taken over dimension $j$. The results returned are the probability of o in the bin, normalized by $1/dx(j)$. If the input array is Fourier transformed, by using the transforms attribute in the observe function, then one must set $dx(j) = p.dk(j)$ for transformed dimensions j. If the dx vector is omitted, or a scalar dx is used, the binning is over the first space direction.

### 9.6.3 D1(a[, dir,ind], p)

Takes a scalar or vector field $a$ and returns a derivative with direction *dir* using finite differences. Set $dir = 2$ for an x-derivative, $dir = 3$ for a y-derivative, and so on. The default value is $dir = 2$, which is an x-derivative. If the direction is input, an index $ind$ can be included to take a derivative of one component. If this is omitted, derivatives of all components are returned. Boundary conditions are from the *boundaries* input. The D1 input uses the entire field to identify components and boundary values. It can be made periodic (*boundaries = 0*), which is the default, or Neumann/Robin with specified derivatives using *boundaries = -1*, or Dirichlet with specified field using *boundaries = 1*.

### 9.6.4 D2(a[, dir,ind], p)

This takes a scalar or vector field $a$ and returns the second derivative in direction *dir*. Set $dir = 2$ for an x-derivative, $dir = 3$ for a y-derivative, and so on. Other properties are the same as *D1()*.

### 9.6.5 Int(o[, dx, bounds], p)

This function takes any vector or scalar field or observable and returns a space integral over selected dimensions with vector measure *dx*. If $dx(j) > 0$, dimension $j$ is integrated. Time integrals are only possible for observables. Space dimensions are labelled from $j = 2,3,...dimensions$. To integrate over the lattice, set $dx = p.dx$, otherwise set *dx(j) = p.dx(j)* for integrated dimensions and *dx(j) = 0* for non-integrated dimensions.

   If the input array is Fourier transformed by using the *p.transforms* attribute, one must set *dx(j) = p.dk(j)* for transformed dimensions *j*, to get correct results. If the *dx* vector is omitted, the integral is over all available space dimensions, assuming no Fourier transforms. The optional input *bounds* is an array of size *[p.dimensions,2]*, which specifies lower and upper integration bounds in each direction. This is only available if *dx* is input. If omitted, integration is over the whole domain.

## 9.7 Arrays and indices

Knowing the details of array indexing inside xSPDE isn't usually necessary. Yet it becomes important if you want to write your own functions to extend xSPDE, interface xSPDE with other functions, or read and write xSPDE data files with external programs. It also helps to understand how the program works.

### 9.7.1 Array tables

There are two main internal xSPDE arrays:, *fields* labelled $a$ and output *data* labelled $d$. The *fields* contain stochastic variables, the *data* contains the averaged outputs and errors estimates.

   Important array and index definitions are:

| Label | Indices | Description |
|---|---|---|
| $a$ | $[f, \mathbf{i}, e]$ | Stochastic field array |
| $v$ | $[m_1, \mathbf{i}, e]$ | Initial random variable array |
| $w$ | $[m_2, \mathbf{i}, e]$ | Noise field array |
| $r\{2\}, k\{2\}....$ | $(1, \mathbf{i}, 1)$ | Numbered space/momentum coordinates |
| $x,y,z,kx,ky,kz$ | $(1, \mathbf{i}, 1)$ | Labelled space/momentum coordinates |
| $o$ | $\{n\}(\ell, \mathbf{j})$ | Cell array of all observed averages |
| $d$ | $\{s\}\{n\}(\ell, \mathbf{j}, c)$ | Cell array of data with errors |
| $rawdata$ | $\{s, c, h\}(f, \mathbf{j}, e)$ | Raw trajectories |
| $points$ | $[pt_1, pt_2 \ldots pt_d]$ | Vector of lattice sizes: |
| $ensembles$ | $[e, h_1, h_2]$ | Vector of ensemble sizes: |

Here:

- $f$ is the field index, combining fields and auxiliary fields

- $\mathbf{i}$ is the space index

- $e$ is the vector ensemble index

- $\mathbf{m}$ is the random or noise index

- $\mathbf{j} = [j_1, \mathbf{i}]$ is the space-time index

- $\ell$ is the line index

- $n$ is the observe and/or function index

- $h$ is the high-level ensemble index (combines $h_1, h_2$)

- $s$ is the sequence index

- $c$ is the check index

When fields are passed to *observe*, or to *rawdata* outputs, the defined or auxiliary fields are included as well. Apart from the first dimension, the common dimensionality for internal arrays used in computations is $d.lattice = [d.space, ensembles(1)]$.

### 9.7.2  Simulation data in xSIM

In xSIM, the space-time dimension $d$ is unlimited, although xGRAPH can only plot up to three chosen axes at a time. All fields are stored in real or complex numerical arrays of rank $1 + d$. Average results are stored are stored in real or complex numerical arrays of rank $2 + d$.

The array index ordering in xSPDE integrated fields is $(f_i, \mathbf{i}, e)$, where:

- The first index is an integrated field index $f_i$, not including auxiliary fields

- The next $d - 1$ indices are $\mathbf{i}$, which is a space index with no time index.

- The last is an ensemble index $e$, to store low-level parallel trajectories.

The array index ordering in graphical averaged data is $(\ell, \mathbf{j}, c)$where:

- The first index is a line index $\ell$.

- The next $d$ indices are $\mathbf{j} = [j_1, \ldots j_d] = [j, \mathbf{i}]$, for time *and* space.

- The last is a check index $c$, for comparisons and errors.

Stored data uses heterogenous cell arrays to package numerical arrays with additional high level indices. The first cell index is the sequence index, $s$. Inside each sequence, data cell arrays all have a graph index $n$. This distinguishes the different averages generated for output graphs and data. Raw data has cell indices for the sequence, time-step and high level ensembles.

In summary, the xSPDE internal arrays are as follows:

- **Field** arrays $a(f_i, \mathbf{i}, e)$ - these have a field index, a space index and low-level ensemble index $e$.

- **Auxiliary** arrays $ax(f_a, \mathbf{i}, e)$ - these are appended to the field arrays for raw data and observables.

- **Random** and **noise** arrays $w(r, \mathbf{i}, e)$ - these are initial random fields or noise fields. The first index may have a different range to the field index.

- **Coordinate** arrays $x(1, \mathbf{i})$ - these contain the coordinates at grid-points, with labels $x, y, z$, and $j_1 = 1$. Numeric labels $x\{l\}$ are used for $d > 4$, where $l = 2, \ldots d$. The same sizes are used for:

  - momentum coordinates $kx, ky, kz$ (alternatively $k\{2\}, k\{3\}, \ldots$)
  - spectral derivative arrays $Dx, Dy, Dz$ (alternatively $D\{2\}, D\{3\}, \ldots$) .

- **Raw data** arrays $r\{s, c, h\}(f, \mathbf{j}, e)$ - these are cell arrays of generated trajectories, including integrated and defined field values. They are optional, as they use large amounts of memory. These are saved in cell arrays with indices $s$ for the sequence, $c$ for the time-step error-check and $h$ for high level ensemble index. The cell indices are:

  - $s = 1, \ldots S$ for the sequence number,
  - $c = 1, 2$ for the error-checking time-step used, first coarse then fine,
  - $h = 1, \ldots ensemble(2) * ensemble(3)$ for a high level parallel and serial ensemble index.

- **Observe** data arrays $o\{n\}(\ell, \mathbf{j})$ - these are generated in xSIM by the observe functions, then used to store generated average data at all time points. The cell index $n$ is the *observe* index, which indexes overs the observe functions. The internal index $\ell$ is a *line* index generated by an observe function.

- **Graphics** data arrays $g\{s\}\{n\}(\ell, \mathbf{j}, c))$ - these store the final results. Check indices are added to store error estimates and comparisons, where $c = 1$ for the average, $c = 2$ for the time-step error, and $c = 3$ for the sampling error. If there is comparison data, it uses $c = 4$ up to $c = 6$, to allow for error bars. Graphics cell data uses cell indices $\{s\}$ for the *sequence* index, and $\{n\}$ for the *graph* index. This has a default of the index of the *observe* function. Otherwise, if this is overwritten by an xSIM output *function*, the graph index equals the function index.

## 9.8   Internal parameter table

The internal parameter structures in xSPDE are available to the user if required. Internally, all xSPDE parameters are stored in the parameter structures passed to functions. This includes the data given above from the input structures. In addition, it includes the computed parameters given below, which includes internal array dimensions.

Data in $k-$space is stored in two alternative lattices, each having their own axis vectors. The propagation grid is used while propagating, and is compatible with numerical FFT conventions where the first index value is $k = 0$. The graphics grid is centered around $k = 0$, and is used for graphics and data storage, following scientific conventions.

For more than four total dimensions, the spatial grid, momentum grid and derivative grid notation of $t, x, y, z,\ \omega, kx, ky, kz$ and $Dx, Dy, Dz$ is changed to use numerical labels that correspond to the dimension numbers, i.e., $D\{2\}, \ldots D\{d\}$, $r\{1\}, \ldots r\{d\}$, $k\{1\}, \ldots k\{d\}$.

Numeric dimension labeling can also be used even for lower dimensionality if preferred.

| Label | Type | Typical value | Description |
|---|---|---|---|
| $t, x, y, z$ | array | - | Space-time grid of $t, x, y, z$ |
| $\omega, kx, ky, kz$ | array | - | Frequency-momentum grid of $k_x, k_y, k_z$ |
| $Dx, Dy, Dz$ | array | - | Derivative grid of $D_x, D_y, D_z$ |
| $r\{1\}, \dots r\{d\}$ | array | - | Space-time grid of $r_1, \dots r_d$ |
| $k\{1\}, \dots k\{d\}$ | array | - | Graphics momentum grid of $k_1, \dots k_d$ |
| $D\{2\}, \dots D\{d\}$ | array | - | Derivative grid of $D_2, \dots D_d$ |
| $dx$ | vector | *[0.2,..]* | Steps in $[t, x, y, z]$ |
| $dk$ | vector | *[0.61,....]* | Steps in $[\omega, k_x, k_y, k_z]$ |
| $dt$ | double | *0.2000* | Output time-step |
| $dtr$ | double | *0.1000* | Computational time-step |
| $v$ | real | *1* | Spatial lattice volume |
| $kv$ | real | *1* | Momentum lattice volume |
| $dv$ | real | *1* | Spatial cell volume |
| $dkv$ | real | *1* | Momentum cell volume |
| $xc\{d\}$ | cells of vectors | *[-5,... 5]* | Coordinate axes in $t, x, y, z$ |
| $kc\{d\}$ | cells of vectors | *[-5,..5]* | Graphics axes in $[\omega, k_x, k_y, k_z]$ |
| $kcp\{d\}$ | cells of vectors | *[0,...]* | Propagation axes in $[\omega, k_x, k_y, k_z]$ |
| $s.dx$ | double | 1 | Initial stochastic normalization |
| $s.dxt$ | double | *3.1623* | Propagating stochastic normalization |
| $s.dk$ | double | 1 | Initial k stochastic normalization |
| $s.dkt$ | double | *3.1623* | Propagating k stochastic normalization |
| $nspace$ | integer | 35 | Number of spatial lattice points |
| $nlattice$ | integer | 3500 | Total lattice: *ensembles(1) x n.space* |
| $ncopies$ | integer | 20 | *ensembles(2) x ensembles(3)* |
| $inrandoms$ | vector | [2,0] | Number of initial random fields |
| $noises$ | vector | [2,0] | Number of noise fields |
| $d.space$ | vector | *[35, 35]* | Space dimensions: [points(2), points(3), ...] |
| $d.lattice$ | vector | *[1, 1]* | Lattice dimensions: [d.space, ensembles(1)] |
| $d.a$ | vector | *[1, 1]* | Dimensions for $a$ field |
| $d.r$ | vector | *[1, 1]* | Dimensions for coordinates |
| $d.fields$ | vector | *[1, 1]* | Dimensions for $a$ field (including time) |
| $d.aplus$ | vector | *[1, 1, 1]* | Dimensions for integrated plus defined fields |
| $d.k$ | vector | *[0, 1, 1]* | Dimensions for noise transforms |
| $d.obs$ | vector | *[1, 35]* | Dimensions for observations |
| $d.data$ | vector | *[1, 35, 3]* | Dimensions for average data |
| $d.raw$ | vector | *[1, 51, 35, 100]* | Dimensions for raw data |

## 9.9   xSIM overall structure

### 9.9.1   xSPDE

The control program, $xspde$, calls the *xsim* integration and *xgraph* graphics functions successively

$$\textbf{xspde} \rightarrow \begin{cases} \textbf{xsim}\ (simulations) \\ \textbf{xgraph}\ (graphics) \end{cases}$$

### 9.9.2   xSIM

The integration function, *xsim*, generates all data. It first carries out elementary checks in *xpreferences* and constructs the grid of lattice points in *xlattice*. Then it generates the nested ensembles in *xensemble*, and integrates each subensemble using *xpath*. The output data is written to files, if required, in *xwrite*.

$$\begin{aligned} \textbf{xsim} &\rightarrow \textbf{xpreferences} \rightarrow \textbf{xlattice}\ (checks\ inputs) \\ &\rightarrow \textbf{xensemble} \leftrightarrow \textbf{xpath} \leftrightarrow \textbf{xdata}\ (simulates) \\ &\rightarrow \textbf{xwrite}\ \ (stores\ data) \end{aligned}$$

# 10   xGRAPH reference

*This section gives a reference explanation of the xGRAPH parameters and functions.*

## 10.1   Overview

The graphics function provided is a general purpose multidimensional batch graphics code, xGRAPH, which is automatically called by xSPDE when xSIM is finished. The results are graphed and output if required. Alternatively, xGRAPH can be replaced by another graphics code, or it can be used to process the data generated by the xSIM function at a later time.

The *xgraph* function call syntax is:

- *xgraph (data [,input])*

This takes simulation *data* and *input* cell arrays, then plots graphs. The *data* should have as many cells as there are *input* cells, for sequences.

If *data = 'filename.h5'* or *'filename.mat'*, the specified file is read both for *input* and *data*. Here *.h5* indicates an HDF5 file, and *.mat* indicates a Matlab file.

When the *data* input is a filename, parameters in the file can be replaced by new *input* parameters that are specified. Any stored *input* in the file is then overwritten when graphs are generated. This allows graphs of data to be modified retrospectively, if the simulation takes too long to be run again in a reasonable timeframe.

### 10.1.1   Parameter and data structures

This is a batch graphics function, intended to process quantities of graphics data, input as a cell array of multi-dimensional data. Theoretical and/or experimental data is passed to the graphics program, including the complete *data* cell array and a cell array of graphics parameters for plotting each graph.

To explain xGRAPH in full detail,

- Data to be graphed are recorded sequentially in a cell array, with *data={d1,d2,...}*.

- Graphics parameters including defaults are given in the *input* cell array.

- This describes a sequence of graph parameters, so that *input={p1,p2,...}*.

- For a one member sequence, a dataset and parameter structure can be used on its own.

- Each dataset and parameter structure describes a set of graphs.

The data input to *xGRAPH* can either come from a file, or from data generated directly with *xSIM*. The main graphics data is a nested cell array. It contains several numerical graphics arrays. Each defines one independent set of averaged data, the observed data averages, stored in a cell array indexed as $data\{s\}\{n\}(\ell, \mathbf{j}, c)$. To graph these also requires a corresponding cell array of structures of graphics parameters.

The output is unlimited, apart from memory limits. The program also generates error comparisons and chi-squared values if required. The data structure for input is as follows:

1. The input *data* is a cell array of *datasets,* which can be collapsed to a single dataset

2. The *parameters* are also a cell array of parameter structures, which can be collapsed to one structure

3. The *dataset* is a cell array of multidimensional *graphs*, each with arbitrary dimensionality.

4. Each *graph* is an array which can generate multiple plots, as defined by the parameters.

5. The first index of each graph allows multiple lines, with different line-styles

6. The last index of each graph array is optionally used for error and comparison fields.

### 10.1.2 Comparisons

For every type of observation in xSIM, the observe function can be accompanied by a comparison function, *compare(p)*. This generates a vector of analytic solutions or experimental data which is compared to the stochastic results. Results are plotted as additional lines on the two-dimensional graphical outputs, and comparison differences can be graphed in any dimension.

Comparisons are possible for either moments or probabilities, and can be input in any number of dimensions. When there are error estimates, a chi-squared test is carried out to determine if the difference is within the expected step-size and sampling error bars. If the comparison has errors, for example from experimental data, the chi-squared test will include the experimental errors.

Comparison data can be added to the graphics files from any source. It must match the corresponding space-time lattice or probability bins that are in the graphed data. Note that the *compare* functions are specified during the simulation. The graphics code does not generate comparison data, as it is dedicated to graphics, not to generating data.

### 10.2 Parameter table

The complete cell array of the simulation data is passed to the *xGRAPH* program, along with graphics parameters for each observable, to create an extended graphics data structure. Graphics parameters have default values which are user-modifiable by editing the *xgpreferences* function.

Some input parameters are global parameters for all graphs. However, most *xGRAPH* parameters are cell arrays indexed by graph index. These graphics parameters are individually set for each output that is plotted, using the cell index $\{n\}$ in a curly bracket. If present they replace the global parameters like labels.

If a graph index is omitted, and the parameter is not a nested array, the program will use the same value for all graphs. The *axes, glabels, legends, lines, logs,* and *xfunctions* of each graph are nested cell arrays, as there can be any number of lines and axis dimensions. In the case of the *logs* switch, the observable axis is treated as an extra dimension.

The plotted result can be an arbitrary function of the generated average data, by using the optional input *gfunction.* If this is omitted, the generated average data that is input is plotted.

Comparisons are plotted if present in the input data indexed by the last error index with $e > errors$, where $errors = 3$ in all data generated by xSIM.

A table of the graphics parameters is given below.

 

| Label | Default value | Description |
|---|---|---|
| $axes\{n\}$ | {0,..} | Points plotted for each axis |
| $chisqplot\{n\}$ | 0 | Chi-square plot options |
| $cutoff$ | 1.e-12 | Global lower cutoff for chi-squares |
| $cutoffs\{n\}$ | $cutoff$ | Probability cutoff for n-th graph |
| $diffplot\{n\}$ | 0 | Comparison difference plot options |
| $errors$ | 0 | Index of last error field in $data$ |
| $esample\{n\}$ | 1 | Size and type of sampling error-bar |
| $font\{n\}$ | 18 | Font size for graph labels |
| $gfunction\{n\}$ | @(d,~) d{n} | Functions of graphics data |
| $glabels\{n\}$ | {'t','x','y','z'} | Graph-specific axis labels |
| $graphs$ | $[1:max]$ | Vector of all the required graphs |
| $gsqplot\{n\}$ | 0 | G-square (likelihood) plot options |
| $headers\{n\}$ | '' | Graph headers |
| $images\{n\}$ | 0 | Number of movie images |
| $imagetype\{n\}$ | 0 | Type of 3D image |
| $klabels$ | {'\omega','k_x','k_y','k_z'...} | Global transformed axis labels |
| $legends\{n\}$ | {'label1',..} | Legends for multi-line graphs |
| $limits\{n\}$ | {[lc1,uc1],[lc2,uc2]} | Axis limits, first lower then upper |
| $linestyle\{n\}$ | {'-',..} | Line styles for multiline 2D graphs |
| $linewidth\{n\}$ | 0.5 | Line width for 2D graphs (in points) |
| $logs\{n\}$ | {0,..} | Axis logarithmic switch: 0 linear, 1 log |
| $minbar\{n\}$ | 0.01 | Minimum relative error-bar |
| $mincount$ | 10 | Global counts for chi-square cutoffs |
| $name$ | '' | Global graph header |
| $olabels\{n\}$ | 'a_1' | Observable labels |
| $pdimension\{n\}$ | 3 | Maximum plot dimensions |
| $saveeps$ | 0 | Switch, set to 1 to save eps files |
| $savefig$ | 0 | Switch, set to 1 to save figure files |
| $scale\{n\}$ | 1 | Scaling: Counts/ probability density |
| $transverse\{n\}$ | 0 | Number of transverse plots |
| $xfunctions\{n\}$ | {@(t,~) t,@(x,~) x,..} | Axis transformations |
| $verbose$ | 0 | 0 for brief, 1 for informative, 2 for full output |
| $xlabels$ | {'t','x','y','z'...} | Global axis labels |
| $octave$ | 0 | 0 for Matlab, 1 for octave environment |

- Up to 6 types of input data can occur, including errors and comparisons, indexed by the last index. The original mean data always has $c = 1$. If there are no errors or comparisons, one graph is plotted for each dimensional reduction.

- The input data has up to two error bars (I and II), and optional comparisons also with up to two error bars.

- Type (I) errors labeled $c = 2$ have standard vertical error bars. Type II error bars labeled $c = 3$, usually standard deviation errors from sampling, have two solid lines.

- If $esample = -1$, both error bars are combined and the RMS errors are plotted as a single

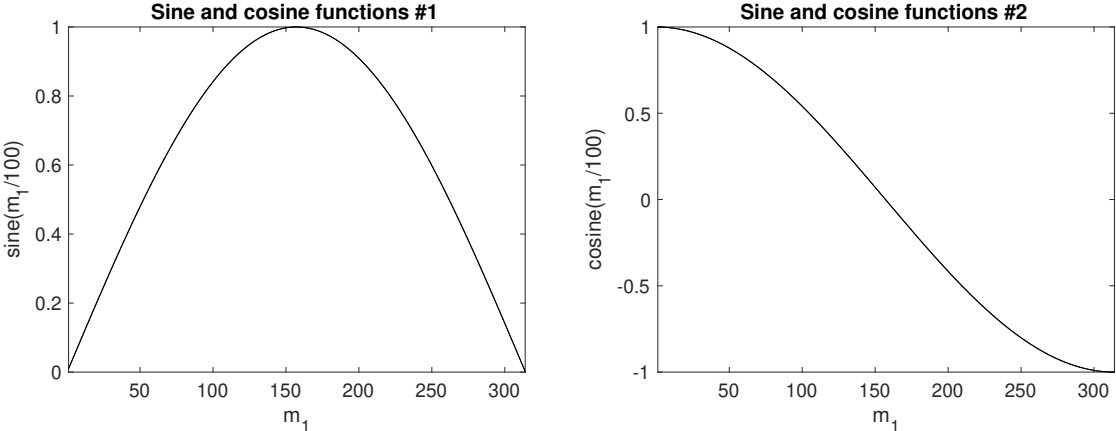

Figure 11: Example: xgraph output of two plots

error bar.

- If $diffplot > 0$, differences are plotted as unnormalized ($diffplot = 1$), or normalized ($diffplot = 2$) by the total RMS errors. If $diffplot = 3$, raw comparison data is plotted.

- When differences are plotted, the total comparison errors are treated as type (I) error bars, while total simulation errors are treated as type (II) errors with parallel lines in the graphs, to distinguish them.

A detailed description of each parameter is listed in Sec (10.7).

## 10.3   Example

A simple example of data and input parameters, but without errors or comparisons is as follows

```
p.name = 'Sine and cosine functions';
p.olabels = {'sine(m_1/100)','cosine(m_1/100)'};
data = {sin([1:100*pi]/100),cos([1:100*pi]/100)};
xgraph(data,p);
```

## 10.4   xGRAPH data arrays

The data input to *xGRAPH* can come from a file, or from data generated directly from any compatible program.

The data is stored in a cell array $data$ with structure:

$$data\{s\}\{n\}(\ell, \mathbf{j}, c)$$

Each member of the outer cell array *data{s}* defines a number of related sets of graphical data, all described by common parameters *input{s}*. Comparisons and errors are plotted if there are errors and comparison data in the input, indexed by $c$. This generates comparison plots, as well as error totals and $\chi$- squared error estimate when there are statistical variances available.

An individual member of *data{s}{n}* is a multidimensional array, called a *graph* in the xSPDE User's guide. For each *graph*, multiple different plots with different dimensionality can be obtained from the dataset *data{s}{n}*, either through projections and slices or by generating additional data defined with graphics functions. Either or both alternatives are available.

Note that:

- If a sequence has one member, the outer cell array can be omitted.

- In this simplified case, if there is only one *graph* array, the inner cell array can be omitted.

The graphics data for a single dataset is held in a multidimensional real array, where:

- $\ell$ is the index for lines in the graph. Even for one line, the first dimension is retained.

- $\mathbf{j} = j_1, \dots j_d$ is the array index in each dimension, where $d \geq 1$.

- Averages in momentum space have the momentum origin as the central index.

- If integrals or spatial averages are used, the corresponding dimension has one index $j_d = 1$.

- With probabilities, extra dimensions are added to $\mathbf{j}$ to store the bin indices.

- $c$ indexes error-checks and comparisons in the data. If not present, the last dimension is omitted.

- If $c > p.errors$, the extra fields are comparison inputs, where $p.errors = 0$ is the default.

When the optional comparison fields are used, an input parameter *errors* is required to indicate the maximum error index, to distinguish data from comparisons. Parameter structures from xSIM have *errors* = 3 set to allow for both sampling errors and discretization errors. If this is omitted, the default is *errors* = 0, which implies that there is no error or comparison data

If *errors* > 0, the last index can have larger values with $c > errors$, for comparisons. The special case of *errors* = 1 is used if the data has no error bars, but there are comparisons in the data. Larger indices are used to index the comparison data, which can also have two types of errors. The largest usable last index is *errors* + 3.

It is possible to directly plot the *raw* data using xGRAPH. One can even combine the raw data with a graphics parameter input. But since the raw data has no error estimates - it is raw data - one must set $p.errors = 0$, since the xsim output parameters have a default of $p.errors = 3$. This will give a single trajectory.

However, the raw data from a simulation typically includes many trajectories if $ensembles(1) > 0$, and then one must select particular trajectory datasets from the raw cell array, to plot just one.

### 10.4.1 Input parameters and defaults

A sequence of graph parameters is obtained from inputs in a cell array, as *input = {in1, in2, ...}*. The input parameters of each simulation in the sequence are specified in a Matlab structure. The inputs are numbers, vectors, strings, functions and cell arrays. All metadata has preferred values, so only changes from the preferences need to be input. The resulting data is stored internally as a sequence of structures in a cell array, to describe the simulation sequence.

The graphics parameters are also stored in the cell array *input* as a sequence of structures *p*. This only need to be input when the graphs are generated and can be changed at a later time to alter the graphics output. A sequence of simulations is graphed from *input* specifications.

If there is one simulation, just one structure can be input, without the sequence braces. The standard way to input each parameter value is:

$$p.label = parameter$$

The standard way to input a function handle is:

$$p.label = @function$$

The inputs are scalar or vector parameters or function handles. Quantities relating to graphed averages are cell arrays, indexed by the graph number. The available inputs, with their default values in brackets, are given below.

Simulation metadata, including default values that were used in a particular simulation, can be included in the input data files. This is done in both the *.mat* and the *.h5* output files generated by xSIM, so the entire graphics input can be reconstructed or changed.

Parameters can be numbers, vectors, strings or cell arrays. Conventions that are used are that:

- All input parameters have default values

- Vector inputs of numbers are enclosed in square brackets, *[...]*.

- Cell arrays of strings, functions or vectors are enclosed in curly brackets.

- Vector or cell array inputs with only one member don't require brackets.

- Incomplete parameter inputs are completed with the last used default value.

- Function definitions can be handles pointing elsewhere, or defined inline.

If any inputs are omitted, there are default values which are set by the internal function *xgpreferences*. The defaults can be changed by editing *xgpreferences*.

In the following descriptions, *graphs* is the total number of graphed variables of all types. The space coordinate, image, image-type and transverse data can be omitted if there is no spatial lattice, that is, if the dimension variable is set to one.

For uniformity, the graphics parameters that reference an individual data object are cell arrays. These are indexed over the graph number using braces *{}*. If a different type of input is used, like a scalar or matrix, xSPDE will attempt to convert the type to a cell array.

Axis labels are cell arrays, indexed over dimension. The graph number used to index these cell arrays refers to the data object. In each case there can be multiple generated plots, depending on the graphics input.

## 10.5   Cascaded plots

The xGRAPH function generates a default range of graphs, but this can be modified to suit the user. In the simplest case of one dimension, one graph dataset will generate a single plot. For higher dimensions, a cascade of plots is generated to allow visualization, starting from 3D movies, then 3D static plots and finally 2D slices. These can also be user modified.

Note that for all probabilities, the plot dimension is increased by the bin range dimensionality.

### 10.5.1   Plot dimensions

The *pdimension* input sets the maximum plotted dimensions. For example, $pdimension\{1\} = 1$ means that only plots vs $r_1$ are output for the first function plotted. Default values are used for the non-plotted dimensions, unless there are axes specified, as indicated below.

The graphs cascade down from higher to lower dimensions, generating different types of graphs. Each type of graph is generated once for each function index.

### 10.5.2   Plot axes

The graphics axes that are used for plotting and the points plotted are defined using the optional *axes* input parameters, where $axes\{n\}$ indicates the $n$-th specified graph or set of generated graph data.

If there are no *axes* inputs, or the *axes* inputs are zero - for example, $axes\{1\} = \{0,0,0\}$ - only the lowest dimensions are plotted, up to *3*. If either the data or *axes* inputs project one point in a given dimension, - for example, $axes\{1\} = \{0,31,-1,0\}$, this dimension is suppressed in the plots, which reduces the effective dimension of the data - in this case to two dimensions.

Examples:

- $axes\{1\} = \{0\}$ - For function 1, plot all the first dimensional points; higher dimensions get defaults.

- $axes\{2\} = \{-2,0\}$ - For function 2, plot the maximum value of $r_1$ (the default) and all higher-dimensional x-points.

- $axes\{3\} = \{1 : 4 : 51, 32, 64\}$ - For function 3, plot every 4-th $x_1$ point at $x_2$ point 32, $x_3$ point 64

- $axes\{4\} = \{0, 2 : 4 : 48, 0\}$ - For function 4, plot every $x_1$ point , every 4-th $x_2$ point, and all $x_3$-points.

Points labelled $-1$ indicates a default 'typical' point, which is the midpoint. If one uses $-2$, this is the last point.

Lower dimensions are replaced by corresponding higher dimensions if there are *dimensions* or *axes* that are suppressed. Slices can be taken at any desired point, not just the midpoint. The notation of $axes\{1\} = \{6 : 3 : 81\}$, is used to modify the starting, interval, and finishing points for complete control on the plot points.

The graphics results depend on the resulting **effective** dimension, which is equal to the actual input data dimension unless there is an *axes* suppression, described above. Since the plot has to include a data axis, the plot itself will usually have an extra data axis.

One can plot only three axes directly using standard graphics tools. The strategy to deal with the higher effective dimensionality is as follows. For simplicity, "time" is used to label the first effective dimension, although in fact any first dimension is possible:

**dimensions = 1** For one lattice dimension, a 2D plot of observable *vs t* is plotted, with data at each lattice point in time. Exact results, error bars and sampling error bounds are included if available.

**dimensions = 2** For two lattice dimensions, a 3D image of observable *vs x,t* is plotted. A movie of distinct 2D graphic plots is also possible. Otherwise, a slice through $x = 0$ is used tp reduce the lattice dimension to 1.

***dimensions = 3*** For three lattice dimensions, if $images > 1$, a movie of distinct 3D graphic images of observables are plotted as $images$ slices versus the first plot dimension. Otherwise, a slice through the chosen point, is used at the highest dimension to reduce the lattice dimension to 2.

***dimensions = 4,5..*** For higher lattice dimensions, a slice through a chosen point, or the default midpoint is used to reduce the lattice dimension to 3.

As explained above, in addition to graphs versus $x_1$ the **xGRAPH** function can generate *images* (3D) and *transverse* (2D) plots at specified points, up to a maximum given by the number of points specified. The number of these can be individually specified for each graph number. The images available are specified as *imagetype*$= 1, \ldots 4$, giving:

1. 3D perspective plots (Matlab *surf* - the default)

2. 2D filled color plots (Matlab *contourf* )

3. contour plots (Matlab *contour* )

4. pseudo-color plots (Matlab *pcolor* )

Error bars, sampling errors and multiple lines for comparisons are only graphed for 2D plots. Error-bars are not plotted when they are below a user-specified size, with a default of 1% of the maximum range, to improve graphics quality. Higher dimensional graphs do not output error-bar data, but they are still recorded in the data files.

## 10.6   Probabilities and parametric plots

Probability data can be input and plotted like any other data. It is typically generated from simulation programs using the *binranges* data for binning. It is plotted like any other graph, with any dimension, except that the total dimension is extended by the number of variables or lines in the *observe* function.

### 10.6.1   Chi-squared plots

In addition the program can make a $\chi^2$ plot, which is a plot of the $\chi^2$ comparison with a comparison probability density against space and/or time. This allows a test of the simulated data against a known target probability distribution, provided that the following input data conditions are satisfied:

- The input data dimension exceeds the p.*dimensions* parameter,

- The switch p.*chisqplot* is set to 1or 2, and

- The input data includes comparison function data.

The $\chi^2$ plots, depending on $p.chisqplot$ are:

1. a plot of $\chi^2$ and $k$, where $k$ is the number of valid data points,

2. a plot of $\sqrt{2\chi^2}$ and $\sqrt{2k-1}$, which should have a unit variance.

Here, for one point in space and time, with $m$ bins, $N_j$ counts per bin and $E_j$ expected counts:

$$\chi^2 = \sum_{j=1}^{m} \frac{(N_j - E_j)^2}{E_j}.$$ (214)

The number $k$ is the number of valid counts, with $N_j, E_j > mincount$. This is partly determined from the requirement that the probability count data per bin is greater than the $p.mincount$ parameter. The default is set to give a number of samples $> 10$. The program prints a summary that sums over of all the $\chi^2$ data.

The $p.scale\{n\}$ parameter gives the number of counts per bin at unit probability density. This is needed to set the scale of the $\chi^2$ results, ie, $N_j = scale\{n\} \times p_j$, where $p_j$ is the probability density that is compared and plotted in the simulation data. Note that a uniform bin size is assumed here, to give a uniform scaling.

### 10.6.2 Comparisons with variances

It can be useful to compare two probability distributions with different variances. For one point in space and time, with $m$ bins, $p_j$ probability density and $e_j$ expected probability density,

$$\chi^2 = \sum_{j=1}^{m} \frac{(p_j - e_j)^2}{\sigma_j^2 + \sigma_{e,j}^2}.$$ (215)

In this case, $\sigma_j^2$ and $\sigma_{e,j}^2$ are the sampling errors in the simulation data and comparison data, so that built-in error fields in the data are used to work out the $\chi^2$ results. This option is chosen if $p.scale\{n\} = 0$, and the cutoff for the data is then specified so that $p_j, e_j > p.cutoffs\{n\}$. This only has a $\chi^2$ distribution if points are independent.

### 10.6.3 Maximum likelihood

It is also possible to plot the $G^2$ or maximum likelihood plot of the data, which is an alternative means to compare distributions, where

$$G^2 = 2 \sum_{j=1}^{m} N_j \ln(N_j/E_j).$$ (216)

The expected values $E_j$ are automatically scaled so that $\sum N_j = \sum E_j$, with the same minimum count cutoff that is used for the $\chi^2$ data. The result is similar to the $\chi^2$ results. It is obtained if p.*gsqplot* is set to 1 or 2 and requires for the input that $p.scale\{n\} > 0$. It is sometimes regarded as a preferred method for comparisons.

### 10.6.4 Parametric plots

Any input dataset can be converted to a parametric plot, where a second data input is plotted along the horizontal axis instead of the time coordinate. It is also possible to substitute a second data input for the x-axis data if a parametric plot in space is required instead. This allows visualization of how one type of data changes as a function of a second type of data input.

The two datasets that are plotted must have the same number of lines, that is, the first index range should be the same, in order that multiple lines can be compared. This is achieved where

required using the *p.scatters* input in the simulation code. The details of the parametric plot are specified using the input:

$$p.parametric\{n\} = [n1, p2] \tag{217}$$

Here $n$ is the graph number which is plotted, and must correspond to an input dataset. The number $n1$ is the graph number of the observable that is plotted on the horizontal axis, ignoring functional transformations. The second number is the axis number where the parametric value is substituted, which can be the time (axis 1) or the x-coordinate (axis 2), if present.

In all cases the vertical axis is used to plot the original data. The specified horizontal axis is used for the parametric variable. Only vertical error-bars are available. An example is given in xAMPLES/SDE_1/SHO, which is a noise-driven harmonic oscillator, with several lines plotted of *x* vs *y*.

## 10.7 Parameter reference

### 10.7.1 *axes{n}*

**Default:** *{0,0,0,..}*

Gives the axis points plotted for the $n$-th plotted function, in each dimension. Each entry value is a vector range for a particular plot and dimension. Thus, $p = 5$ gives the fifth point only, and a vector input $p = 1:4:41$ plots every fourth point. Single points generate graphics projections, allowing the other dimensions to be plotted. Zero or negative values are shorthand. For example, $p = -1$ generates a default point at the midpoint, $p = -2$ the endpoint, and $p = 0$ is the default value that gives the vector for the every axis point. For each graph type, i.e. *n=1,..graphs* the axes can be individually specified in each dimension, *d=1,..dimensions*. If more than three axes are specified to be vectors, only the first three are used, and others are set to default values in the plots.

**Example:** *p.axes{4} = {1:2:10,0,0,-1}*

### 10.7.2 *diffplot{n}*

**Default:** *0*

Differences are plotted as a comparison dashed line on 2*D* plots as a default. Otherwise, a separate difference plot is obtained which is unnormalized (*diffplot = 1*), or normalized (*diffplot = 2*) by the total RMS errors. If *diffplot = 3*, the comparison data is plotted directly as an additional graph.

**Example:** *p.diffplot{3} = 2*

### 10.7.3 *errors*

**Default:** *0*

Indicates if the last index in the graphics input data arrays is used for error-bars and/or comparisons. Should be set to zero if there is no error or comparison data. If non-zero, this will give the highest last index used for errors. The standard *xsim* output sets $p.errors = 3$ automatically. As a special case, $p.errors = 1$ is used to indicate that there is comparison data but no error data.

If $p.errors > 0$, the data indexed up to $p.errors$ gives the data, then a maximum of two types of error bars. Up to three further index values, up to $p.errors + 3$, are available to index all comparison data and its error fields. The maximum last index value used is 6.

**Example:** $p.errors = 2$

### 10.7.4  *esample{n}*

**Default:** *1*

This sets the type and size of sampling errors that are plotted. If $esample = 0$, no sampling error lines are plotted, just the mean. If $esample = -n$, $\pm n\sigma$ sampling errors are included in the error-bars. If $esample = n$, separate upper and lower $\pm n\sigma$ sampling error lines are plotted. In both cases, the magnitude of esample sets the number of standard deviations used.

**Example:** *p.esample{3} = -1*

### 10.7.5  *font{n}*

**Default:** *18*

This sets the default font sizes for the graph labels, indexed by graph. This can be changed per graph.

**Example:** *p.font{4}=18*

### 10.7.6  *functions*

**Default:** number of functional transformations

This gives the maximum number of output graph functions and is available to restrict graphical output. The default is the length of the cell array of input data. Normally, the default will be used.

**Example:** *p.functions = 10*

### 10.7.7  *glabels{n}*

**Default:** *xlabels* or *klabels*

Graph-dependent labels for the independent variable labels. This is a nested cell array with first dimension of *graphs* and second dimension of *dimensions*. This is used to replace the global values of *xlabels* or *klabels* if the axis labels change from graph to graph, for example, if the coordinates have a functional transform. These can be set for an individual coordinate on one graph if needed.

**Example:** $p.glabels\{4\}\{2\} = 'x\char`\^2'$

### 10.7.8  *graphs*

**Default:** observables to plot

This gives the observables to plot. The default is a vector of indices from one to the length of the cell array of observe functions. Normally not initialized, as the default is used. Mostly used to reduce graphical output on a long file.

**Example:** *p.graphs = 10*

### 10.7.9 *gtransforms{n}*

**Default:** [0,0,...]

This switch specifies the Fourier transformed graphs and axes for graphics labeling. Automatically equal to *ftransforms* if from an earlier xSIM input, but can be changed. If altered for a given graph, all the axis Fourier switches should be reset. This is ignored if there is no *dimensions* setting to indicate space dimensions.

**Example:** *p.gtransforms{1} = [0,0,1]*

### 10.7.10 *headers{n}*

**Default:** "

This is a string variable giving the graph headers for each type of function plotted. The default value is an empty string. Otherwise, the header string that is input is used. Either is combined with the simulation name and a graph number to identify the graph. This is used to include simulation headers to identify graphs in simulation outputs. Graph headers may not be needed in a final published result. For this, either edit the graph, or use a space to make plot headers blank: *p.headers{n} = ' '*, or *p.name = ' '* .

**Example:** *p.headers{n} = 'my_graph_header'*

### 10.7.11 *images{n}*

**Default:** *0*

This is the number of 3D, transverse o-x-y images plotted as discrete time slices. Only valid if the input data dimension is greater than 2. If present, the coordinates not plotted are set to their central value when plotting the transverse images. This input should have a value from zero up to a maximum value of the number of plotted points. It has a vector length equal to *graphs*.

**Example:** *p.images{4} = 5*

### 10.7.12 *imagetype{n}*

**Default:** *1*

This is the type of transverse o-x-y movie images plotted. It has a vector length equal to *graphs*.

- *imagetype = 1* gives a perspective surface plot

- *imagetype = 2*, gives a 2D plot with colors

- *imagetype = 3* gives a contour plot with 10 equally spaced contours

- *imagetype = 4* gives a pseudo-color map

**Example:** *p.imagetype{n} = 1, 2, 3, 4*

### 10.7.13  *klabels*

**Default:**  *{'\omega', 'k_x', 'k_y', 'k_z'}" or "{'k_1', 'k_2', 'k_3', 'k_4',...}*

Labels for the graph axis Fourier transform labels, vector length of *dimension*s. The numerical labeling default is used when the "*p.numberaxis*" option is set. Note, these are typeset in Latex mathematics mode! When changing from the default values, all the required new labels must be set.

**Example:**  *p.klabels= {'\Omega', 'K_x', 'K_y',}*

### 10.7.14  *legends{n}*

**Default:**  *{",")*

Graph-dependent legends, specified as a nested cell array of strings for each line.

**Example:**  *p.legends{n} = {labels(1), ..., labels(lines)}*

### 10.7.15  *limits{n}*

**Default:**  *{0,0,0,0; ...}*

Graph-dependent limits specified as a cell array with dimension *graphs*. Each entry is a cell array of graph limits indexed by the dimension, starting from $d = 1$ for the time dimension. The limits are vectors, indexed as 1,2 for the lower and upper plot limits. This is useful if the limits required change from graph to graph. If an automatic limit is required for either the upper or lower limit, it is set to *inf*.

    An invalid, scalar or empty limit vector, like [0,0] or 0 or [] is ignored, and an automatic graph limit is used.

**Example:**  *p.limits{n} = {[t1,t2],[x1,x2],[y1,y2]...,}*

### 10.7.16  *linestyle{n}*

**Default:**  *{'-k','--k',':k','-.k','-ok','--ok',':ok','-.ok','-+k','--+k'}*

Line types for each line in every two-dimensional graph plotted. If a given line on a two-dimensional line is to be removed completely, set the relevant line-style to zero. For example, to remove the first line from graph 3, set p.linestyle{3} ={0}. This is useful when generating and changing graphics output from a saved data file. The linestyle uses Matlab terminology. It allows setting the line pattern, marker symbols and color for every line. The default lines are black ('k'), but any other color can be used instead.

    The specifiers must be chosen from the list below, eg, '-ok', although the marker can be omitted if not required.

- Line patterns: '-' (solid), '–' (dashed), ':' (dotted) ,'-.' (dash-dot)

- Marker symbols: '+','o','*','.','x','s','d','^','v','>','<','p'

- Colors: 'r','g','b','c','m','y','k','w'

**Example:**  *p.linestyle{4} = {'-k','--ok',':g','-.b',}*

### 10.7.17  *linewidth{n}*

**Default:** 0.5

Line width for plotted lines in two-dimensional graphs. For example, to make the lines wider in graph 3, set p.linewidth{3} =1. This is useful for changing graphics output appearance if the default lines are too thin.

**Example:** *p.linewidth{n} = 1*

### 10.7.18  *minbar{n}*

**Default:** *{0.01, ...}*

This is the minimum relative error-bar that is plotted. Set to a large value to suppress unwanted error-bars, although its best not to ignore the error-bar information! This can be changed per graph.

**Example:** *p.minbar{n} = 0*

### 10.7.19  *name*

**Default:** ”

Name used to label simulation graphs, usually corresponding to the equation or problem solved. This can be removed from individual graphs by using *headers{n}* equal to a single blank space. The default is a null string. To remove all headers globally, set *name* equal to a single blank space: *name = ' '*.

**Example:** *p.name = 'Wiener process simulation'*

### 10.7.20  *olabels{n}*

**Default:** *'a'*

Cell array of labels for the graph axis observables and functions. These are text labels that are used on the graph axes. The default value is *'a_1'* if the default observable is used, otherwise it is blank. This is overwritten by any subsequent label input when the graphics program is run:

**Example:** *p.olabels{4} = 'v'*

### 10.7.21  *parametric{n}*

**Default:** *[0,0]*

Cell array that defines parametric plots, for each graph number. The first number is the graph number of the alternative observable plotted on the horizontal axis. The second number is the axis number where the parametric value is substituted, which can be the time (axis 1) or the x-coordinate (axis 2), if present.

   If both are zero, the plot against an independent space-time coordinate is calculated as usual. If nonzero, a parametric plot is made for two-dimensional plots. In all cases the vertical axis is used

to plot the original data. The specified horizontal axis is used for the parametric variable. Only vertical error-bars are available. Can be usefully combined with *scatters{n}* to plot individual trajectories, but the number of scatters should be the same in each of the two graphs that are parametrically plotted against each other.

**Example:** *p.parametric{n} = [p1,p2] >= 0*

### 10.7.22   *pdimension{n}*

**Default:** *3*

This is the maximum plotted space-time dimension for each plotted quantity. The purpose is eliminate unwanted graphs. For example, it is useful to reduce the maximum dimension when averaging in space. Higher dimensional graphs are not needed, as the data is duplicated. Averaging can be useful for checking conservation laws, or for averaging over homogeneous data to reduce sampling errors. All graphs are suppressed if it is set to zero. Any three dimensions can be chosen to be plotted, using the *axes* parameter to suppress the unwanted data points in other dimensions.

**Example:** *p.pdimension{4} = 2*

### 10.7.23   *saveeps*

**Default:** 0

If set to 1, all plots are saved to the current folder as .eps files, numbered consecutively. It is best to use the *close all* command first to remove unwanted displayed xFIGURES, before running *xgraph* with this option.

**Example:** *p.saveeps =1*

### 10.7.24   *savefig*

**Default:** 0

If set to 1, all plots are saved to the current folder as .fig files, numbered consecutively. It is best to use the *close all* command first to remove unwanted displayed xFIGURES, before running *xgraph* with this option.

**Example:** *p.savefig =1*

### 10.7.25   *transverse{n}*

**Default:** *0*

This is the number of 2D transverse images plotted as discrete time slices. Only valid if *dimensions* is greater than 2. If present, the $y, z$-coordinates are set to their central values when plotting transverse images. Each element can be from 0 up to the number of plotted time-points. The cell array has a vector length equal to *graphs*.

**Example:** *p.transverse{n}= 6*

### 10.7.26  *verbose*

**Default:** 0

Print flag for output information while running xGRAPH. Print options are:

- Minimal if *verbose = -1*: Prints just the start-up time and hard error messages

- Brief if *verbose = 0*: Additionally prints the final, total chi-squared errors where present

- Informative if *verbose = 1*: Also prints the graph progress indicators

- Full if *verbose = 2*: Prints everything including the internal parameter structure data.

In summary, if *verbose = 0*, most output is suppressed except the final data, *verbose = 1* displays a progress report, and *verbose = 2* additionally generates a readable summary of the graphics parameter input.

**Example:** *p.verbose = 0*

### 10.7.27  *xlabels*

**Default:** *{'t', 'x', 'y', 'z'}* or *{'x_1', 'x_2', 'x_3', 'x_4',...}*

Global labels for the independent variable labels, vector length equal to *dimensions*. The numerical labeling default is used when the *numberaxis* option is true. These are typeset in Latex mathematics mode. When changing from the default values, all the required new labels must be set.

**Example:** *p.xlabels = {'tau'}*

## 10.8   User function reference

It is possible to simply run *xgraph* as is, without much intervention. However, there are customization options, including user defined functions. These are as follows:

### 10.8.1  *gfunction{n} (d,p)*

This is a cell array of graphics function handles. Use when a graph is needed that is a functional transformation of the observed averages. The default value generates the *n-th* graph *data* array directly from the *n-th* input *data*. The input is the data cell array for all the graphs in the current sequence number with their graph parameters *x*, and the output is the *n-th* data array that is plotted.

   An arbitrary number of functions of these observables can be plotted, including vector observables. The input to graphics functions is the observed data averages or functions of averages in a given sequence, each stored in a cell array $d\{n\}(\ell, \mathbf{j}, c)$. If there are more graphics functions than input data cells, this generate additional data for plotting.

### 10.8.2 *xfunctions{n} {nd} (ax,p)*

This is a nested cell array of axis transformations. Use when a graph is needed with an axis that is a function of the original axes. The input is the original axis coordinates, and the output is the new coordinate set. The default value generates the input axes. Called as *xfunctions{n}{nd}(ax,p)* for the *n*-th graph and axis direction *dir*, where *ax* is a vector of coordinates for that axis.There is one graphics function for each separate graph dimension or axis. The default value is the coordinate vector $xk\{nd\}$ stored in the input parameter structure p, or else the relevant index if *xk{nd}* is omitted.

## 10.9 xGRAPH structure

The graphics function, $xgraph$, plots the simulation data. The general structure is:

$$\textbf{xgraph} \rightarrow \textbf{xgpreferences} \; (checks\,inputs)$$
$$\rightarrow \textbf{xmultigraph} \leftrightarrow \textbf{xreduce} \leftrightarrow \textbf{xcompress} \; (structures\,data\,arrays)$$
$$\rightarrow \textbf{ximages} \rightarrow \textbf{xtransverse} \rightarrow \textbf{xplot3} \rightarrow \textbf{xplot2} \; (graphs\,all\,data)$$

Most graphics functions simply work, but two important functions are listed here for reference.

### 10.9.1 *xgraph(data,input)*

The *xgraph* function graphs multidimensional data files.

- Input: graphics data cells *data*, input parameter cells *input*.

- Output: graphs, displayed and/or stored as *eps* or *fig* files.

- If no numeric *data* present, reads data from a file named *data*.

- If *data* is present but without any *input* parameters it plots using default parameters.

- First data dimension is the line index, last dimension are the error-bars and comparisons

- Needs: *xread, xmakecell, xgpreferences, xmultiplot*

### 10.9.2 *input = xgpreferences (input,oldinput)*

The *xgpreferences* function sets default values for graphics inputs.

- Input: *input* cell array and optionally previous inputs from a datafile, *oldinput*.

- Note that each cell array is a sequence of graphics parameter structures

- Output: the updated plus default graphics parameters

- Called by: *xgraph*

- Needs: *xprefer, xcprefer*

# 11 Examples

A variety of examples are given in the xAMPLES folder distributed with xSPDE. These can all be run using *Batchtest.m*, which has a typical runtime of $50-100s$, and runs 34 different case studies. This shows your distribution is intact. All the graphs produced are deleted. It also lists the many different templates available, some of which are listed here.

## 11.1 SDE examples

### 11.1.1 Kubo

This solves a multiplicative SDE with initial condition $a(0) = 1$ and:

$$\frac{\partial a}{\partial t} = iaw(t). \tag{218}$$

The function uses the RK4 algorithm together with both vector and series ensembles, then stores the computed averages with a comparison of the variance and an exact solution,

$$\langle a^n \rangle = e^{-tn^2/2}.$$

```
function [e] = Kubo()
p.name = 'Kubo oscillator';
p.ensembles = [1000,8];
p.method = @RK4;
p.initial = @(w,p) 1;
p.deriv = @(a,w,p) 1i*w.*a(1,:)   ;
p.file = 'Kubo.mat';
p.observe{2} = @(a,p) a.^2;
p.olabels{2} = {'<a^2>'};
p.compare = {@(p) exp(-p.t/2),@(p) exp(-2*p.t)};
e = xsim(p);
p2.name = 'Kubo oscillator edited title';
xgraph(p.file,p2);
end
```

**Notes**

- The algorithm is changed from the default to RK4.

- The data is stored to 'Kubo.mat'.

- This is re-read and edited using a second parameter structure, p2.

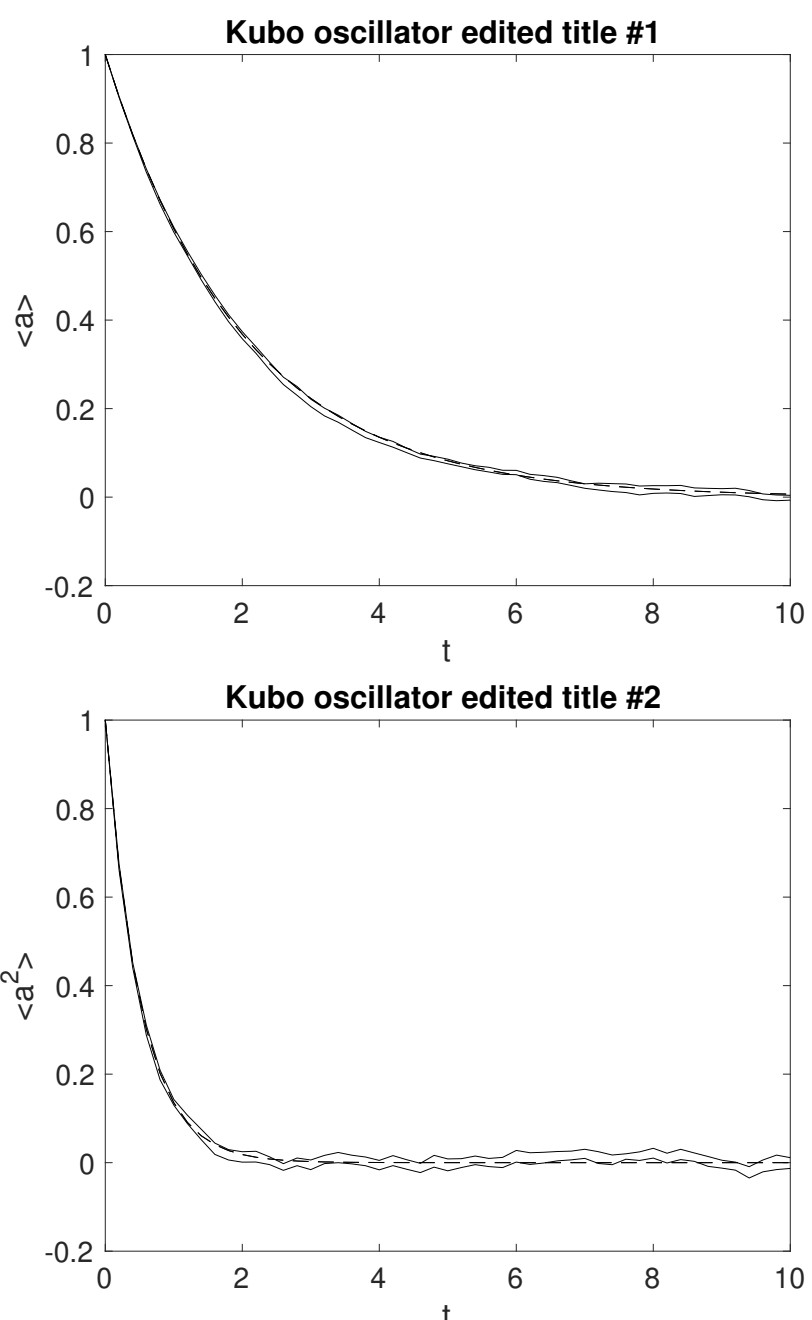

Figure 12: Example: Kubo oscillator. The graph shows the sampling error-bars as two parallel lines. The discretization error-bars are less than the minimum, and are not shown.

### 11.1.2  Loss/Gain with noise

This solves an SDE with a complex Gaussian distributed initial condition having $\langle|a(0)|^2\rangle = 1$ and a sequence of SDE equations, such that

$$\frac{\partial a}{\partial t} = \begin{cases} -a + w_1(t) + iw_2(t) & 0 < t < 4 \\ a + w_1(t) + iw_2(t) & 4 < t < 8 \end{cases}. \tag{219}$$

The computed variance is compared with an exact solution,

$$\langle a^2 \rangle = \begin{cases} 1 & 0 < t < 4 \\ 2e^{2(t-4)t} - 1 & 4 < t < 8 \end{cases}.$$

.

```
function [e] = Gain()
p.name = 'Loss with noise';
p.ranges = 4;
p.noises = [2,0];
p.ensembles = [10000,1,10];
p.initial = @(w,~) (w(1,:)+1i*w(2,:))/sqrt(2);
p.deriv = @(a,w,p) -a + w(1,:)+1i*w(2,:);
p.observe = {@(a,~) a.*conj(a)};
p.olabels = {'|a|^2'};
p.compare = {@(p) 1};
p2 = p;
p2.steps = 2;
p2.name = 'Gain with noise';
p2.deriv = @(a,w,~) a + w(1,:)+1i*w(2,:);
p2.compare = {@(p) 2*exp(2*p.t)-1};
e = xspde({p,p2});
end
```

**Notes**

- Low and high level parallel ensembles optimize use of multi-core vector hardware.

- Two distinct simulations are run in series, with a change in the equation.

- The simulation name is changed in sequence 2, to distinguish the graphical outputs

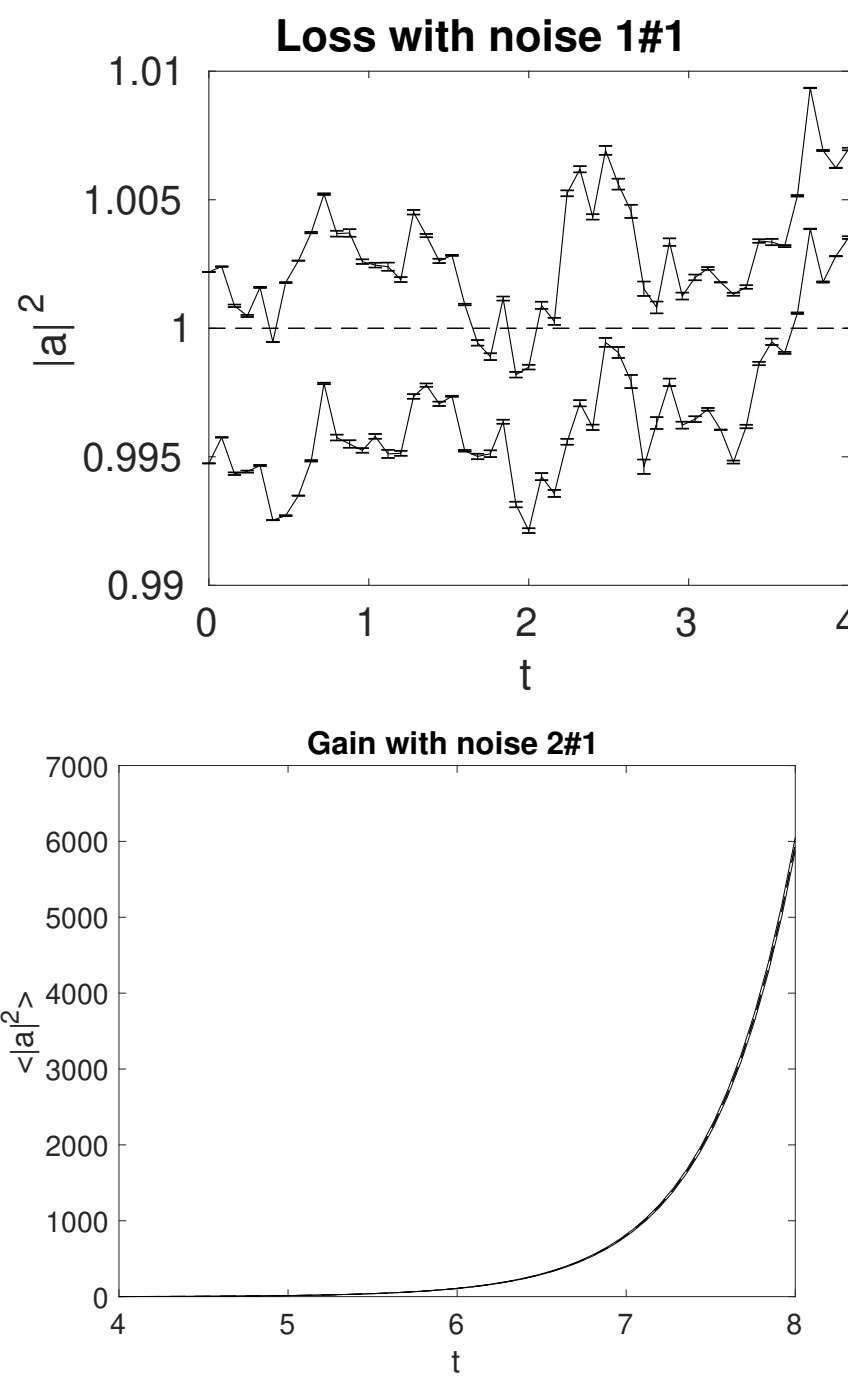

Figure 13: Top figure: amplitude squared with loss balanced by noise. Bottom figure, amplitude squared with gain. Graphs show excellent agreement with theory up to the sampling errors of less than ±0.005 in the initial phase, shown by the parallel lines, with step errors of order ±0.001 indicated by error-bars.

## 11.2 Spectral examples

### 11.2.1 Equilibrium

This solves an SDE with a complex Gaussian initial condition having $\left\langle |a(0)|^2 \right\rangle = 1$ and:

$$\frac{\partial a}{\partial t} = -a + w_1(t) + iw_2(t). \tag{220}$$

The equation is such that the initial distribution is also the equilibrium probability distribution. The computed ordinary and spectral variances are compared with exact solutions and graphed, where

$$\lim_{t \to \infty} \left\langle |a(t)|^2 \right\rangle = 1.$$

$$\left\langle |a(\omega)|^2 \right\rangle = \frac{T}{\pi(1+\omega^2)}.$$

```
function [e] = Equilibrium()
p.name = 'Equilibrium spectrum';
p.points = 101;
p.ranges = 100;
p.seed = 241;
p.noises = [2,0];
p.ensembles = [100,5];
p.initial = @(w,~) (w(1,:)+1i*w(2,:))/sqrt(2);
p.deriv = @(a,w,~) -a + w(1,:)+1i*w(2,:);
p.observe{1} = @(a,~) a.*conj(a);
p.observe{2} = @(a,~) a.*conj(a);
p.transforms = {0,1};
p.olabels = {'|a(t)|^2','|a(\omega)|^2'};
p.compare = {@(p) 1, @(p)p.ranges(1)./(pi*(1+p.w.^2))};
e = xspde(p);
end
```

**Notes**

- A fixed random seed is input using the *p.seed* parameter.

- The *p.transforms* cell array gives a Fourier transform for *p.observe{2}* only.

- A small number of ensembles and time-steps is used to improve error visibility.

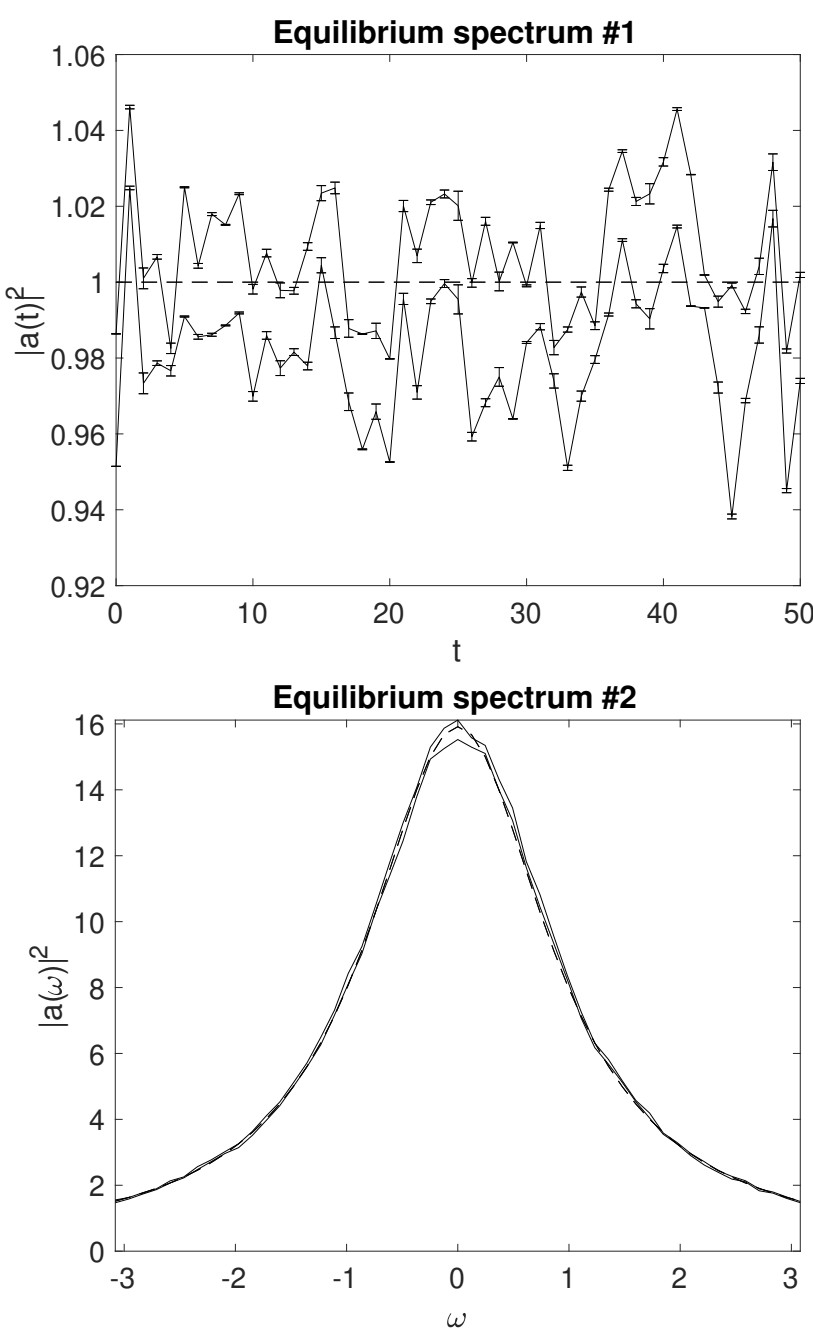

Figure 14: Top figure: Mean amplitude squared, showing invariant behavior with time, apart from sampling errors. Bottom figure: Mean spectrum as a function of frequency. The dashed lines are exact results, solid lines are upper and lower sampling error bounds ($\pm\sigma$), from sampling the stochastic equations, the error-bars are errors due to the step-size. Error bars are less than the minimum size for graphics display in the bottom figure.

### 11.2.2   Quantum

This solves an SDE for a quantum harmonic oscillator in the (truncated) Wigner phase-space calculus. It is initialized as a vacuum state, corresponding to a complex Gaussian initial condition having $\left\langle |a(0)|^2 \right\rangle = 1$. It is subject to vacuum noise, here realized by the auxiliary field $a_{in}$. An output field is given through the input-output relations and is realized by the auxiliary field $a_{out}$.

$$
\begin{aligned}
\frac{\partial a}{\partial t} &= -a + \sqrt{2}a_{in}. \\
a_{in} &= \frac{1}{2}\left(w_1(t) + iw_2(t)\right) \\
a_{out} &= \sqrt{2}a - a_{in}
\end{aligned}
\tag{221}
$$

The computed spectral variances are compared with exact solutions and graphed, where:

$$
\begin{aligned}
\frac{2\pi}{T}\left\langle |a(\omega)|^2 \right\rangle &= \frac{1}{(1+\omega^2)}. \\
\left\langle |a_{in}(\omega)|^2 \right\rangle &= \frac{1}{2} \\
\left\langle |a_{out}(\omega)|^2 \right\rangle &= \frac{1}{2}.
\end{aligned}
\tag{222}
$$

**Notes**

- Demonstrates how to include defined fields

- There are 4 steps per point, to give better accuracy due to finite steps

- The observe functions are all transformed, and include defined fields.

```
function e = Quantum()
p.name = 'Quantum harmonic oscillator spectrum';
p.points = 160;
p.steps = 4;
p.ranges = 120;
p.fields = 1;
p.auxfields = 2;
p.noises = 2;
p.ensembles = [400,1,12];
p.initial = @(w,~) (w(1,:)+1i*w(2,:))/(2);
p.a1 = @(w) (w(1,:)+1i*w(2,:))/2;
p.deriv = @(a,w,~) -a(1,:)+sqrt(2)*p.a1(w);
p.define = @(a,w,p) [p.a1(w);sqrt(2)*a(1,:)-p.a1(w)];
T = @(p) p.ranges(1);
p.observe{1} = @(a,p) (2.*pi/T(p))*a(1,:).*conj(a(1,:));
p.observe{2} = @(a,p) (2.*pi/T(p))*a(2,:).*conj(a(2,:));
p.observe{3} = @(a,p) (2.*pi/T(p))*a(3,:).*conj(a(3,:));
p.transforms = {1,1,1};
p.olabels{1} = '|a(\omega)|^2';
p.olabels{2} = '|a_{in}(\omega)|^2';
p.olabels{3} = '|a_{out}(\omega)|^2';
p.compare{1} = @(p) 1./(1+p.w.^2);
p.compare{2} = @(p) 0.5;
p.compare{3} = @(p) 0.5;
e = xspde(p);
end
```

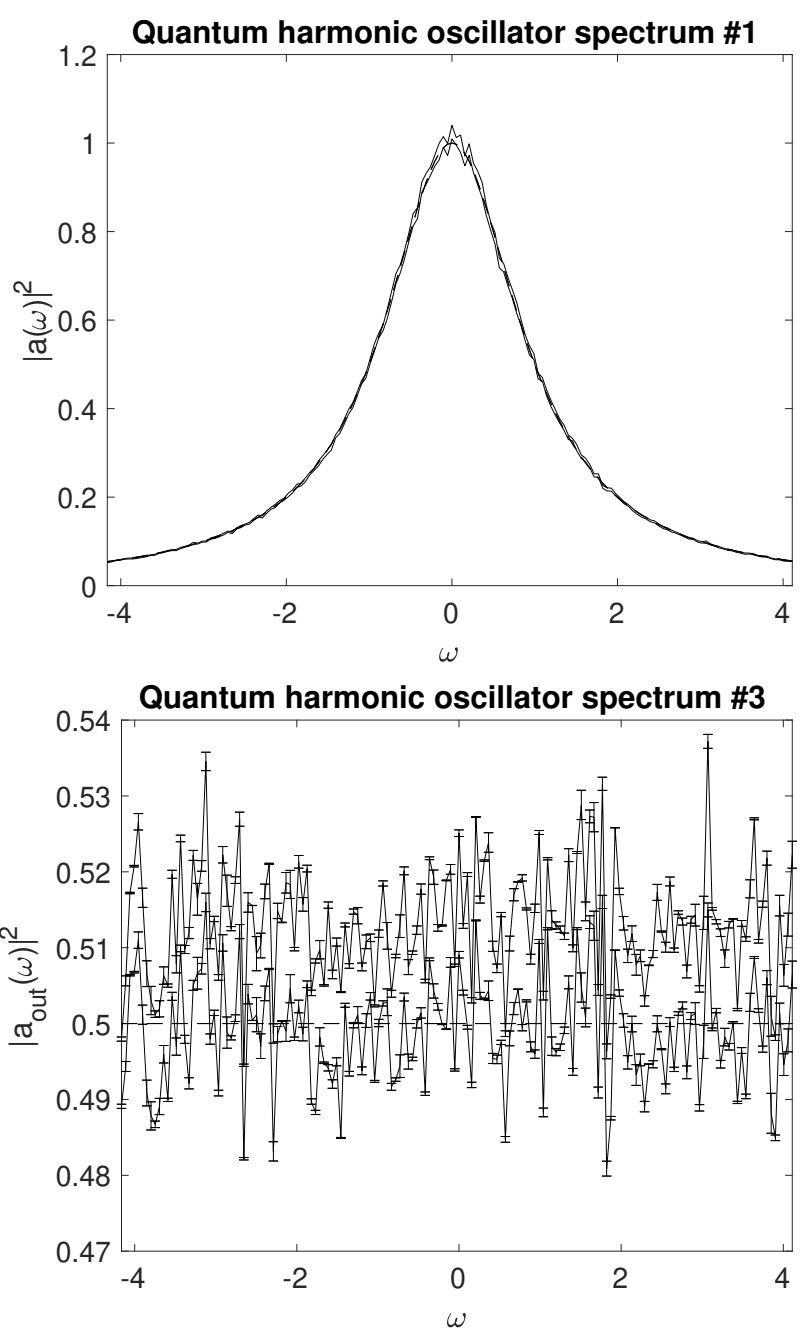

Figure 15: Top figure: Spectral density of the quantum state. Bottom figure: Spectral density of the output field. The solid lines indicate upper and lower sampling error bounds ($\pm\sigma$), from sampling the stochastic equations. The dashed lines are exact results, the error-bars indicate step-size errors. Error bars are less than the minimum size for display in the top figure.

## 11.3 Probability examples

### 11.3.1 Probability density, Wiener process

Solves an SDE with an initial condition $\langle a(0) \rangle^2 = \frac{1}{4}$ and

$$\dot{a} = w(t). \tag{223}$$

Records the probability density and compares this with an exact solution:

$$
\begin{aligned}
P(x,t) &= \frac{1}{\sqrt{2\pi\sigma^2(t)}} e^{-\frac{x^2}{2\sigma^2(t)}} \\
\sigma^2(t) &= \frac{1}{4} + t.
\end{aligned}
\tag{224}
$$

**Notes**

- The script outputs a 3D plot of $P(x,t)$, together with the time evolution of $P(0,t)$

- There are 5 "transverse" plots of transient probabilities at intermediate times.

- Legends are plotted to identify the simulated and the analytic comparison lines.

```
function e = Wienerprob()
p.name = 'Wiener SDE distribution';
p.noises = 1;
p.points = 10;
p.ensembles = [10000,10];
p.initial = @(v,p) v/2;
p.sig = @(p) .25 + p.r{1};
p.deriv = @(a,w,p) w;
p.observe{1} = @(a,p) a;
p.compare{1} = @gaussprob;
p.transverse{1} = 5;
p.olabels{1} = 'P(x)';
p.binranges{1} = {-5:0.25:5};
p.legends{1} = {'Sampled P(x,\tau) \pm \sigma','Exact
P(x,\tau)'};
p.xlabels = {'\tau','x'};
e = xspde(p);
end
%
function p = gaussprob(p)
p = exp(-(p.r{2}.^2)./(2*p.sig(p)))./sqrt(2*pi*p.sig(p));
end
```

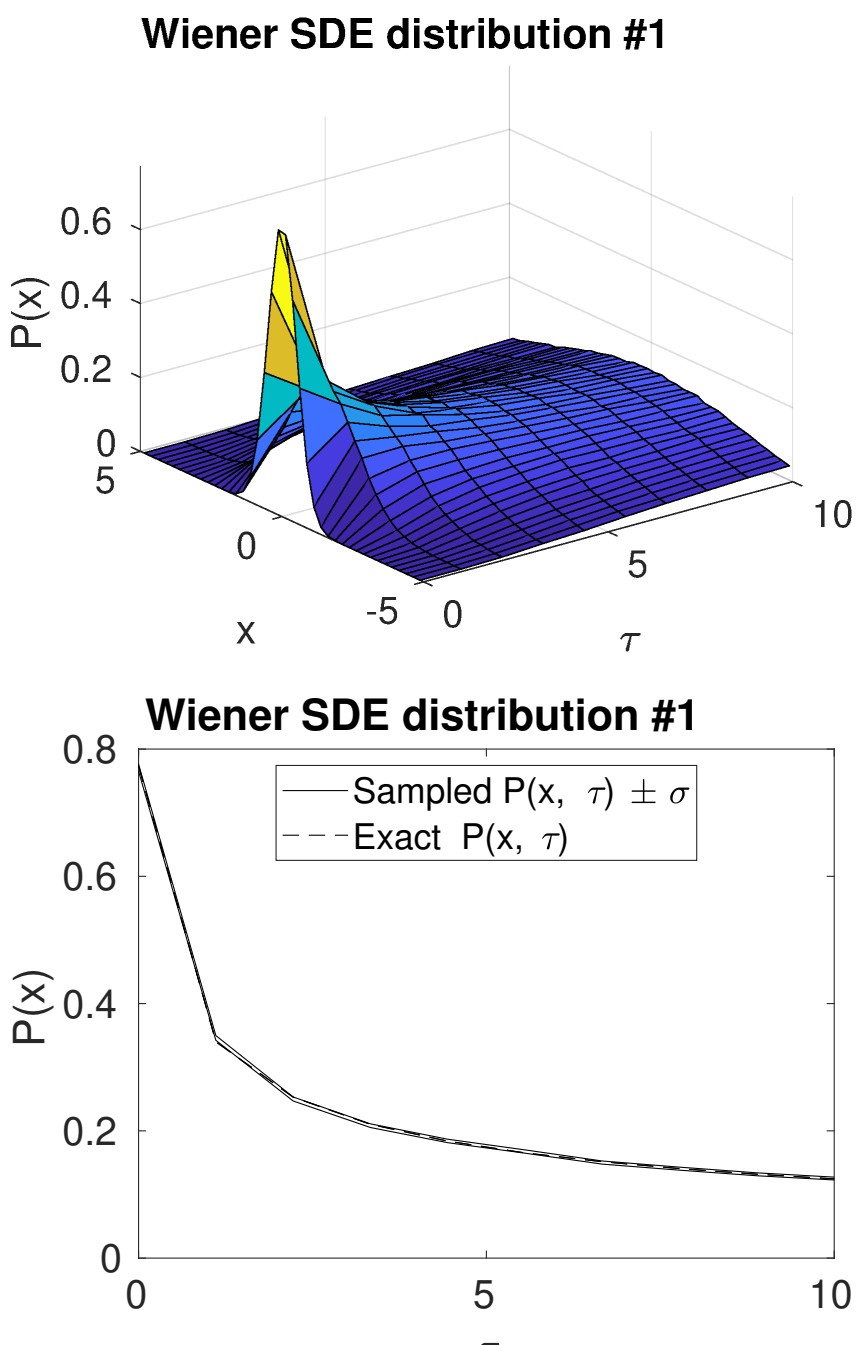

Figure 16: Top figure: 3D plot of the computed probability density of the simulated Wiener process as a function of time ($\tau$) and "position" ($x$). Bottom figure: Time evolution of the computed probability density for $x = 0$. The solid lines indicate upper and lower sampling error bounds, while the dashed line indicates theoretical predictions.

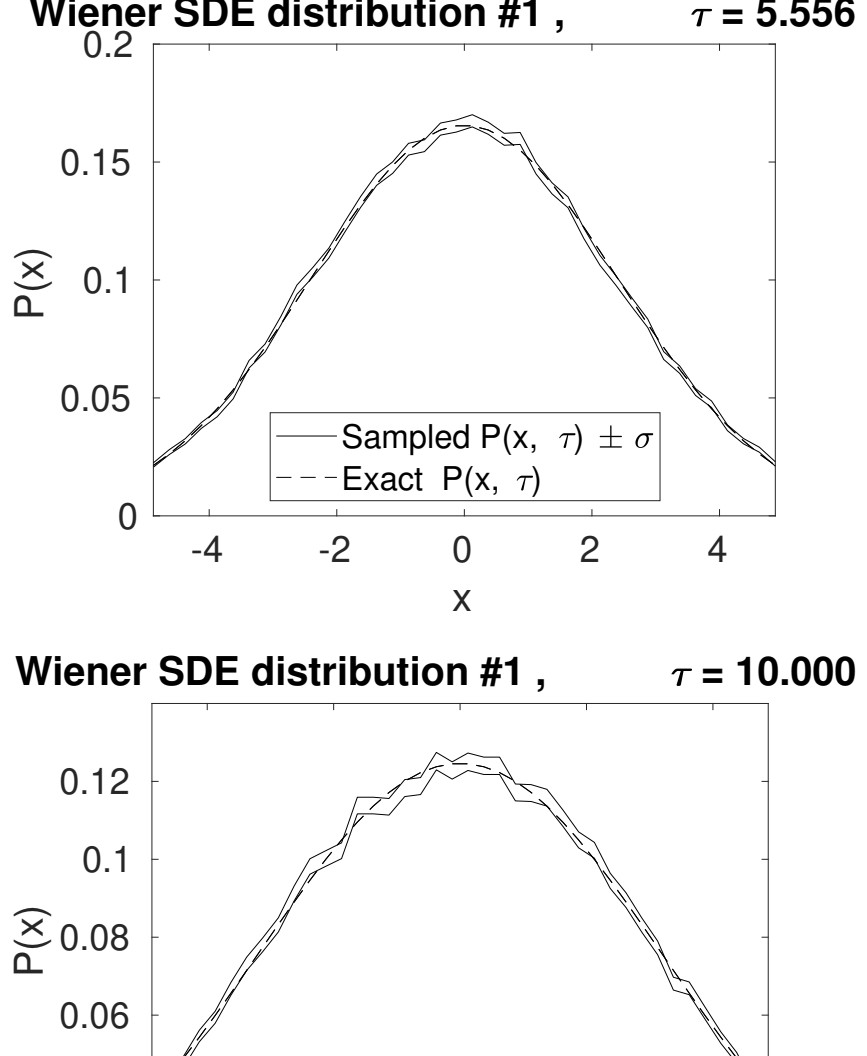

Figure 17: Top and bottom figure: Computed probability densities of the simulated Wiener process at $\tau = 5.556$ and $\tau = 10$, respectively. In total, 5 of these transverse plots are generated, however, only 2 are presented here.

## 11.4   SPDE examples

### 11.4.1   Nonlinear Schrodinger equation with Dirichlet boundary conditions

This solves a (1+1)-dimensional PSDE with an initial condition of $a(t=0,x) = sech(x)$ and

$$\frac{\partial a}{\partial t} = i \cdot \left( a \cdot \left( |a|^2 - \frac{1}{2} \right) + \frac{1}{2} \frac{\partial^2 a}{\partial x^2} \right). \tag{225}$$

The solution is subject to Neumann boundary conditions with boundary values at zero

$$a_x(t, \pm x_m) = 0. \tag{226}$$

The equation is a deterministic nonlinear Schrodinger equation, which applies to nonlinear optics, Bose-Einstein condensates and plasma physics. The observables are $o_1 \equiv |a|^2$ and $o_2 \equiv \int_{-x_m}^{x_m} \left| \frac{\partial}{\partial x} a \right|^2 dx$,

**Notes**

- The boundary conditions are specified with *p.boundaries*

```
function [e] = SolitonDerivN()
p.dimensions = 2;
p.points = [101,101];
p.ranges = [10,15];
p.initial = @(v,p) sech(p.x);
p.observe{1} = @(a,p) a.*conj(a);
p.observe{2} = @(a,p) Int(abs(D1(a,2,p)).^2,p);
p.olabels = {'|a|^2','\int |da/dx|^2 dx'};
p.name = 'NLS soliton:spectral method + Neumann';
p.boundaries{2} = [-1,-1];
p.transverse = {3};
p.deriv = @(a,~,p) 1i*a.*(conj(a).*a);
p.linear = @(p) 0.5*1i*(p.Dx.^2-1);
e = xspde(p);
end
```

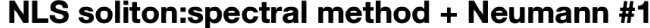

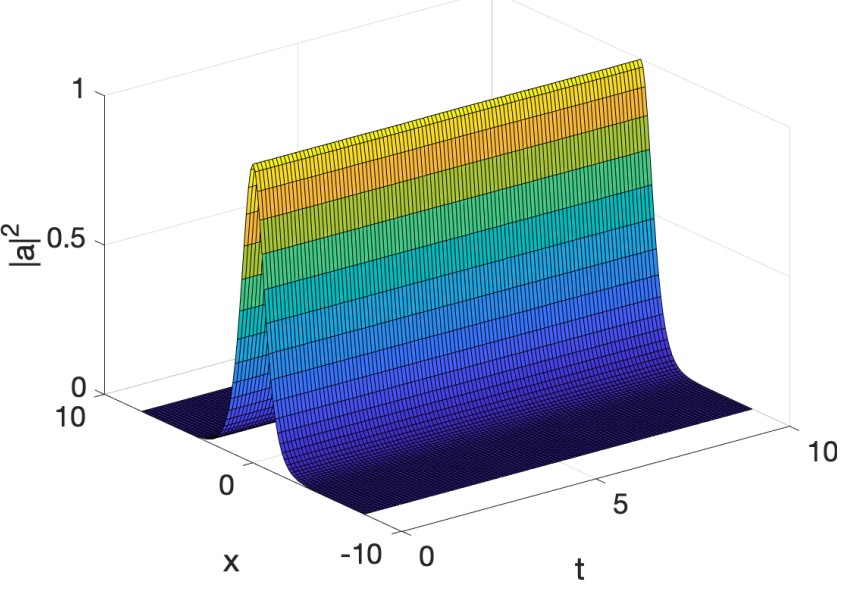

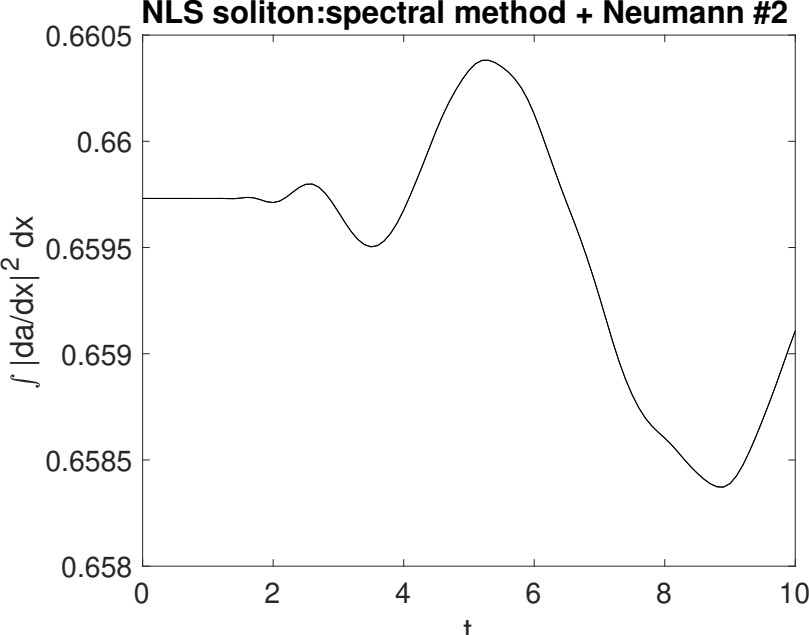

Figure 18: Top figure: 3D plot of the solution for $|a|^2$ as a function of time and position. Bottom figure: time evolution of $\int_{-x_m}^{x_m} \left| \frac{\partial}{\partial x} a \right|^2 dx$.

### 11.4.2 Planar noise growth

This solves a (1+2)-dimensional PSDE describing the growth of noise in an extended planar vector field with additive noise and a diffraction term giving rise to noise dispersion. There are 240 trajectories in the total ensemble. The equation is:

$$\frac{\partial \mathbf{a}}{\partial t} = \frac{i}{2}\left(\frac{\partial^2}{\partial x^2} + \frac{\partial^2}{\partial x^2}\right)\mathbf{a} + \eta(t, x). \tag{227}$$

The initial conditions are that $\mathbf{a} = (\mathbf{v}_x + i\mathbf{v}_y)/\sqrt{2}$, where:

$$\left\langle v_i(\mathbf{x}) v_j(\mathbf{x}')\right\rangle = \delta(\mathbf{x} - \mathbf{x}')\delta_{ij}$$

the noise correlations are that $\eta = (\mathbf{w}_x + i\mathbf{w}_y)/\sqrt{2}$, where:

$$\left\langle w_i(\mathbf{r}) w_j(\mathbf{r}')\right\rangle = \delta(t - t')\delta_{ij}(\mathbf{x} - \mathbf{x}') \tag{228}$$

The solution is subject to periodic boundary conditions. The noise correlations for the second field are specified in momentum space. As there are no filters specified, the noise terms are delta-correlated in both momentum ($\mathbf{k}$) and in space ($x$). Using a Fourier transform, one can show that the exact results for comparison for the correlations within each field are similar in position and momentum space:

$$\left\langle |a_i(t, \mathbf{x})|^2\right\rangle = (1 + t)/\Delta A_x.$$
$$\left\langle |a_i(t, \mathbf{k})|^2\right\rangle = (1 + t)/\Delta A_k.$$

Here, $\Delta A_{x,k}$ is the area of a lattice cell in space or momentum space respectively. This is $\Delta A_x = 1/49$ for the parameters used here. On integration over the whole lattice, the correlation is proportional to $N_s$, the number of points in the spatial lattice, which is $35^2 = 1225$ for the default spatial lattice used here:

$$\int \left\langle |a_i(t, \mathbf{x})|^2\right\rangle d\mathbf{x} = \int \left\langle |a_i(t, \mathbf{k})|^2\right\rangle d\mathbf{k} = N_s(1 + t).$$

**Notes**

- All three types of ensemble are used

- The much lower sampling error after integration is evident in the graphs

- Spatially resolved graphs show larger sampling errors

```
function [e] = Planar()
p.name = 'Planar noise growth';
p.dimensions = 3;
p.fields = 2;
p.ranges = [1,5,5];
p.points = 10;
p.noises = [2,2];
p.ensembles = [10,2,12];
p.initial = @Initial;
p.deriv = @D_planar;
p.linear = @Linear;
p.observe{1} = @(a,p) Int(a(1,:).*conj(a(1,:)),p);
p.observe{2} = @(a,p) Int(a(2,:).*conj(a(2,:)),p.dk,p);
p.observe{3} = @(a,p) real(Ave(a(1,:).*conj(a(2,:)),p));
p.observe{4} = @(a,p) a(2,:).*conj(a(2,:));
p.transforms = {[0,0,0],[0,1,1],[0,1,1]};
p.olabels{1} = '<\int|a_1(x)|^2 d^2x>';
p.olabels{2} = '<\int|a_2(k)|^2 d^2k>';
p.olabels{3} = '« a_1(k)a^*_2(k)»';
p.olabels{4} = '<|a_2(x)|^2>';
p.compare{1} = @(p) (1+p.t)*p.nspace;
p.compare{2} = @(p) (1+p.t)*p.nspace;
p.compare{3} = @(p) 0.0;
e = xspde(p);
end

function a0 = Initial(v,~)
a0(1,:)  = (v(1,:)+1i*v(2,:))/sqrt(2);
a0(2,:)  = (v(3,:)+1i*v(4,:))/sqrt(2);
end

function da = D_planar(~,w,~) %%Derivatives
da(1,:)  = (w(1,:)+1i*w(2,:))/sqrt(2);
da(2,:)  = (w(3,:)+1i*w(4,:))/sqrt(2);
end

function L = Linear(p)
lap = p.Dx.^2+p.Dy.^2;
L(1,:)  = 1i*0.5*lap(:);
L(2,:)  = 1i*0.5*lap(:);
end
```

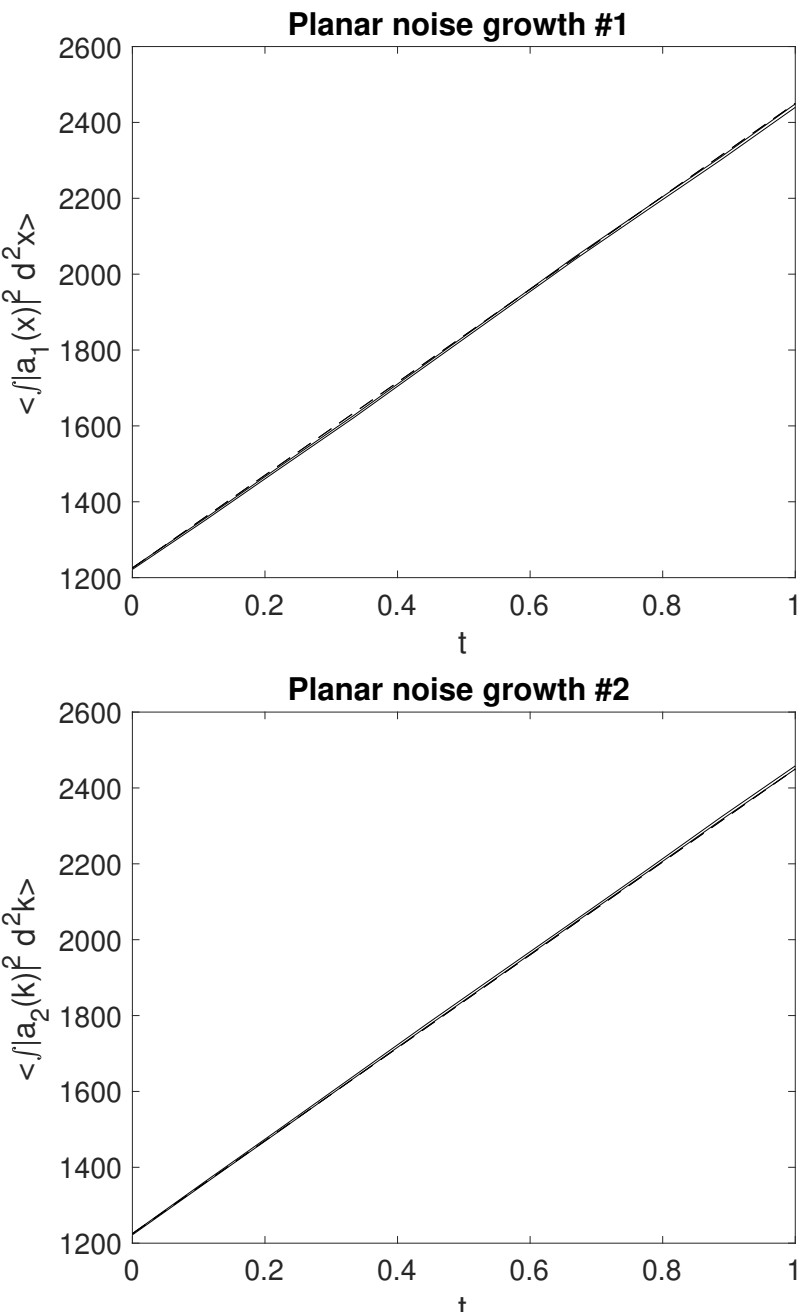

Figure 19: Top and bottom figure: Time evolution of the integrated modulus square of the first and second field, respectively. The solid lines indicate upper and lower bounds of the stochastic error, which the dashed lines indicate theoretical predictions.

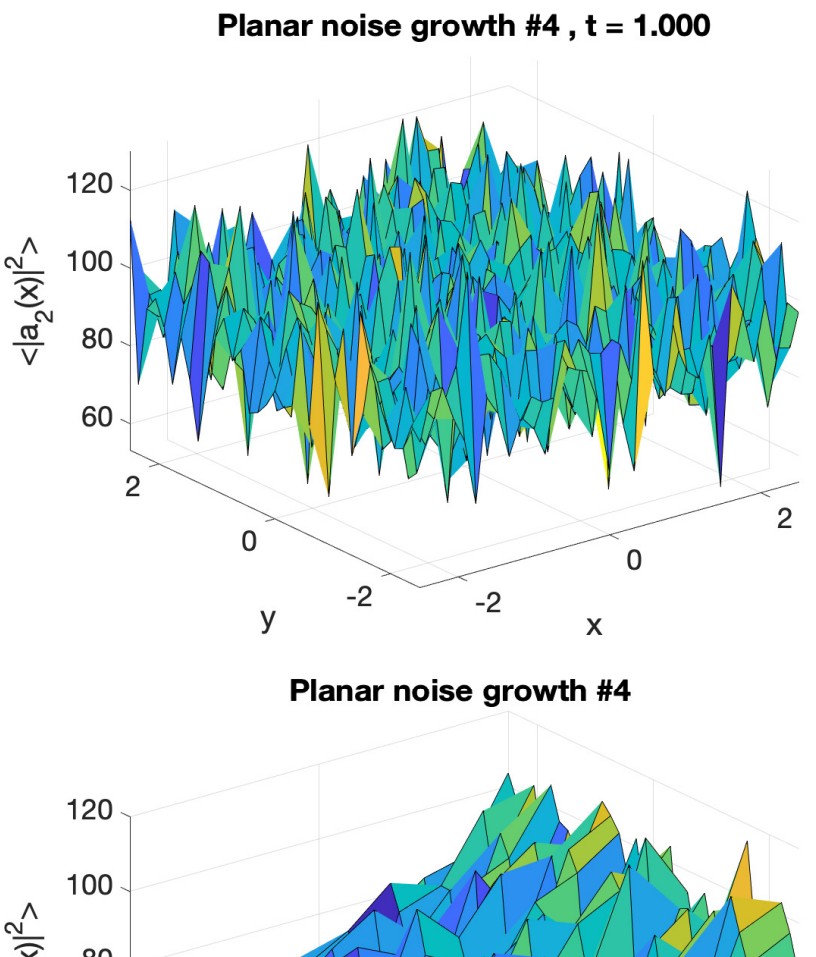

Figure 20: Top figure: 3D plot of the modulus square of $a_2$ at $t = 1$ as a function of $x$ and $y$. Bottom figure: 3D plot of the modulus square of $a_2$ for $y = 0$ as a function of $x$ and $t$.

### 11.4.3   Gross-Pitaevskii equation with vortex formation

This solves a (1+2)-dimensional PDE called the Gross-Pitaevskii equation. In addition to the standard GPE terms, it includes the vortex forming term $(\mathbf{x} \times \nabla)\, a$. There is just one ensemble member, to demonstrate how a single trajectory can be imaged. The equation is:

$$
\begin{aligned}
\frac{\partial a}{\partial t} &= \left( \frac{1}{2}\nabla^2 a - \left\| \left( \left( V(\mathbf{x}) + 200\,|a|^2 \right) + 0.6i \cdot (\mathbf{x} \times \nabla) \right) a \right\| \right) \\
V(\mathbf{x}) &= 0.35 \left( x^2 + y^2 \right) \\
\|b(\mathbf{x})\| &= \frac{b(\mathbf{x})}{\int |b|^2 \, d\mathbf{x}}.
\end{aligned}
\tag{229}
$$

Here, $\|\cdot\|$ is the normalized derivative and $\times$ indicates the two-dimensional cross-product. The system is initialized as

$$
a(t=0, \mathbf{x}) = 0.1 \cdot \exp\left( -V(\mathbf{x}) \right).
\tag{230}
$$

**Notes**

- This is a deterministic partial differential equation case

- The 15 intermediate *steps* used are necessary to reduce integration errors

- The trap potential is an inline function, and is not a parameter

- Normalization is used because otherwise particle number is not conserved

- The output includes transverse *images* to show how the vortices develop

- Different *imagetypes* are used to show different 3D features

```
function [e] = GPEvortex2D()
p.name = 'GPEvortex2D';
p.dimensions = 3;
p.fields = 1;
p.points = [50,40,40];
p.ranges = [15,16,16];
p.steps = 15;
g = 200;
om = 0.6;
L = @(a,p) 1i*(p.x.*D1(a,3,p)-p.y.*D1(a,2,p));
V = @(p) 0.35*(p.x.^2+p.y.^2);
p.initial = @(v,p) 0.1*exp(-V(p));
rho = @(a) g*conj(a).*a;
p.deriv = @normda;
p.da1 = @(a,w,p) -a.*(V(p)+rho(a))+om*L(a,p);
p.linear = @(p) 0.5*(p.Dx.^2+p.Dy.^2);
p.observe{1} = @(a,p) a(1,:).*conj(a(1,:));
p.observe{2} = @(a,p) a(1,:).*conj(a(1,:));
p.images = {2,2};
p.imagetype = {1,2};
p.olabels = {'|a|^2','|a|^2'};
e = xspde(p);

function b = normda(a,w,p)
% b = NORMDA(a,z,r) is a normalized derivative
% Takes a derivative and returns a normalized step
b = a+p.da1(a,w,p)*p.dtr;
norm = sqrt(Int(abs(b).^2,p.dx,p));
b = (b./norm-a)/p.dtr;
end
end
```

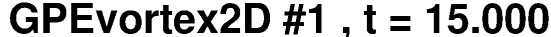

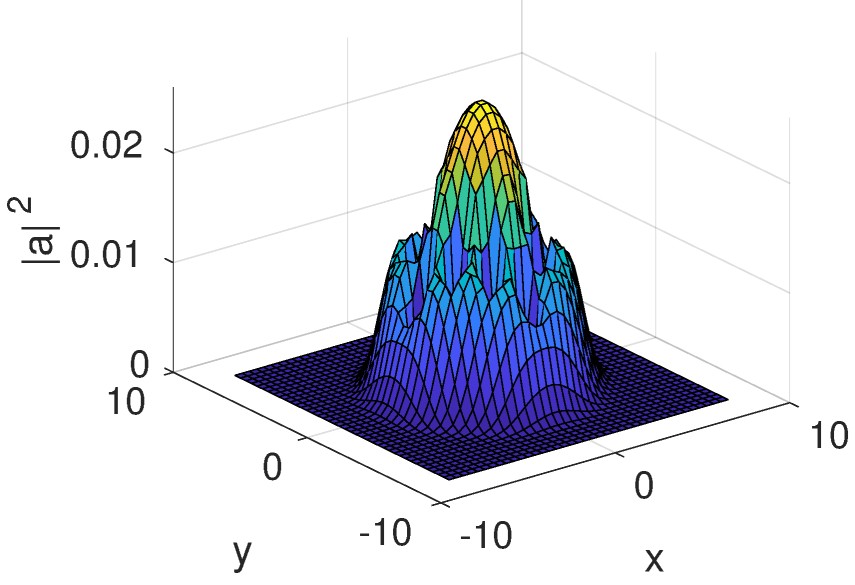

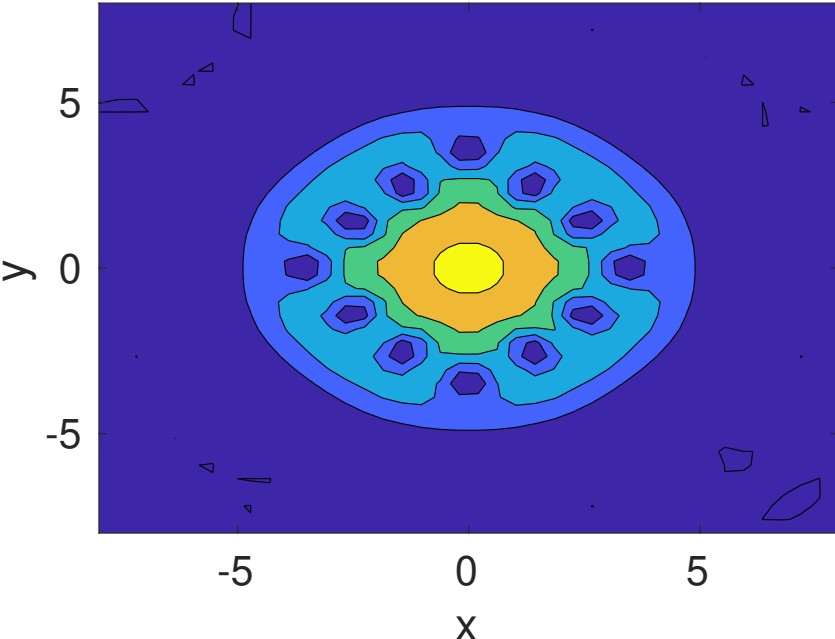

Figure 21: Top and bottom figure: The computed solution for $|a|^2$ at $t = 15$ as a function of $x, y$ as a 3D plot (top) and as a color map (bottom).

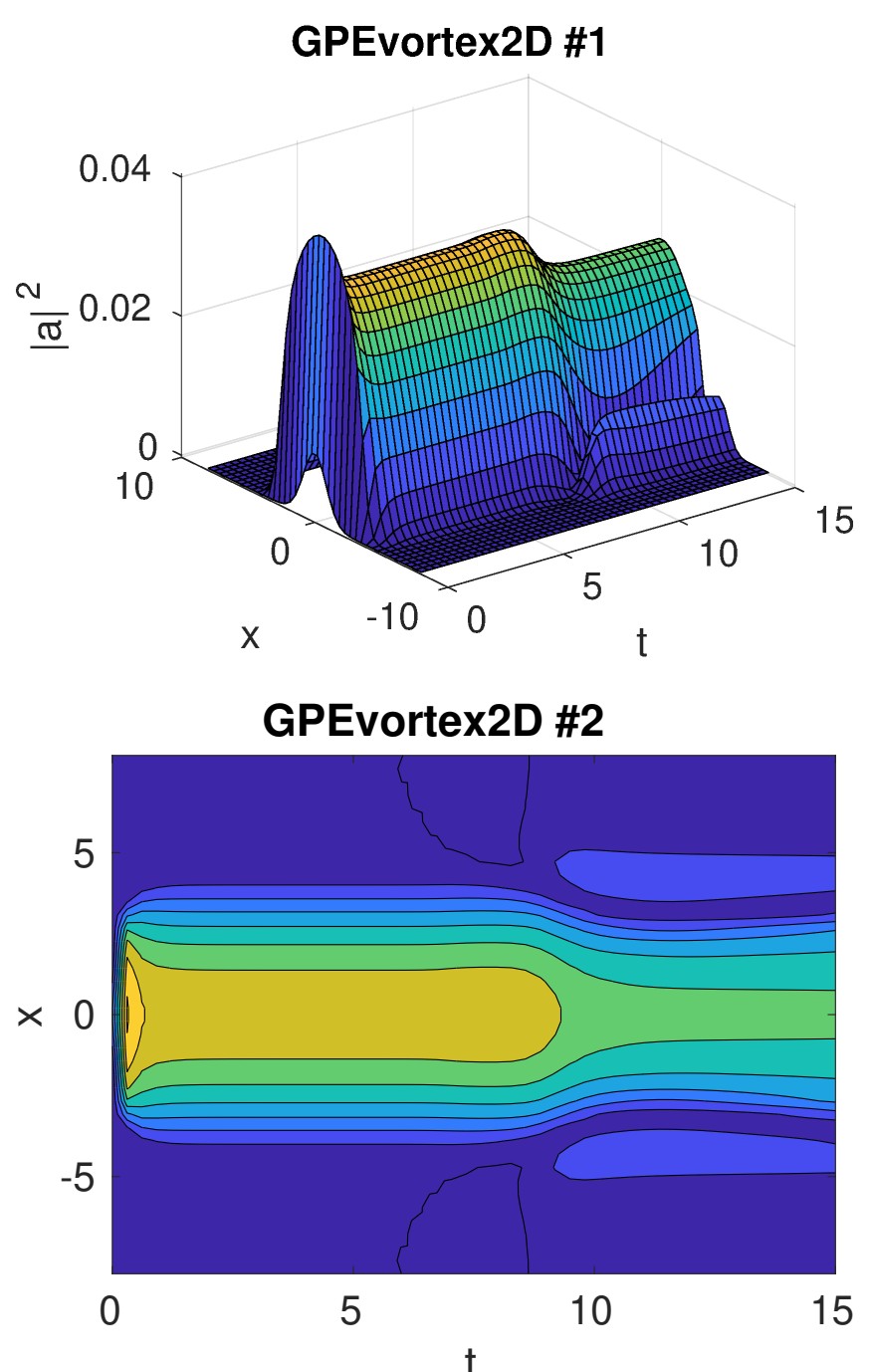

Figure 22: Top and bottom figure: The computed solution for $|a|^2$ for $y = 0$ as a function of $x, t$ as a 3D plot (top) and as a color map (bottom).

### 11.4.4    Heat equation with finite-difference and propagators

This very simple example solves a (1+1)-dimensional PDE with an initial condition of $\mathbf{a}(t=0,x)=\mathbf{f}(x)$ and

$$\frac{\partial \mathbf{a}}{\partial t} = \frac{\partial^2 \mathbf{a}}{\partial x^2}. \tag{231}$$

The solution is subject to periodic boundary conditions or Dirichlet and/or Neumann with boundary values at zero, so that $a(t,\pm x_m)=0$ or $\partial a/\partial x(t,\pm x_m)=0$. Each component has different combinations of boundary types. Using spectral methods the solutions here are exact, up to round-off errors of order $10^{-15}$, and are also much faster than with finite differences, which is demonstrated in the example.

     In all cases the grid range is from $x=0$ to $x=\pi$, and the time duration is from $t=0$ to $t=4$. In the examples, the spectral propagation error is reduced by more than $10^{10}$ and the time is reduced by a factor of 20 compared to the finite-difference methods. The periodic method has boundaries just outside the grid.

**Dirichlet-Dirichlet**    Here $a_x(0)=a_x(\pi)=0$, then the exact solution has the form:

$$a = \sum_{n=1}^{\infty} S_n \sin(nx) e^{-n^2 t}. \tag{232}$$

Suppose that

$$a(x,0) = 4\sin(x) + \sin(2x), \tag{233}$$

For this case:

$$a(x,t) = 4\sin(x) e^{-t} + \sin(2x) e^{-4t}. \tag{234}$$

**Neumann-Neumann**    with $a(0)=a(\pi)=0$, then the exact solution has the form:

$$a = \sum_{n=0}^{\infty} C_n \cos(nx) e^{-n^2 t}. \tag{235}$$

Suppose that

$$a(x,0) = 5 + 4\cos(x) + \cos(2x), \tag{236}$$

For this case:

$$a(x,t) = 5 + 4\cos(x) e^{-t} + \cos(2x) e^{-4t}. \tag{237}$$

**Dirichlet-Neumann**    Here $a(0)=a_x(\pi)=0$, then the exact solution has the form:

$$a = \sum_{n=1}^{\infty} S_n \sin((2n-1)x/2) e^{-(2n-1)^2 t/4}. \tag{238}$$

Suppose that

$$a(x,0) = 4\sin(x/2) + \sin(3x/2), \tag{239}$$

For this case:

$$u(x,0) = 4\sin(x/2) e^{-t/4} + \sin(3x/2) e^{-9t/4}. \tag{240}$$

**Neumann-Dirichlet** Here $a_x(0) = a(\pi) = 0$, then the general solution has the form:

$$a = \sum_{n=1}^{\infty} C_n \cos\left((2n-1)x/2\right) e^{-(2n-1)^2 t/4}. \tag{241}$$

Suppose that

$$a(x,0) = 4\cos\left(x/2\right) + \cos\left(3x/2\right). \tag{242}$$

For this case:

$$a(x,t) = 4\cos\left(x/2\right) e^{-t/4} + \cos\left(3x/2\right) e^{-9t/4}. \tag{243}$$

**Periodic** Here $a(0) = a(\epsilon\pi)$, where $\epsilon = N/(N-1)$ accounts for the periodic boundaries being outside the grid range, then the general solution has the form:

$$a = \sum_{n=1}^{\infty} S_n \sin\left(2nx\right) e^{-4n^2 t/\epsilon^2}$$

$$+ \sum_{n=0}^{\infty} C_n \cos\left(2nx\right) e^{-4n^2 t/\epsilon^2}. \tag{244}$$

Suppose that

$$a(x,0) = 2 + \cos\left(2x/\epsilon\right) + \sin\left(4x/\epsilon\right). \tag{245}$$

For this case:

$$u(x,0) = 2 + 2\cos\left(2x/\epsilon\right) e^{-4t/\epsilon^2} + \sin\left(4x/\epsilon\right) e^{-16t/\epsilon^2}. \tag{246}$$

**Notes**

- This is another deterministic pde case, although noise can be added

- Different boundary conditions apply to each component

- Sequential integration is used, but the initial condition is just recycled.

- In *p1*, the 40 intermediate *steps* are necessary to reduce finite-difference errors

```
function [e] = Boundaries()
p.dimensions = 2;
p.points = [51,51];
p.order = 0;
p.verbose = 1;
p.fields = 5;
p.ranges = [4,pi];
p.origins = [0,0];
p.initial = @heat_in;
p.observe = {@(a,p) a(1,:),@(a,p) a(2,:),@(a,p) a(3,:)...
@(a,p) a(4,:),@(a,p) a(5,:)};
p.compare = {@heat_1,@heat_2,@heat_3,@heat_4,@heat_5};
p.diffplot = {1,1,1,1,1};
p.olabels = {'a, DD','a, NN','a, DN','a, ND','a, PP'};
p.name = 'Heat test, spectral';
p.boundaries{2}= [1,1;-1,-1;1,-1;-1,1;0,0];
p.linear = @(p) p.Dx.^2;
p1 = p;
p1.linear = @(p) [];
p1.deriv = @(a,w,p) D2(a,2,p);
p1.steps = 40;
p1.transfer = @(~,p,~,~) heat_in(0,p);
p1.name = 'Heat test, finite diffs';
e = xspde({p,p1});
end

function a = heat_in(~,p)
a(1,:)  = 4*sin(p.x)+sin(2*p.x);
a(2,:)  = 5+4*cos(p.x)+cos(2*p.x);
a(3,:)  = 4*sin(p.x/2)+sin(3*p.x/2);
a(4,:)  = 4*cos(p.x/2)+cos(3*p.x/2);
a(5,:)  = 2+cos(2*p.x/1.02)+sin(4*p.x/1.02);
end

function o = heat_1(p)
o = 4*sin(p.x).*exp(-p.t)+sin(2*p.x).*exp(-4*p.t);
end
function o = heat_2(p)
 o = 5+4*cos(p.x).*exp(-p.t)+cos(2*p.x).*exp(-4*p.t);
end
function o = heat_3(p)
 o = 4*sin(p.x/2).*exp(-p.t/4)+sin(3*p.x/2).*exp(-9*p.t/4);
end
function o = heat_4(p)
 o = 4*cos(p.x/2).*exp(-p.t/4)+cos(3*p.x/2).*exp(-9*p.t/4);
end
function o = heat_5(p)
o = 2+cos(2*p.x/1.02).*exp(-4*p.t/1.02^2)+...
 sin(4*p.x/1.02).*exp(-16*p.t/1.02^2);
end
```

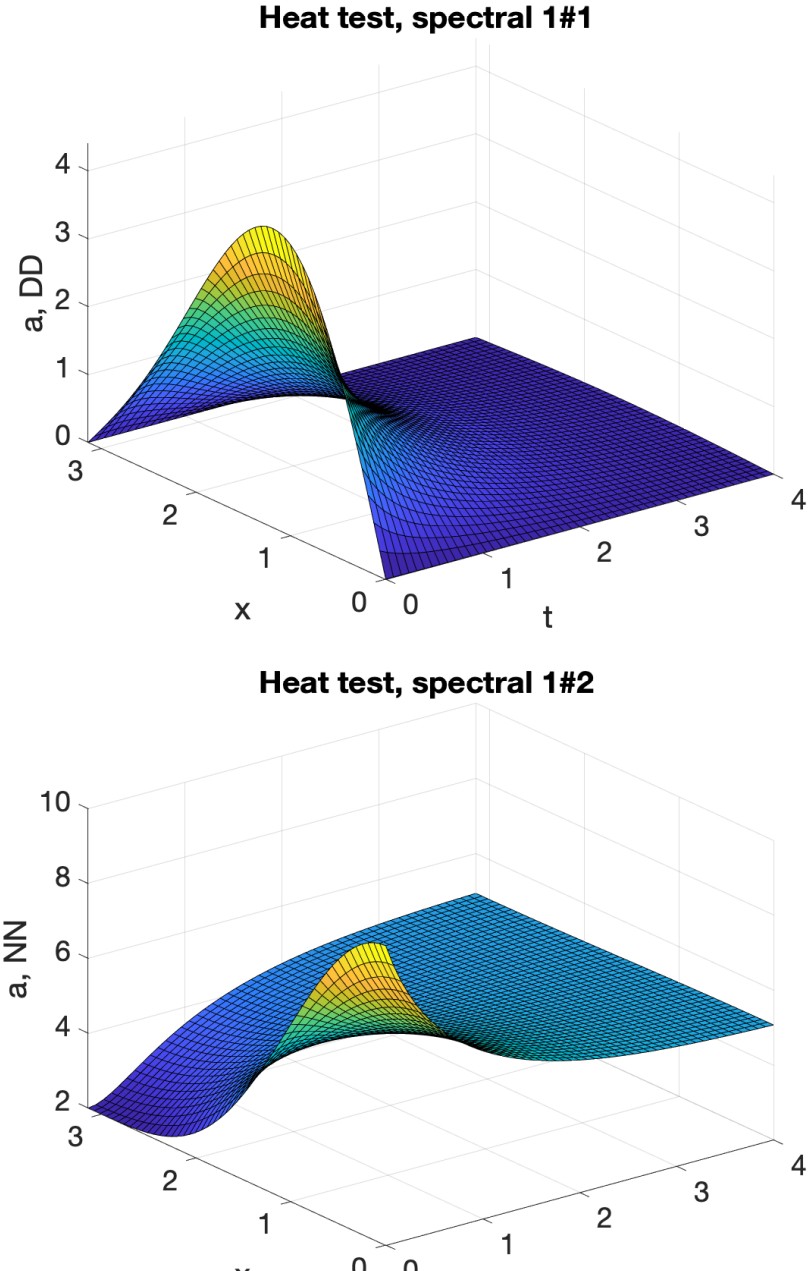

Figure 23: Top figure: Spectral solution for $a$ as a function of time and position with Dirichlet-Dirichlet boundaries. Bottom figure: Plot of the solution with Neumann-Neumann boundaries.

## 11.5 Projection examples

### 11.5.1 SDE with catenoid projection

This solves an SDE with 3 field variables $\mathbf{a} = (a_1, a_2, a_3)^T$. The Stratonovich diffusion equation is

$$\frac{\partial \mathbf{a}}{\partial t} = \mathcal{P}_{\mathbf{a}}^{\|}[\mathbf{w}], \tag{247}$$

where $\mathcal{P}_{\mathbf{a}}^{\|}[\cdot]$ indicates a projected onto the surface of a catenoid manifold defined by

$$f = x_1^2 + x_2^2 - \sinh^2(x_3) - 1 = 0. \tag{248}$$

The initial condition is given by $\mathbf{a}(o) = (1, 0, 0)^T$. Here $\mathbf{w} = (w_1, w_2, w_3)^T$ consists of 3 independent noise variables

**Notes**

- This is a projected sde case

- The Euclidean distance from the initial point is computed

- This is compared with the predicted analytic value $\langle R^2 \rangle = 2t$.

```
function [e] = Catenoid
p.name = '3D Catenoid diffusion';
p.iterproj = 3;
p.X0 = [1,0,0]';
p.fields = 3;
p.ranges = 5;
p.points = 51;
p.ensembles = [400, 10];
p.compare{2} = @(p) 2*p.t;
p.deriv = @(a, w, p) w;
p.initial = @(w, p) p.X0;
p.observe{2} = @(a, p) sum((p.X0-a).^2,1);
p.diffplot{2} = 1;
p.function{1} = @(o, p) o{2}.^2;
p.olabels = {'\langle R^2 \rangle^2','\langle R^2 \rangle'};
p.project = @Catproj;
p.method = @MPnproj;
e = xspde(p);
end
```

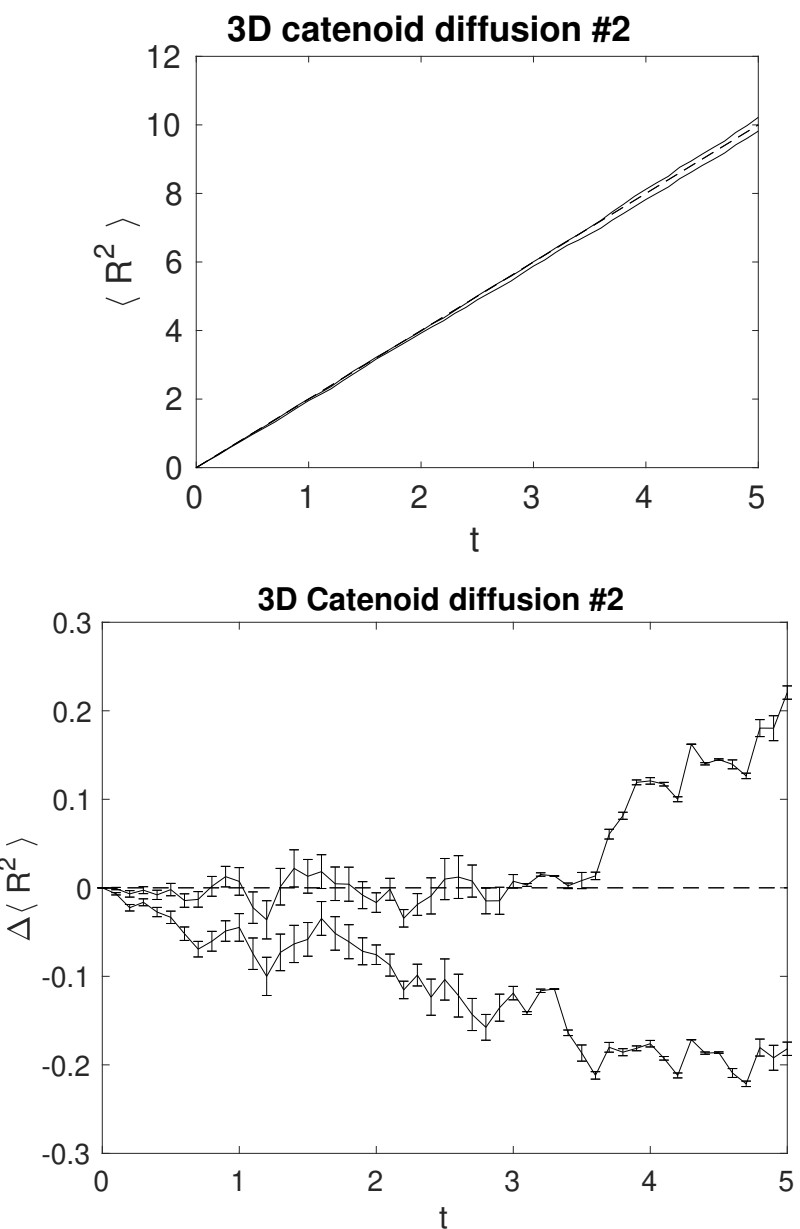

Figure 24: Top figure: Computed time evolution of the catenoid squared Euclidean diffusion distance $|\mathbf{x}_0 - \mathbf{x}(t)|^2$, where $\mathbf{x}_0 = (1,0,0)^T$, as a function of time. Bottom figure: Differences between the computed time evolution of the catenoid squared Euclidean distance $|\mathbf{x}_0 - \mathbf{x}(t)|^2$ and the exact result. In both cases the solid lines indicate the stochastic error bounds, error-bars are the time-step errors, while the dashed line indicates the theoretical prediction.

## 12 Conclusion

We have introduced a framework compatible with Matlab and Octave for the numerical integration of an extremely wide variety of stochastic differential equations. As we hope has become clear from this manual, xSPDE offers an extraordinarily clear and user-friendly interface. Simulating stochastic partial differential equations (SPDEs) can be achieved with virtually no extra effort compared to ordinary stochastic differential equations. Spatial derivatives can be obtained in an efficient and numerically stable way using Fourier or trigonometric transform techniques for many cases.

Finite difference methods are available in more complex cases, and different combinations of boundary values can be imposed independently on any vector component and in any spatial dimension. A number of algorithms for the integration of stochastic equations are included, while the modular and transparent design of xSPDE makes it easy to implement other custom algorithms as well if preferred.

Error estimations are included, and parallel operation is supported. Simulation results are available both as graphical output and in an interoperable data format. Importantly, the discretization and sampling error estimates are carried out at the level of the final estimated probabilities and averages, providing much more compact and useful results than raw data. However, raw data output is also available if needed.

With xSPDE, we aim to make stochastic simulations, particularly those involving SPDEs, accessible to scientists, engineers and researchers. The software has been used in several research works [8, 10, 11, 13–19, 21, 23, 25, 26, 41, 97]. Currently, the use of more efficient spectral methods is limited to periodic boundaries, zero value Dirichlet, and/or zero derivative Neumann boundary conditions.

In future releases, we aim to support a more general class of boundary conditions and integration methods, discrete networks, and more complex problems involving multiple spatial grids with different dimensions. A Julia version is under development for those users without Matlab licenses who find Octave too slow. Currently, xSPDE simulations are exclusively CPU or cluster-based, which is suitable and efficient for many purposes. Support for GPU-enabled simulations is a longer term goal for xSPDE.

### Acknowledgements

We would like to thank the many users and researchers whose feedback was invaluable, including Rodney Polkinghorne, Bogdan Opanchuk, King Ng, Alex Dellios, Run Yan Teh, Manushan Thenabadu, Margaret Reid, Jesse van Rhijn and Thomas Rodriguez.

**Funding information**   This work was funded through the Australian Research Council Discovery Project scheme under Grants DP180102470 and DP190101480. The authors also wish to thank NTT Research for their financial and technical support.

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
