# Peer review of "xSPDE3: extensible software for stochastic ordinary and partial differential equations"

_SciPost Physics Codebases, doi:SciPost Phys. Codebases 17 (2023) , SciPost Phys. Codebases 17-r3.44 (2023)_

## Round 1 · Referee Report · Said Rodriguez (Referee 1) · 2023-1-31

Strengths
- Excellent presentation of an excellent computational toolbox for solving stochastic ordinary and partial differential equations
- Very well written and organized manuscript
- Fair amount of theoretical background
- Plenty of examples illustrating the capabilities of the toolbox
- Sufficient technical details about algorithms, analysis tools, and other options of the toolbox
- The xSPDE software itself is powerful, user friendly, and easy to build upon
Weaknesses
- No significant weaknesses
- There are some minor points that could be improved. I discuss these in the list of requested changes.
Report
While the authors have my unconditional support for the publication of this work in SciPost Physics, I would still like to make a few suggestions. I provide these in the list of "requested changes".
Requested changes
-
For those of us familiar with xSPDE, which was much more briefly presented in a previous publication in SoftwareX, the main question is: What is new in xSPDE3? What are the main improvements and additions? It would be good to have their essence mentioned in the abstract and introduction, and then described in a bit more detail wherever relevant in the manuscript.
-
Equation 38, one of the simplest models used to introduce the capabilities of the toolbox, is presented as a "damped quantum harmonic oscillator". There are a couple of minor semantics issues here. First, there is nothing "quantum" about this oscillator. It is a simple oscillator whose statistical properties are fully described by classical physics. Second, there is no discussion about the damping constant in this equation. Of course, I understand that a is complex and its real part describes the damping and the imaginary part describes the oscillation frequency. But it would be good if the authors could explain this in 1 or 2 sentences . The extra clarity could be especially useful for students, since this is one of the first problems they will encounter in the manuscript.
-
Related to the previous point: Right below Equation 46 , the authors argue that the system (the same one of a simple classical oscillator) is essentially quantum mechanical. I disagree. There is nothing quantum in this system. Moreover, similar input-output relations as in Equations 45 and 46 hold for purely classical systems.
-
The example in section 4.7.5. seems to be a new one. It is great that the authors included it! However, there are too few details for the vast majority of users to understand what is going on here. The authors give a few generic references to books where similar equations are used. But it would be very valuable if the authors could also explain, in this manuscript, where these equations come from. To begin, what does each term describe? what is a1, a2, c, and lambda? The equations seem to describe two damped oscillators which are coupled both linearly and nonlinearly. In the parlance of optics, the nonlinear coupling would be called a cross-Kerr nonlinearity. Can the authors confirm that? And what is the relevance of the particular type of multiplicative noise they assume? Then, the authors say that one can use this model to investigate purely quantum effects like entanglement, EPR paradoxes, and Bell violations. I'm sorry, but I do not understand how that is possible. There are only 2 equations for mean fields plus fluctuations. I guess these equations can be derived from a full quantum model by applying the truncated Wigner approximation (TWA). But then, if that is the case, how do quantum effects come about? I was under the impression that the TWA cannot capture purely quantum effects like entanglement, Wigner negativities, etc. But maybe my assumption is wrong. If so, it would be valuable for the authors to explain this, since I think my assumption is shared by many in the community.
-
Having used xSPDE myself and in my group, one problem we haven't solved is how to access the noise vector. Sometimes, one would like to know the exact value of the noise field at each time step. Is there a way to retrieve that a posteriori?

---

## Round 2 · Referee Report · Said Rodriguez (Referee 1) · 2023-4-20

Strengths

See my previous report for a list of strengths.

Weaknesses

No weaknesses.

Report

I read the authors' response and revised manuscript . The authors properly addressed all my questions and concerns. I appreciate the various improvements to the manuscript which the authors made. The improvements include several more detailed explanations, new references, a good overview of what is new in this version of xSPDE, and additional technical details. I am confident that this manuscript will become a valuable resource for many researchers simulating stochastic systems. I recommend publication of the current version of this manuscript in SciPost.

---

## Round 2 · Author Response

xSPDE3: extensible software for stochastic ordinary and partial differential equations

by Simon Kiesewetter , Ria R. Joseph, Peter D. Drummond

Dear Scipost,

Thanks for the response on this paper. The referee has requested minor changes. We appreciate these very useful suggestions.

In response to the requests, we have made a number of changes, both in the code and in the manual. In summary, our response is as follows:

(1) What is new in xSPDE3?

A summary of all the 12 major innovations is now included in the Introduction.

(2a) Equation 38, ..is presented as a "damped quantum harmonic oscillator". .. First, there is nothing "quantum" about this oscillator....It is a simple oscillator whose statistical properties are fully described by classical physics.

Quantum phase-space representations were pioneered by Nobel prize winners Schrodinger, Wigner and Glauber. In this representation the harmonic oscillator treatment given is fully quantum mechanical. There is a new section (2.8) to explain this background, with references.

(2b) .. there is no discussion about the damping constant ...it would be good if the authors could explain this in 1 or 2 sentences .

An explanation of the damping parameters, together with the original master equation, is now included. We thank the referee for this helpful suggestion.

(3) There is nothing quantum in this system. Moreover, similar input-output relations as in Equations 45 and 46 hold for purely classical systems.

This mapping uses quantum phase-space methods. The size of the vacuum fluctuations, and their dependence on the operators, comes from quantum theory. These issues are now explained and referenced. Such techniques are used to analyse quantum technology experiments.

(4a) The equations seem to describe two damped oscillators which are coupled both linearly and nonlinearly. In the parlance of optics, the nonlinear coupling would be called a cross-Kerr nonlinearity. Can the authors confirm that? And what is the relevance of the particular type of multiplicative noise they assume?

A summary of the full derivation is now included. The nonlinearity is a parametric one, not a cross-Kerr nonlinearity. This type of nonlinearity is called a chi(2) effect. The noise terms are not assumed, but rather are derived rigorously from quantum mechanics.

(4b) ..the authors say that one can use this model to investigate purely quantum effects like entanglement, EPR paradoxes, and Bell violations...I was under the impression that the TWA cannot capture purely quantum effects like entanglement, Wigner negativities, etc. I'm sorry, but I do not understand how that is possible.

These can all be investigated using quantum phase-space stochastic equations. The relevant references are now included. Some quantum effects, like entanglement, can be treated using a truncated Wigner method, but the method used here is an exact one. These results are described in the referenced papers, and have been used in the quantum optics literature for many years. There are multiple textbooks and reviews available to describe how it is possible.

  1. Having used xSPDE myself and in my group, one problem we haven't solved is how to access the noise vector.

This is now available in XSPDE v3, using the p.auxfields defined parameter. When using error-checking, the noise terms that are accessed are the coarse time-step noises, to maintain compatibility with xSPDE output file standards. Otherwise, all noise terms are accessible.

The details of how to do this are now explained in section (4.6).

Yours sincerely,

Simon Kiesewetter , Ria R. Joseph, Peter D. Drummond

---

## Round 2 · List of Changes

To summarise the changes,

(1) The Introduction is rewritten, listing the new features.
(2) A more complete set of references was added to reference earlier work
(3) A new section (2.8) is included to explain quantum phase-space methods
(4) There is a new section (4.6) to explain auxiliary fields and noise outputs
(5) The quantum harmonic oscillator and nonlinear examples are rewritten
(6) Changes and improvements are included in the xSIM and xGRAPH reference

---

## Editorial Decision

published